# OPTIMAL NON-ASYMPTOTIC RATES OF VALUE ITERATION FOR AVERAGE-REWARD MDPS

**Jongmin Lee**
Seoul National University
Department of Mathematical Sciences
dlwhd2000@snu.ac.kr

**Ernest K. Ryu**
UCLA
Department of Mathematics
eryu@math.ucla.edu

## ABSTRACT

While there is an extensive body of research on the analysis of Value Iteration (VI) for discounted cumulative-reward MDPs, prior work on analyzing VI for (undiscounted) average-reward MDPs has been limited, and most prior results focus on asymptotic rates in terms of Bellman error. In this work, we conduct refined non-asymptotic analyses of average-reward MDPs, obtaining a collection of convergence results that advance our understanding of the setup. Among our new results, most notable are the $\mathcal{O}(1/k)$-rates of Anchored Value Iteration on the Bellman error under the multichain setup and the span-based complexity lower bound that matches the $\mathcal{O}(1/k)$ upper bound up to a constant factor of $8$ in the weakly communicating and unichain setups.

## 1 INTRODUCTION

Average-reward Markov decision processes (MDPs) are a fundamental framework for modeling decision-making, where the goal is to maximize long-term, steady-state performance. However, compared to the discounted cumulative-reward counterpart, the average-reward setup is more complex to analyze, and there has been less prior work on it. It is known that while iterates of VI diverge to infinity, the normalized iterates and the Bellman error converge to the optimal average reward under a certain aperiodicity condition. However, despite this understanding of convergence, quantifying the convergence *rates* of such methods for various classes of average-reward MDPs has been open.

In this work, we conduct refined non-asymptotic analyses of average-reward MDPs, obtaining a collection of convergence results advancing our understanding of the setup. Notably, we establish $\mathcal{O}(1/k)$ convergence rates of Anchored Value Iteration on the Bellman error under the multichain setup, and we present a span-based complexity lower bound that matches the $\mathcal{O}(1/k)$-upper bound up to a constant factor of $8$ in the weakly communicating and unichain setups.

| Prior works | Non-asym multi MDP | Asym multi MDP | Non-asym w.c. MDP | Asym w.c. MDP | Non-asym uni MDP | Asym uni MDP |
|---|---|---|---|---|---|---|
| [1, 2] | ✗ | ✗ | ✗ | ✗ | ✓ | ✓ |
| [3, 4] | ✗ | ✓ | ✗ | ✓ | ✓ | ✓ |
| [5] | ✗ | ✗ | ✗ | ✗ | ✓ | ✓ |
| Our work | ✓ > ✓ | ✓ | ✓ = ✓ | ✓ | ✓ = ✓ | ✓ |

Table 1: Summary of our contributions. (1: Federgruen et al. (1978), 2: Van Der Wal (1981), 3: Schweitzer & Federgruen (1977), 4: Schweitzer & Federgruen (1979), 5: Bertsekas (1998), 'Non-asym' stands for non-asymptotic convergence, 'Asym' for asymptotic convergence, 'multi' for multichain, 'w.c.' for weakly communicating, and 'uni' for unichain.) One check mark indicates a convergence result (upper bound) and two check marks with a strict inequality sign indicate a convergence result accompanied by a complexity lower bound but they do not match. Two check marks with an equal sign indicate a matching complexity lower bound. For multichain MDPs, we present the first non-asymptotic convergence result in Theorem 2. For weakly communicating MDPs, we present the first optimal complexity by matching the non-asymptotic convergence result in Corollary 2 with the complexity lower bound in Theorem 3.

## 1.1 NOTATION AND PRELIMINARIES

We quickly review basic definitions and concepts of average-reward Markov decision processes (MDPs) and reinforcement learning (RL). For further details, refer to standard references such as Puterman (2014); Bertsekas (2012); Sutton & Barto (2018b).

**Average-reward Markov decision processes.** Let $\mathcal{M}(\mathcal{X})$ be the space of probability distributions over $\mathcal{X}$. Write $(\mathcal{S}, \mathcal{A}, P, r)$ to denote the infinite-horizon undiscounted MDP with finite state space $\mathcal{S}$, finite action space $\mathcal{A}$, transition matrix $P \colon \mathcal{S} \times \mathcal{A} \to \mathcal{M}(\mathcal{S})$, and bounded reward $r \colon \mathcal{S} \times \mathcal{A} \to \mathbb{R}$. Denote $\pi \colon \mathcal{S} \to \mathcal{M}(\mathcal{A})$ for a policy, $g^\pi(s) = \liminf_{T \to \infty} \frac{1}{T} \mathbb{E}_\pi \left[ \sum_{t=0}^{T-1} r(s_t, a_t) \,|\, s_0 = s \right]$ for average-reward of a given policy, where $\mathbb{E}_\pi$ denotes the expected value over all trajectories $(s_0, a_0, s_1, a_1, \ldots, s_{T-1}, a_{T-1})$ induced by $P$ and $\pi$. We say $g^\star$ is optimal average reward if $g^\star(s) = \max_\pi g^\pi(s)$ for all $s \in \mathcal{S}$. We say $\pi$ is an $\epsilon$-optimal policy if $\|g^\star - g^\pi\|_\infty \leq \epsilon$.

**Value Iteration.** Let $\mathcal{F}(\mathcal{X})$ denote the space of bounded measurable real-valued functions over $\mathcal{X}$. With the given undiscounted MDP $(\mathcal{S}, \mathcal{A}, P, r)$, for $V \in \mathcal{F}(\mathcal{S})$, define the Bellman consistency operators $T^\pi$ as
$$T^\pi V(s) = \mathbb{E}_{a \sim \pi(\cdot \,|\, s), s' \sim P(\cdot \,|\, s, a)} \left[ r(s, a) + V(s') \right]$$
for all $s \in \mathcal{S}$, and the Bellman optimality operators $T$ as
$$TV(s) = \max_{a \in \mathcal{A}} \left\{ r(s, a) + \mathbb{E}_{s' \sim P(\cdot \,|\, s, a)} \left[ V(s') \right] \right\}$$
for all $s \in \mathcal{S}$. For notational conciseness, we write $T^\pi V = r^\pi + \mathcal{P}^\pi V$, where $r^\pi(s) = \mathbb{E}_{a \sim \pi(\cdot \,|\, s)} \left[ r(s, a) \right]$ is the reward induced by policy $\pi$ and
$$\mathcal{P}^\pi(s \to s') = \operatorname{Prob}(s \to s' \,|\, a \sim \pi(\cdot \,|\, s), s' \sim P(\cdot \,|\, s, a))$$
is transition matrix induced by policy $\pi$. We define the standard Value Iteration (VI) for the Bellman optimality operator as
$$V^k = TV^{k-1}, \qquad \text{for } k = 1, 2, \ldots,$$
where $V^0$ is an initial point. After executing $K$ iterations, VI returns the near-optimal policy $\pi_K$ as a greedy policy satisfying $T^{\pi_K} V^K = TV^K$.

**Fixed-point iterations.** Given an operator $T$, classical Banach fixed-point theorem (Banach, 1922) states that if $T$ is contractive, fixed point of $T$ exists and following *Picard iteration*
$$x^k = Tx^{k-1} \qquad \text{for } k = 1, 2, \ldots,$$
converges to the unique fixed point of $T$. If $T$ is nonexpansive but not contractive such as the rotation operator, Picard iteration may not converge to a fixed point. (For undiscounted MDPs, the Bellman optimality operator is nonexpansive but not necessarily contractive.) In such cases, one may use *Kransnosel'skiǐ-Mann iteration* (Mann, 1953; Krasnosel'skiǐ, 1955)
$$x^k = \lambda_k x^{k-1} + (1 - \lambda_k) Tx^{k-1} \qquad \text{for } k = 1, 2, \ldots,$$
where $\{\lambda_k\}_{k \in \mathbf{N}} \in [0, 1]$, or *Halpern iteration* (Halpern, 1967)
$$x^k = \lambda_k x^0 + (1 - \lambda_k) Tx^{k-1} \qquad \text{for } k = 1, 2, \ldots,$$
where $x^0$ is an initial point and $\{\lambda_k\}_{k \in \mathbf{N}} \in [0, 1]$, to guarantee convergence.

**Classification of MDPs.** MDPs are classified as follows by the structure of transition matrices. (For definitions of basic concepts of MDPs such as irreducible class, recurrent class, transient states, accessibility, etc., please refer to Puterman (2014, Appendix A.2).)

MDP is *unichain* if the transition matrix corresponding to every deterministic policy consists of a single irreducible recurrent class plus a possibly empty set of transient states. MDP is *weakly communicating*, if there exists a closed set of states where each state in that set is accessible from every other state in that set under some determinisitc policy, plus a possibly empty set of states which is transient under every policy. MDP is *multichain*, if the transition matrix corresponding to any deterministic policy contains one or more irreducible recurrent classes.

MDP is weakly communicating if MDP is unichain, and MDP is multichain if MDP is weakly communicating. Since every MDP is multichain, we use the expressions *multichain* and *general* interchangeably.

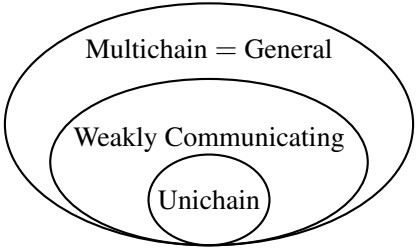

Figure 1: Classification of MDPs: Unichain $\subset$ Weakly Communicating $\subset$ Multichain (General)

**Modified Bellman equations.**   Following Puterman (2014, Section 9.1.1), we consider the *modified Bellman equations* defined as

$$\max_a \left\{ \sum_{s' \in \mathcal{S}} P(s' \,|\, s, a) g(s') \right\} = g(s), \qquad \max_a \left\{ r(s, a) + \sum_{s' \in \mathcal{S}} P(s' \,|\, s, a) h(s') \right\} = h(s) + g(s)$$

for all $s \in \mathcal{S}$, and we express these more concisely as

$$\max_\pi \{ \mathcal{P}^\pi g \} = g, \qquad \max_\pi \{ r^\pi + \mathcal{P}^\pi h \} = h + g.$$

We say $(g^\star, h^\star)$ is a solution of the modified Bellman equations if $(g^\star, h^\star)$ satisfies the two equations and there exists a policy $\pi^\star$ attaining maximum simultaneously. It is known that solutions of modified Bellman equations always exist (Puterman, 2014, Proposition 9.1.1). Furthermore, $g^\star$ is unique and it is equal to the optimal average reward (Puterman, 2014, Theorem 9.1.2, 9.1.6). Finally, a policy $\pi^\star$ simultaneously attaining the maximum in the modified Bellman equations is an optimal policy (Puterman, 2014, Theorem 9.1.7, 9.1.8).

If the MDP is weakly communicating or unichain, $g^\star \in \mathbb{R}^d$ is a uniform constant vector, i.e., $g^\star = c\mathbf{1}$ for some $c \in \mathbb{R}$, where $\mathbf{1} \in \mathbb{R}^n$ is the vector with entries all 1 (Puterman, 2014, Theorem 8.3.2, 8.4.1). Then, first modified Bellman equations holds automatically, and modified Bellman equations reduce to

$$\max_\pi \{ r^\pi + \mathcal{P}^\pi h \} = h + g.$$

## 1.2   PRIOR WORKS

**Average-reward MDP.**   The setup of average-reward MDP was first introduced by Howard (1960) in the dynamic programming literature. Blackwell (1962) provided a theoretical framework for analyzing average-reward MDP. Yushkevich (1974); Denardo & Fox (1968) studied modified Bellman equations of multichain MDPs and solutions were characterized by Schweitzer & Federgruen (1978); Schweitzer (1984). In reinforcement learning (RL), average-reward MDP was mainly considered in the sample-based setup to find optimal policy when the transition matrix and reward are unknown (Dewanto et al., 2020). For this setup, Burnetas & Katehakis (1997); Jaksch et al. (2010) analyze regret minimization problem for unichain and communicating MDPs. Also, model-based algorithms (Zhang & Ji, 2019), model-free algorithms (Wei et al., 2020; Wan et al., 2021), policy gradient method (Kakade, 2001), and finite time analysis (Zhang et al., 2021) have been studied.

**Convergence of Value Iteration.**   Value iteration (VI) was first introduced in the DP literature (Bellman, 1957) and serves as a fundamental dynamic programming algorithm for computing the value functions. Its approximate and sample-based variants, such as Temporal Different Learning (Sutton, 1988), Fitted Value Iteration (Ernst et al., 2005; Munos & Szepesvári, 2008), Deep Q-Network (Mnih et al., 2015), are the workhorses of modern RL algorithms (Bertsekas & Tsitsiklis, 1996; Sutton & Barto, 2018a; Szepesvári, 2010). VI is also routinely applied in diverse settings, including factored MDPs (Rosenberg & Mansour, 2021), robust MDPs (Kumar et al., 2024), MDPs with reward machines (Bourel et al., 2023), and MDPs with options (Fruit et al., 2017).

The convergence of VI in average-reward MDPs has been extensively studied. For unichain MDPs, delta coefficient and ergodicity coefficient have been considered as the linear rate of VI (Seneta, 2006; Hübner, 1977), (Puterman, 2014, Theorem 6.6.1), and the J-stage span contraction demonstrates

linear rate of VI for every $J$-th iterations in terms of span seminorm (Federgruen et al., 1978; Van Der Wal, 1981), (Puterman, 2014, Theorem 8.5.2). Bertsekas (1998) proposes $\lambda$-SSP, which exhibits non-asymptotic linear convergence under the recurrent assumption. When MDP is multichain, it is known that normalized iterates converge to the optimal average reward (Puterman, 2014, Theorem 9.4.1) while policy error might not converge to zero. Schweitzer & Federgruen (1977; 1979) established necessary and sufficient conditions of convergence of VI and established asymptotic linear convergence on Bellman error.

For convergence of iterates to the $h^\star$ solution of modified Bellman equations, White (1963) introduced Relative Value Iteration (RVI) which subtracts a uniform constant for every iteration. Morton & Wecker (1977) studied sufficient conditions of convergence of RVI. Bravo & Cominetti (2024) studied asymptotic convergence rates of Rx-RVI on Bellman error in Q-learning setup, and Bravo & Contreras (2024) also considered Q-learning version of Halpern iteration in average-reward MDP and study sample complexity.

In Section A of the appendix, we present several tables that thoroughly compare the our new results with the prior results of the literature, and refer to Della Vecchia et al. (2012) for further detailed conditions of convergence of VI.

**Fixed point iterations.** The Banach fixed-point theorem (Banach, 1922) establishes the convergence of the standard fixed-point iteration with a contractive operator. As a generalization of Picard iteration, Kransnosel'skiĭ-Mann iteration (KM) (Mann, 1953; Krasnosel'skiĭ, 1955) was introduced, and its convergence with general nonexpansive operators was shown by Martinet (1970). The Halpern iteration (Halpern, 1967) converges for nonexpansive operators on Hilbert spaces (Wittmann, 1992) and uniformly smooth Banach spaces (Reich, 1980; Xu, 2002).

When a nonexpansive operator $T$ is assumed to have a fixed point, the fixed-point residual $\|Tx_k - x_k\|$ is a commonly used error measure for fixed-point problems. In Hilbert spaces, the KM iteration with nonexpansive operators was shown to exhibit $\mathcal{O}(1/\sqrt{k})$-rate by Matsushita (2017). Sabach & Shtern (2017) first established an $\mathcal{O}(1/k)$-rate for the Halpern iteration, and the constant was later improved by Lieder (2021); Kim (2021). In general normed spaces, KM iteration with nonexpansive opeator was proven to exhibit $\mathcal{O}(1/\sqrt{k})$-rate (Baillon & Bruck, 1992; Cominetti et al., 2014; Bravo & Cominetti, 2018). The Halpern iteration was shown to exhibit $\mathcal{O}(1/k)$-rate for (nonlinear) nonexpansive operators (Leustean, 2007; Sabach & Shtern, 2017; Contreras & Cominetti, 2022).

**Inconsistent fixed-point iteration.** A fixed-point iteration for a nonexpansive operator $T$ *without* a fixed point is referred to as the *inconsistent* setup, and it is the analog relevant to the average-reward MDP setup. There exist a line of researches about convergence of inconsistent fixed-point iteration in both Hilbert space (Pazy, 1971; Applegate et al., 2024; Bauschke et al., 2014; Liu et al., 2019) and Banach space (Browder & Petryshyn, 1966; Reich, 1973; Baillon, 1978; Reich & Shafrir, 1987). Notably, Park & Ryu (2023) studied sublinear convergence rates of KM iteration and Halpern iteration of the inconsistent setup in Hilbert spaces and established optimality by providing complexity lower bound.

**Complexity lower bounds.** With the information-based complexity analysis (Nemirovski, 1992), complexity lower bound on first-order methods for convex minimization problem has been thoroughly studied (Nesterov, 2018; Drori, 2017; Drori & Taylor, 2022; Carmon et al., 2020; 2021; Drori & Shamir, 2020). If a complexity lower bound matches an algorithm's convergence rate, it establishes optimality of the algorithm (Nemirovski, 1992; Kim & Fessler, 2016; Salim et al., 2022; Taylor & Drori, 2023; Drori & Teboulle, 2016; Park & Ryu, 2022). In Hilbert spaces, Park & Ryu (2022) showed exact complexity lower bound on fixed-point residual for deterministic fixed-point iterations with contractive and nonexpansive operators. In fixed-point problems, Colao & Marino (2021) established $\Omega\big(1/k^{1-\sqrt{2/q}}\big)$ lower bound on distance to solution for Halpern iteration with a nonexpansive operator in $q$-uniformly smooth Banach spaces. In general normed space, Contreras & Cominetti (2022) provided $\Omega(1/k)$ lower bound on the fixed-point residual for the general Mann iteration with a nonexpansive linear operator, which includes Picard iteration, KM iteration, and Halpern iteration.

In discounted MDPs, Goyal & Grand-Clément (2022) provided a lower bound on the Bellman error and distance to optimal value function for fixed-point iterations satisfying span condition with $\gamma$-

contractive Bellman operators. Lee & Ryu (2023) improved upon the prior lower bound on Bellman error by a factor $1 - \gamma^{k+1}$, and further established $\Omega(1/k)$ bound in undiscounted MDP. However, none of these works consider the average-reward MDP setup. Zurek & Chen (2023) studied sample complexity of learning a near-optimal policy in an average-reward MDP under generative model.

## 1.3 CONTRIBUTION

We summarize the contributions of this work as follows.

**Non-asymptotic rates.** For multichain MDPs, we establish the first non-asymptotic convergence rates on Bellman error. Theorems 1 and 2 and Corollary 1 and 2 present the non-asymptotic sublinear rates on both Bellman and policy errors in multichain MDPs (see Tables A.1 and A.2 of the Appendix). For the Relative Value Iteration (RVI) and its variants as described in Section 6, Theorems 5 and 6 establish the non-asymptotic sublinear rates on both Bellman and policy errors and point convergence in weakly communicating MDPs (see Tables A.5 and A.6 of the Appendix).

**Complexity lower bound.** Theorems 3 and 4 present the first complexity lower bounds for the average-reward MDP setup, one with a multichain MDP and another with a unichain MDP. These complexity lower bounds apply both to the Bellman error and normalized iterates for value-iteration-type methods satisfying the span condition.

**Characterization of optimal complexity.** Through our matching the convergence rates (upper bound) and the complexity lower bounds, we first establish the optimal complexity of standard VI in terms the normalized iterates and of Anc-VI in terms of the Bellman error.

## 2 PERFORMANCE MEASURES

We quickly review the standard performance measures used to quantify convergence rates of value-iteration-type methods for average-reward MDPs. Let $T$ be the Bellman optimality operator of the given MDP, and suppose a method generates sequences $\{V^k\}_{k=0,1,\dots}$ and $\{\pi^k\}_{k=0,1,\dots}$. We call $\frac{V^k - V^0}{\alpha_k}$ with an appropriate scaling factor $\alpha_k > 0$ for $k = 0, 1, \dots$ the *normalized iterates*. We call $TV^k - V^k$ the *Bellman error* at $V^k$ for $k = 0, 1, \dots$. We call $g^\star - g^{\pi_k}$ the *policy error* at $\pi_k$ for $k = 0, 1, \dots$. Again, we call the $V^k = TV^{k-1}$ for $k = 1, 2, \dots$ standard Value Iteration (VI) with greedy policy $\pi_k$ satisfying $T^{\pi_k} V^k = TV^k$.

**Fact 1** (Classical result, (Puterman, 2014, Theorem 9.4.1)). *Consider a general (multichain) MDP. Then, for $k \geq 1$, the normalized iterates of standard VI with $\alpha_k = k$ exhibit the rate*

$$\left\| \frac{V^k - V^0}{k} - g^\star \right\|_\infty \leq \frac{2}{k} \left\| V^0 - h^\star \right\|_\infty .$$

Fact 1 shows that the normalized iterates converge to optimal average reward in multichain MDPs with a non-asymptotic rate. As we will later show with Theorem 4, the $\mathcal{O}(1/k)$-rate on the normalized iterates of Fact 1 is exactly optimal. However, it is known that the convergence of normalized iterates *does not* guarantee convergence of policy error (Della Vecchia et al., 2012, Example 4).

**Fact 2** (Classical result, (Puterman, 2014, Theorem 9.1.7, 8.5.5)). *Consider a general (multichain) MDP. If $\|TV - V - g^\star\|_\infty = 0$, $\|g^\star - g^{\pi_V}\|_\infty = 0$, where $\pi_V$ is greedy policy satisfying $T^{\pi_V} V = TV$. Furthermore, if MDP is weakly communicating, $\|g^\star - g^{\pi_V}\|_\infty \leq \|TV - V - g^\star\|_\infty$.*

**Fact 3** (Classical result, (Puterman, 2014, Theorem 9.4.5)). *Consider a general (multichain) MDP. Assume that the transition matrices corresponding to every average-optimal deterministic policy are aperiodic. Then, for standard VI, the Bellman error $\|TV^k - V^k - g^\star\|_\infty$ converges to zero. Furthermore, $\|g^{\pi_k} - g^\star\|_\infty$ also converges to zero.*

Fact 2 shows that, unlike normalized iterate, convergence of Bellman error guarantees convergence of policy error. But the classical asymptotic convergence results on the Bellman error of Fact 3 has no quantitative rate (and also additionally requires aperiodicity), so we establish several stronger non-asymptotic rates throughout this paper.

We also briefly mention another performance measure considered in average-reward MDPs. The *span seminorm* is defined as $\|x\|_{sp} = \max_i x_i - \min_i x_i$ for $x \in \mathbb{R}^n$. The span seminorm of the Bellman error $\|TV^k - V^k\|_{sp}$ has been considered for weakly communicating and unichain MDPs because in such setups, the optimal average reward $g^\star$ is a uniform constant and $\|TV^k - V^k - g^\star\|_{sp} = \|TV^k - V^k\|_{sp}$. Therefore, unlike the $\|\cdot\|_\infty$-norm of the Bellman error, the span seminorm is computable without knowledge of $g^\star$. In this work, we primarily focus on convergence rates of the normalized iterates and $\|\cdot\|_\infty$-norm of the Bellman errors. Nevertheless, we point out that our results on the latter measure imply rates on the span seminorm of the Bellman error in weakly communicating and unichain MDP, since $\|TV - V\|_{sp} \leq 2\|TV - V - g^\star\|_\infty$ (Puterman, 2014, Section 6.6.1) in such setups.

## 3    RELAXED VALUE ITERATION

The *Relaxed Value Iteration* (Rx-VI) is

$$V^k = \lambda_k V^{k-1} + (1 - \lambda_k)TV^{k-1} \tag{Rx-VI}$$

for $k = 1, 2, \ldots$, where $T$ is the Bellman optimality operator, $V^0 \in \mathbb{R}^n$ is a starting point, and $0 \leq \lambda_k < 1$ for $k = 0, 1, \ldots$. $\pi_k$ is a greedy policy satisfying $T^{\pi_k}V^k = TV^k$ for $k = 0, 1, \ldots$. Notably, Rx-VI obtains the next iterate as a convex combination between the output of $T$ and the current point $V^{k-1}$.

We now present our non-asymptotic sublinear converge rates of Rx-VI in terms of the Bellman and policy errors while deferring the proofs to Section F of the appendix.

**Theorem 1.** *Consider a general (multichain) MDP. Let $(g^\star, h^\star)$ be a solution of the modified Bellman equations. For $k > K$, the Bellman and policy errors of Rx-VI with $\lambda_k = 1/2$ exhibits the rate*

$$\|g^\star - g^{\pi_k}\|_\infty \leq \|TV^k - V^k - g^\star\|_\infty \leq \frac{4\|V^0 - h^\star\|_\infty}{\sqrt{(3.141592\ldots)(k - K)}},$$

*where $K = \left(2\|r\|_\infty + 4\|V^0\|_\infty + 16\|V^0 - h^\star\|_\infty + 2\|g^\star\|_\infty\right)/\epsilon$,*

$$0 < \epsilon = \inf_{\pi \in S \setminus \{\pi \,|\, \mathcal{P}^\pi g^\star = g^\star\}} \|\mathcal{P}^\pi g^\star - g^\star\|_\infty,$$

*and $S$ is the set of all deterministic policies. Since $\pi$ denotes the policy in this work, we write $(3.141592\ldots)$ to denote the mathematical constant usually written as $\pi$.*

We clarify that for general (multichain) MDPs, the Bellman error does not bound the policy error. However, our analysis shows that the Bellman error does bound the policy error for $k > K$.

The characterization of $K$ in Theorem 1 is somewhat intricate when considering general MDPs. This is simplified if we focus on specific class of MDPs which includes weakly communicating and unichain MDPs.

**Corollary 1.** *Consider a a general (multichain) MDP satsifying $\mathcal{P}^\pi g^\star = g^\star$ for any policy $\pi$. Let $(g^\star, h^\star)$ be a solution of the modified Bellman equations. For $k \geq 1$, the Bellman and policy errors of Rx-VI with $\lambda_k = 1/2$ exhibit the rate*

$$\|g^\star - g^{\pi_k}\|_\infty \leq \|TV^k - V^k - g^\star\|_\infty \leq \frac{4\|V^0 - h^\star\|_\infty}{\sqrt{(3.141592\ldots)k}}.$$

*Proof of Corollary 1.* We apply Theorem 1. By assumption on MDP, $S/\{\pi \,|\, \mathcal{P}^\pi g^\star = g^\star\} = \emptyset$. So $\epsilon = \inf_{\pi \in \emptyset} \|\mathcal{P}^\pi g^\star - g^\star\|_\infty = \infty$ and $K = 0$. Finally, we plug $K = 0$ into Theorem 1.   $\square$

Note that the weakly communicating MDPs satisfy the assumption of Corollary 1 since $g^\star = c\mathbf{1}$ for some $c \in \mathbb{R}$ (Puterman, 2014, Theorem 8.3.2) and so $\mathcal{P}^\pi c\mathbf{1} = c\mathbf{1}$ for any policy $\pi$. In the next section, we will show that the $\mathcal{O}(1/\sqrt{k})$-rate with Rx-VI can be improved to $\mathcal{O}(1/k)$-rate with Anc-VI. Section B of Appendix presents more general results establishing convergence rates for arbitrary $\lambda_k$ in terms of both the Bellman error and the normalized iterates.

Broadly speaking, Rx-VI is a well-studied algorithm. This averaging mechanism has been widely studied in fixed-point theory literature under the name *Krasnosel'skiĭ–Mann iteration* (Mann, 1953; Krasnosel'skiĭ, 1955; Bauschke & Combettes, 2017; Baillon & Bruck, 1992; Cominetti et al., 2014). In the dynamic programming literature, the *aperiodic transformation* (Puterman, 2014, Section 8.5.4), which averages the transition matrix and identity to make the transition matrix aperiodic, is closely related to this averaging mechanism. In the reinforcement learning literature, TD learning and Q learning use the averaging mechanism to stabilize randomness and ensure convergence (Sutton, 1988; Watkins, 1989; Bertsekas & Tsitsiklis, 1995; Bravo & Cominetti, 2024), and in tabular setup, Kushner & Kleinman (1971); Porteus & Totten (1978); Goyal & Grand-Clément (2022); Akian et al. (2022) studied convergence of Rx-VI in discounted MDP setup. However, to the best of our knowledge, no prior work has established non-asymptotic rates of Rx-VI or any other value-iteration-type method for multichain MDPs. Only Schweitzer & Federgruen (1977; 1979) established asymptotic convergence results for multichain MDPs.

## 4 ANCHORED VALUE ITERATION

The *Anchored Value Iteration* is

$$V^k = \lambda_k V^0 + (1 - \lambda_k) T V^{k-1} \qquad \text{(Anc-VI)}$$

for $k = 1, 2, \ldots$, where $T$ is the Bellman optimality operator, $V^0 \in \mathbb{R}^n$ is a starting point, and $0 \leq \lambda_k < 1$ for $k = 0, 1, \ldots$. $\pi_k$ is a greedy policy satisfying $T^{\pi_k} V^k = T V^k$ for $k = 0, 1, \ldots$. Notably, Anc-VI obtains the next iterate as a convex combination between the output of $T$ and the *starting point* $V^0$ (note, Rx-VI uses $V^{k-1}$ instead of $V^0$). We call the $\lambda_k V_0$ term the *anchor term* since, loosely speaking, it serves to retract the iterates back toward the starting point $V_0$. Generally, $\lambda_k$ is set to be a decreasing sequence, and then the strength of the anchor mechanism diminishes as the iteration progresses.

We now present our non-asymptotic sublinear converge rates of Anc-VI in terms of the Bellman and policy errors while deferring the proofs to Section G of the Appendix.

**Theorem 2.** *Consider a general (multichain) MDP. Let $(g^\star, h^\star)$ be a solution of the modified Bellman equations. For $k > K$, the Bellman and policy errors of Anc-VI with $\lambda_k = \frac{2}{k+2}$. exhibits the rate*

$$\left\| g^\star - g^{\pi_k} \right\|_\infty \leq \left\| T V^k - V^k - g^\star \right\|_\infty \leq \frac{8}{k+1} \left\| V^0 - h^\star \right\|_\infty + \frac{K}{k+1} \left\| g^\star \right\|_\infty,$$

*where $K = \left( 3 \left\| r \right\|_\infty + 12 \left\| V^0 - h^\star \right\|_\infty + 3 \left\| g^\star \right\|_\infty \right) / \epsilon$,*

$$0 < \epsilon = \inf_{\pi \in S \setminus \{\pi \,|\, \mathcal{P}^\pi g^\star = g^\star\}} \left\| \mathcal{P}^\pi g^\star - g^\star \right\|_\infty,$$

*and $S$ is the set of all deterministic policies.*

As before, Theorem 2 claims that the Bellman error bounds the policy error for $k > K$, and the characterization of $K$ in Theorem 2 is simplified if we focus on a specific class of MDPs which includes weakly communicating and unichain MDPs.

**Corollary 2.** *Consider a general (multichain) MDP satsifying $\mathcal{P}^\pi g^\star = g^\star$ for any policy $\pi$. Let $(g^\star, h^\star)$ be a solution of the modified Bellman equations. For $k \geq 1$, the Bellman and policy errors of Anc-VI with $\lambda_k = \frac{2}{k+2}$ exhibits the rate*

$$\left\| g^\star - g^{\pi_k} \right\|_\infty \leq \left\| T V^k - V^k - g^\star \right\|_\infty \leq \frac{8}{k+1} \left\| V^0 - h^\star \right\|_\infty.$$

*Proof of Corollary 2.* Follows from the same line of argument as for Corollary 1. □

Note, the anchoring mechanism allows us to improve the rate to $\mathcal{O}(1/k)$. In the next section, we will show that the $\mathcal{O}(1/k)$ rate is optimal in the weakly communicating setup by providing a matching complexity lower bound. Section C of Appendix presents more general results establishing convergence rates for arbitrary $\lambda_k$ in terms of both the Bellman error and the normalized iterates.

The anchor mechanism has been widely studied in minimax optimization and fixed-point problems (Halpern, 1967; Sabach & Shtern, 2017; Lieder, 2021; Park & Ryu, 2022; Contreras & Cominetti,

2022; Yoon & Ryu, 2021). In the context of reinforcement learning, Lee & Ryu (2023) applied the anchoring mechanism to VI to achieve an accelerated convergence rate for cumulative-reward MDPs, and Bravo & Contreras (2024) applied the anchoring mechanism to Q-learning for average-reward MDPs. However, to the best of our knowledge, no prior work established a non-asymptotic rate for value-iteration-type methods for multichain MDP.

We further clarify that our non-asymptotic convergence results in Section 3 and 4 are neither a direct application nor a direct adaptation of the prior convergence. VI for the average-reward MDP setup can be thought of as a fixed point iteration *without* a fixed point, and so most prior analyses assuming the existence of a fixed point do not apply. Bravo & Contreras (2024); Bravo & Cominetti (2024) study the convergence of Rx-RVI and Anc-RVI in unichain MDPs by applying results derived from the fixed-point iteration setup, but their analyses do not extend to mulichain MDPs. In the inconsistent fixed point iteration setup, analog relevant to the average-reward MDPs setup, prior analyses for Hilbert space (Pazy, 1971; Applegate et al., 2024; Bauschke et al., 2014; Liu et al., 2019; Park & Ryu, 2023) are not applicable to Bellman operators since $\mathbb{R}^d$ with $\|\cdot\|_\infty$-norm is not Hilbert space. The prior analyses for Banach space assuming uniformly Gateaux differentiable norm (Browder & Petryshyn, 1966; Reich, 1973; Reich & Shafrir, 1987) or uniform convexity (Browder & Petryshyn, 1966) are not applicable either since $\|\cdot\|_\infty$-norm is not uniformly Gateaux differentiable norm and $\mathbb{R}^d$ with $\|\cdot\|_\infty$-norm is not uniformly convex space. We note that our analyses specifically utilize the structure of Bellman operators and modified Bellman equation to obtain a non-asymptotic convergence rate on both Bellman and policy errors, adapting proof techiniques from Cominetti et al. (2014); Contreras & Cominetti (2022) to the multichain setup.

## 5 COMPLEXITY LOWER BOUND

We now present complexity lower bounds establishing optimality of Anc-VI in terms of the Bellman error and standard VI in terms of the normalized iterates. To the best of our knowledge, Theorems 3 and 4 are the first complexity lower bounds for value-iteration-type methods in the average-reward MDP setup.

Following the information-based complexity framework (Nemirovski, 1992), we consider the *span condition*

$$V^{k+1} \in V^0 + \text{span}\{TV^0 - V^0, TV^1 - V^1, \ldots, TV^k - V^k\}, \tag{1}$$

where $T$ is the Bellman optimality operator and span$(A)$ is set of all finite linear combinations of the elements of $A$. Standard VI, Rx-VI, and Anc-VI all satisfy equation 1.

**Optimality of Anc-VI for Bellman error.** We now establish the optimality of Anc-VI for weakly communicating and unichain MDPs in terms of the Bellman error.

**Theorem 3.** *Let $k \geq 0$, $n \geq k + 2$, and $V^0 \in \mathbb{R}^n$. Then there exists a unichain MDP with $|\mathcal{S}| = n$ and $|\mathcal{A}| = 1$ such that its modified Bellman equations has a solution $(g^\star, h^\star)$ satisfying*

$$\left\| \sum_{i=0}^{k} a_i(TV^i - V^i) - g^\star \right\|_\infty \geq \frac{1}{k+1} \left\| V^0 - h^\star \right\|_\infty$$

*for any iterates $\{V^i\}_{i=0}^k$ satisfying the span condition equation 1 and any choice of real numbers $\{a_i\}_{i=0}^k$ such that $\sum_{i=0}^k a_i = 1$.*

If we set $a_k = 1$ in Theorem 3, we get $\left\| TV^k - V^k - g^\star \right\|_\infty \geq \frac{1}{k+1} \left\| V^0 - h^\star \right\|_\infty$. Note that the construction of Theorem 3 is a unichain MDP, which is also a weakly communicating MDP. The lower bound matches the $\frac{8}{k+1} \left\| V^0 - h^\star \right\|_\infty$ upper bound of Corollary 5, which applies to both weakly communicating and unichain MDPs. The upper and lower bounds match up to constant of factor 8, and we therefore conclude optimality for both weakly communicating and unichain MDPs.

**Exact optimality of standard VI for normalized iterates.** We now establish the optimality of standard VI for general (multichain) MDPs in terms of the normalized iterates.

**Theorem 4.** *Let $k \geq 0$, $n \geq k + 3$, and $V^0 \in \mathbb{R}^n$. Then there exists a multichain MDP with $|\mathcal{S}| = n$ and $|\mathcal{A}| = 1$ such that its modified Bellman equations has a solution $(g^\star, h^\star)$ satisfying*

$$\left\| \sum_{i=0}^{k} a_i (TV^i - V^i) - g^\star \right\|_\infty \geq \frac{2}{k+1} \left\| V^0 - h^\star \right\|_\infty$$

*for any iterates $\{V^i\}_{i=0}^k$ satisfying the span condition equation 1 and any choice of real numbers $\{a_i\}_{i=0}^k$ such that $\sum_{i=0}^k a_i = 1$.*

If we set $a_i = \frac{1}{k+1}$ for all $i = 0, \ldots, k$ in Theorem 4, we get $\left\| \frac{V^{k+1} - V^0}{k+1} - g^\star \right\|_\infty \geq \frac{2}{k+1} \left\| V^0 - h^\star \right\|_\infty$. This lower bound exactly matches the $\frac{2}{k+1} \left\| V^0 - h^\star \right\|_\infty$ upper bound of Fact 1, and we therefore conclude exact optimality of standard VI in terms of the normalized iterates.

**Discussion.** To clarify, the unichain MDP construction of Theorem 3 is a multichain MDP, so Theorem 3 and Fact 1 together already establish optimality up to a constant factor of 2. However, the multichain construction of Theorem 3 improves the lower bound by a constant factor of 2, and this factor of 2 leads to the exact match.

The span condition used in Theorems 3 and 4 are arguably very natural and is satisfied by Standard VI, Rx-VI, and Anc-VI. The span condition is commonly used in the construction of complexity lower bounds for first-order optimization methods (Nesterov, 2018; Drori, 2017; Drori & Taylor, 2022; Carmon et al., 2020; 2021; Park & Ryu, 2022) and has been used in the lower bound for standard VI and Anc-VI (Goyal & Grand-Clément, 2022; Lee & Ryu, 2023). However, designing an algorithm that breaks the lower bound of Theorem 3 and 4 by violating the span condition remains a possibility. In optimization theory, there is precedence of lower bounds being broken by violating seemingly natural and minute conditions (Hannah et al., 2018; Golowich et al., 2020; Yoon & Ryu, 2021).

## 6 RELAXED AND ANCHORED RELATIVE VALUE ITERATION

The iterates of standard VI, Rx-VI, and Anc-VI *diverge*. For example, the iterates of standard VI asymptotically behave as $V^k \sim kg^\star$ as $k \to \infty$ by Fact 1. Of course, the normalized iterates do converge, but if we want the iterates themselves to converge, the algorithm must be modified.

The *Relative Value Iteration* (RVI) subtracts some uniform constant vector at each iteration:

$$h^k = Th^{k-1} - f(h^{k-1})\mathbf{1}$$

for $k = 1, 2, \ldots$, where $T$ is the bellman optimality operator, $h^0 \in \mathbb{R}^n$ is a starting point, $\mathbf{1} \in \mathbb{R}^n$ is the uniform constant vector with all entries 1, and $f \colon \mathbb{R}^n \to \mathbb{R}$ is a continuous function satisfying $f(x + c\mathbf{1}) = f(x) + c$ for any $c \in \mathbb{R}$. Following is one of known convergence results of RVI.

**Fact 4** (Classical result, (Bertsekas, 2012, Theorem 4.3.2)). *Consider a unichain MDP. Assume that the transition matrices corresponding to every average-optimal deterministic policy are aperiodic, and $f(h) = (Th)_i$ for some fixed $1 \leq i \leq n$. Then, for some solution of modified Bellman equations $(g^\star, h^\star)$, the iterates of standard RVI converge to $h^\star$ and $(Th^k)_i \mathbf{1}$ converges to $g^\star$.*

Like standard VI, iterates of Rx-VI and Anc-VI also diverge as we show in the Theorems 7 and 9 of the Appendix. To ensure convergence of the iterates, we can also subtract uniform constant vectors from the iterate. The *Relaxed Relative Value Iteration* is

$$h^k = \lambda_{k-1} h^{k-1} + (1 - \lambda_{k-1})(Th^{k-1} - f(h^{k-1})\mathbf{1}) \tag{Rx-RVI}$$

for $k = 1, 2, \ldots$, where $0 \leq \lambda_k < 1$ and $h^0$ is starting point. $\pi_k$ is a greedy policy satisfying $T^{\pi_k} h^k = Th^k$ for $k = 0, 1, \ldots$. The *Anchored Relative Value Iteration* is

$$h^k = \lambda_{k-1} h^0 + (1 - \lambda_{k-1})(Th^{k-1} - f(h^{k-1})\mathbf{1}) \tag{Anc-RVI}$$

for $k = 1, 2, \ldots$, where $0 \leq \lambda_k < 1$ and $h^0$ is starting point. $\pi_k$ is a greedy policy satisfying $T^{\pi_k} h^k = Th^k$ for $k = 0, 1, \ldots$.

Now we present our non-asymptotic convergence rates of Rx-RVI and Anc-RVI in terms of Bellman and policy errors while deferring the proofs to Section I in Appendix.

**Theorem 5.** *Consider a weakly communicating MDP. Let $(g^\star, h^\star)$ be a solution of modified Bellman equations. For $k \geq 1$ and , the Bellman and policy errors of Rx-RVI with $\lambda_k = 1/2$ exhibits the rate*

$$\|g^\star - g^{\pi_k}\|_\infty \leq \|Th^k - h^k - g^\star\|_\infty \leq \frac{4}{\sqrt{(3.141592\ldots)k}} \|h^0 - h^\star\|_\infty.$$

*Furthermore, $h^k \to h^\infty$ and $f(h^k)\mathbf{1} \to g^\star$ for some solution of modified Bellman equations $(g^\star, h^\infty)$.*

**Theorem 6.** *Consider a weakly communicating MDP. Let $(g^\star, h^\star)$ be a solution of modified Bellman equations. For $k \geq 1$, the Bellman and policy errors of Anc-RVI with $\lambda_k = \frac{2}{k+2}$ exhibits the rate*

$$\|g^\star - g^{\pi_k}\|_\infty \leq \|Th^k - h^k - g^\star\|_\infty \leq \frac{8}{k+1} \|h^0 - h^\star\|_\infty.$$

*Furthermore, if MDP is unichain, $h^k \to h^\infty$ and $f(h^k)\mathbf{1} \to g^\star$ for some solution of modified Bellman equations $(g^\star, h^\infty)$.*

Since Rx-RVI and Anc-RVI generate same policy as Rx-VI and Anc-VI, respectively, the rates of Bellman errors of Rx-RVI and Anc-RVI in Theorem 5 and 6 are immediately implied by the rates of Rx-VI and Anc-VI in Corollary 1 and 2, respectively. Therefore, the main substance of Corollary 1 and 2 are the convergence results $(h^k, f(h^k)\mathbf{1}) \to (h^\infty, g^\star)$. Section D of Appendix presents more general results establishing convergence rates for arbitrary $\lambda_k$ in terms of the Bellman error and convergence of iterates. Lastly, we briefly note that for weakly communicating MDP, non-asymptotic rate on Bellman error can be obtained from results in Bravo & Contreras (2024); Bravo & Cominetti (2024) by leveraging their convergence analysis with uniform constant $g^\star$ in unichain MDP.

## 7 CONCLUSION

In this work, we present the first non-asymptotic convergence rates for multichain MDPs in terms of the Bellman error. We also provide complexity lower bounds matching the upper bound of Anc-VI in terms of the Bellman error up to a constant factor of $8$ for weakly communicating and unichain MDPs. Finally, we also showed that standard VI is exactly optimal in terms of the normalized iterates for multichain MDPs. Our results and proof techniques open the door to future work on non-asymptotic, sublinear, and optimal rates for average-reward MDPs.

One future direction is to fully characterize the optimal non-asymptotic complexity on Bellman error for multichain MDPs, as our current upper bound of Theorem 2, with its dependence on $K$, does not exactly match the lower bound of Theorem 4. We aim to achieve this goal by enhancing our lower bound through the consideration of more delicate worst-case multichain MDPs.

Finally, we highlight an observation implied by our results: the "correct" rates for (undiscounted) average-reward MDPs are sublinear, i.e., something like $\mathcal{O}(1/k)$. This contrasts with the classical $\gamma$-discounted cumulative-reward MDP setup, where we are accustomed to $\mathcal{O}(\gamma^k)$-rates. We expect future work analyzing other average-reward MDP setups and algorithms to similarly discover optimal sublinear rates.

## 8 ACKNOWLEDGMENTS

This work is supported by the National Research Foundation of Korea (NRF) grant funded by the Korea government (No.RS-2024-00421203). We thank Taeho Yoon for providing valuable feedback.

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

# A    COMPARISON WITH PRIOR CONVERGENCE RESULTS IN TERMS OF PERFORMANCE MEASURES

In this section, we present asymptotic and non-asymptotic convergence results of prior works and our work in terms of Bellman error, policy error, span seminorm, and normalized iterates. We denote the correspondence between numbers and prior works as follows. [1: Federgruen et al. (1978), 2: Van Der Wal (1981), 3: Schweitzer & Federgruen (1977), 4: Schweitzer & Federgruen (1979), 5: Bertsekas (1998), 6: Puterman (2014), 7: Bravo & Cominetti (2024), 8: Bravo & Contreras (2024), 9: Bertsekas (2012)]

## A.1    BELLMAN ERROR

| Prior works | Non-asym multi MDP | Asym multi MDP | Non-asym w.c. MDP | Asym w.c. MDP | Non-asym uni MDP | Asym uni MDP |
|---|---|---|---|---|---|---|
| [1, 2] | ✗ | ✗ | ✗ | ✗ | ✓ | ✓ |
| [3, 4] | ✗ | ✓ | ✗ | ✓ | ✓ | ✓ |
| [5] | ✗ | ✗ | ✗ | ✗ | ✓ | ✓ |
| Rx-VI | ✓ | ✓ | ✓ | ✓ | ✓ | ✓ |
| Anc-VI | ✓ | ✓ | ✓ | ✓ | ✓ | ✓ |

## A.2    POLICY ERROR

| Prior works | Non-asym multi MDP | Asym multi MDP | Non-asym w.c. MDP | Asym w.c. MDP | Non-asym uni MDP | Asym uni MDP |
|---|---|---|---|---|---|---|
| [1, 2] | ✗ | ✗ | ✗ | ✗ | ✓ | ✓ |
| [3, 4] | ✗ | ✓ | ✗ | ✓ | ✓ | ✓ |
| [5] | ✗ | ✗ | ✗ | ✗ | ✓ | ✓ |
| Rx-VI | ✓ | ✓ | ✓ | ✓ | ✓ | ✓ |
| Anc-VI | ✓ | ✓ | ✓ | ✓ | ✓ | ✓ |

## A.3    SPAN SEMINORM

| Prior works | Non-asym multi MDP | Asym multi MDP | Non-asym w.c. MDP | Asym w.c. MDP | Non-asym uni MDP | Asym uni MDP |
|---|---|---|---|---|---|---|
| [1, 2] | N/A | N/A | ✗ | ✗ | ✓ | ✓ |
| [3, 4] | N/A | N/A | ✗ | ✓ | ✓ | ✓ |
| [5] | N/A | N/A | ✗ | ✗ | ✓ | ✓ |
| Rx-VI | N/A | N/A | ✓ | ✓ | ✓ | ✓ |
| Anc-VI | N/A | N/A | ✓ | ✓ | ✓ | ✓ |

## A.4    NORMALIZED ITERATES

| Prior works | Non-asym multi MDP | Asym multi MDP | Non-asym w.c. MDP | Asym w.c. MDP | Non-asym uni MDP | Asym uni MDP |
|---|---|---|---|---|---|---|
| [6] | ✓ | ✓ | ✓ | ✓ | ✓ | ✓ |
| Rx-VI | ✓ | ✓ | ✓ | ✓ | ✓ | ✓ |
| Anc-VI | ✓ | ✓ | ✓ | ✓ | ✓ | ✓ |

## A.5    BELLMAN ERROR (RVI)

| Prior works | Non-asym multi MDP | Asym multi MDP | Non-asym w.c. MDP | Asym w.c. MDP | Non-asym uni MDP | Asym uni MDP |
|---|---|---|---|---|---|---|
| [7] | N/A | N/A | ✗ | ✗ | ✓ | ✓ |
| [8] | N/A | N/A | ✗ | ✗ | ✓ | ✓ |
| [9] | N/A | N/A | ✗ | ✗ | ✓ | ✓ |
| Rx-RVI | N/A | N/A | ✓ | ✓ | ✓ | ✓ |
| Anc-RVI | N/A | N/A | ✓ | ✓ | ✓ | ✓ |

## A.6 POLICY ERROR (RVI)

| Prior works | Non-asym multi MDP | Asym multi MDP | Non-asym w.c. MDP | Asym w.c. MDP | Non-asym uni MDP | Asym uni MDP |
|---|---|---|---|---|---|---|
| [7] | N/A | N/A | ✗ | ✗ | ✓ | ✓ |
| [8] | N/A | N/A | ✗ | ✗ | ✓ | ✓ |
| [9] | N/A | N/A | ✗ | ✗ | ✓ | ✓ |
| Rx-RVI | N/A | N/A | ✓ | ✓ | ✓ | ✓ |
| Anc-RVI | N/A | N/A | ✓ | ✓ | ✓ | ✓ |

## A.7 SPAN SEMINORM (RVI)

| Prior works | Non-asym multi MDP | Asym multi MDP | Non-asym w.c. MDP | Asym w.c. MDP | Non-asym uni MDP | Asym uni MDP |
|---|---|---|---|---|---|---|
| [7] | N/A | N/A | ✗ | ✗ | ✓ | ✓ |
| [8] | N/A | N/A | ✗ | ✗ | ✓ | ✓ |
| [9] | N/A | N/A | ✗ | ✗ | ✓ | ✓ |
| Rx-RVI | N/A | N/A | ✓ | ✓ | ✓ | ✓ |
| Anc-RVI | N/A | N/A | ✓ | ✓ | ✓ | ✓ |

# B CONVERGENCE RATES OF RX-VI WITH ARBITRARY $\lambda_k$

In this section, we present the convergence rates of Rx-VI for arbitrary $\lambda_k$ in terms of both the Bellman error and the normalized iterates.

**Theorem 7.** *Consider a general (multichain) MDP. Let $(g^\star, h^\star)$ be a solution of the modified Bellman equations. For $k > K$, the normalized iterates of Rx-VI with $\alpha_k = \sum_{i=1}^{k}(1 - \lambda_i)$ exhibits the rate*

$$\left\| \frac{V^k - V^0}{\sum_{i=1}^{k}(1 - \lambda_i)} - g^\star \right\|_\infty \leq \frac{2(1 - \Pi_{i=1}^{k}\lambda_i)}{\sum_{i=1}^{k}(1 - \lambda_i)} \left\| V^0 - h^\star \right\|_\infty.$$

**Theorem 8.** *Consider a general (multichain) MDP. Let $(g^\star, h^\star)$ be a solution of the modified Bellman equations. Let $0 < \lambda_j$ for $1 \leq j$ and $\limsup \lambda_j < 1$. Then, there exist $0 < K$ such that for $K < k$, the Bellman and policy errors of Rx-VI exhibit the rates*

$$\|g^\star - g^{\pi_k}\|_\infty \leq \|TV^k - V^k - g^\star\|_\infty \leq \frac{2\|V^0 - h^\star\|_\infty}{\sqrt{(3.141592\dots)\sum_{i=K+1}^{k}\lambda_i(1 - \lambda_i)}}$$

*Specifically, $K$ is the minimum iteration number satisfying if $K \leq k$, $\pi_k$ generated by Rx-VI satisfies $\mathcal{P}^{\pi_k} g^\star = g^\star$, first modified Bellman equation.*

We defer the proofs to Appendix F. Note that Theorems 7 and 8 imply the convergence of normalized iterate and Bellman error to $g^\star$ respectively when $\sum_{i=1}^{\infty}(1 - \lambda_i) = \infty$ and $\sum_{i=1}^{\infty} \lambda_i(1 - \lambda_i) = \infty$.

Interestingly, for normalized iterate, Theorem 1 recovers rate of standard VI in Fact 1.

**Corollary 3.** *Consider a general (multichain) MDP. Let $(g^\star, h^\star)$ be a solution of the modified Bellman equations. The normalized iterate of Rx-VI in Theorem 7 is optimized when $\lambda_k = 0$ with*

$$\left\| \frac{V^k - V^0}{k} - g^\star \right\|_\infty \leq \frac{2}{k} \left\| V^0 - h^\star \right\|_\infty.$$

*Proof.* By AM-GM inequality, we have $\Pi_{i=1}^{k}\lambda_i \leq (\Pi_{i=1}^{k}\lambda_i)^{1/k} \leq \frac{\sum_{i=1}^{k}\lambda_i}{k}$ since $\lambda_i \leq 1$. This implies $\frac{1}{k} \leq \frac{1 - \Pi_{i=1}^{k}\lambda_i}{\sum_{i=1}^{k}(1 - \lambda_i)}$ and if $\lambda_i = 0$ for all $i$, equality holds. Therefore, by plugging $\lambda_i = 0$ in Theorem 7, we get the desired result. □

Lastly, we present the non-asymptotic rate of Rx-VI with arbitrary $\lambda_k$ in specific class of MDPs which includes weakly communicating.

**Corollary 4.** *Consider a a general (multichain) MDP satsifying $\mathcal{P}^\pi g^\star = g^\star$ for any policy $\pi$. Let $(g^\star, h^\star)$ be a solution of the modified Bellman equations. Let $0 < \lambda_j < 1$ for $1 \le j$. Then, for $1 \le k$, the Bellman error of Rx-VI exhibit the rates*

$$\left\| g^\star - g^{\pi_k} \right\|_\infty \le \left\| TV^k - V^k - g^\star \right\|_\infty \le \frac{2\left\| V^0 - h^\star \right\|_\infty}{\sqrt{(3.141592\dots)\sum_{j=1}^k \lambda_i(1-\lambda_i)}}$$

*Proof.* We apply Theorem 8. By assumption on MDP, $S/\{\pi \,|\, \mathcal{P}^\pi g^\star = g^\star\} = \emptyset$. So $\epsilon = \inf_{\pi \in \emptyset} \left\| \mathcal{P}^\pi g^\star - g^\star \right\|_\infty = \infty$ and $K = 0$. Finally, we plug $K = 0$ into Theorem 8 □

## C    CONVERGENCE RATES OF ANC-VI WITH ARBITRARY $\lambda_k$

In this section, we present the convergence rates of Anc-VI for arbitrary $\lambda_k$ in terms of both the Bellman error and the normalized iterates.

**Theorem 9.** *Consider a general (multichain) MDP. Let $(g^\star, h^\star)$ be a solution of the modified Bellman equations. The normalized iterates of Anc-VI with $\alpha_k = \sum_{i=1}^k \Pi_{j=i}^k (1-\lambda_j)$ exhibits the rates*

$$\left\| \frac{V^k - V^0}{\sum_{i=1}^k \Pi_{j=i}^k (1-\lambda_j)} - g^\star \right\|_\infty \le \frac{2(1-\lambda_k)}{\sum_{i=1}^k \Pi_{j=i}^k (1-\lambda_j)} \left\| V^0 - h^\star \right\|_\infty.$$

**Theorem 10.** *Consider a general (multichain) MDP. Let $(g^\star, h^\star)$ be a solution of the modified Bellman equations. Let $\lambda_{k+1} \le \lambda_k < 1$ for $1 \le k$ and $\lim \lambda_k = 0$. Then, there exist $0 < K$ such that for $K < k$, the Bellman and policy errors of Anc-VI exhibit the rates*

$$\left\| g^\star - g^{\pi_k} \right\|_\infty \le \left\| TV^k - V^k - g^\star \right\|_\infty \le 2\left(1 - \sum_{i=1}^k \lambda_i \Pi_{j=i}^k (1-\lambda_j)\right) \left\| V^0 - h^\star \right\|_\infty$$
$$+ 2\Pi_{j=K}^k (1-\lambda_j) \left\| g^\star \right\|_\infty$$

*Specifically, $K$ is the minimum iteration number satisfying if $K \le k$, $\pi_k$ generated by Anc-VI satisfies $\mathcal{P}^{\pi_k} g^\star = g^\star$, first modified Bellman equation.*

We defer the proofs to Appendix G. Note that Theorems 3 and 4 imply the convergence of normalized iterate and Bellman error to $g$ respectively when $\lim_{k \to \infty} \sum_{i=0}^k \Pi_{j=i}^k (1-\lambda_j) = \infty$ and $\lim_{k \to \infty} \sum_{i=1}^k \lambda_i \Pi_{j=i}^k (1-\lambda_j) = 1$. We briefly mention that like Rx-VI, convergence rate of normalized iterate of Anc-VI is optimized when $\lambda_k = 0$ and recover rate of standard VI in Fact 1.

Lastly, we present the non-asymptotic rate of Anc-VI with arbitrary $\lambda_k$ in specific class of MDPs which includes weakly communicating.

**Corollary 5.** *Consider a a general (multichain) MDP satsifying $\mathcal{P}^\pi g^\star = g^\star$ for any policy $\pi$. Let $(g^\star, h^\star)$ be a solution of the modified Bellman equations. Let $\lambda_{k+1} \le \lambda_k < 1$ for $1 \le k$. Then, there exist $0 < K$ such that for $K < k$, the Bellman and policy errors of Anc-VI exhibit the rates*

$$\left\| g^\star - g^{\pi_k} \right\|_\infty \le \left\| TV^k - V^k - g^\star \right\|_\infty \le 2\left(\sum_{i=0}^k \Pi_{j=i+1}^k (1-\lambda_j)\lambda_i^2\right) \left\| V^0 - h^\star \right\|_\infty.$$

*Proof.* We apply Theorem 10. By assumption on MDP, $S/\{\pi \,|\, \mathcal{P}^\pi g^\star = g^\star\} = \emptyset$. So $\epsilon = \inf_{\pi \in \emptyset} \left\| \mathcal{P}^\pi g^\star - g^\star \right\|_\infty = \infty$ and $K = 0$. Finally, we plug $K = 0$ into Theorem 10 □

# D   CONVERGENCE RATES OF RX-RVI AND ANC-RVI WITH ARBITRARY $\lambda_k$

In this section, we present the convergence rates of Rx-RVI and Anc-RVI for arbitrary $\lambda_k$ in terms of both the Bellman error.

**Theorem 11.** *Consider a weakly communicating MDP. Let $(g^\star, h^\star)$ be a solution of the modified Bellman equations. Let $0 < \lambda_j < 1$ for $1 \leq j$. Then, for $1 \leq k$, Rx-RVI exhibit the rates*

$$\left\| g^\star - g^{\pi_k} \right\|_\infty \leq \left\| Th^k - h^k - g^\star \right\|_\infty \leq \frac{2 \left\| h^0 - h^\star \right\|_\infty}{\sqrt{(3.141592\dots)\sum_{j=1}^{k} \lambda_i(1-\lambda_i)}}$$

*and if $\limsup \lambda_j < 1$, $h^k$ converges to $h^\star$ and $f(h^k)\mathbf{1}$ converges to $g^\star$ for some solution of modified Bellman equations $(g^\star, h^\infty)$. .*

**Theorem 12.** *Consider a weakly communicating MDP. Let $(g^\star, h^\star)$ be a solution of the modified Bellman equations. Let $\lambda_{k+1} \leq \lambda_k < 1$ for $1 \leq k$. Then, Anc-RVI exhibit the rates*

$$\left\| g^\star - g^{\pi_k} \right\|_\infty \leq \left\| Th^k - h^k - g^\star \right\|_\infty \leq 2 \left( \sum_{i=0}^{k} \Pi_{j=i+1}^{k}(1-\lambda_j)\lambda_i^2 \right) \left\| h^0 - h^\star \right\|_\infty$$

*and if $\lim \lambda_k = 0$ and MDP is unichain, $h^k$ converges to $h^\star$ and $f(h^k)\mathbf{1}$ converges to $g^\star$ for some solution of modified Bellman equations $(g^\star, h^\infty)$.*

We defer the proofs to Appendix I. We note that rates of Bellman errors of Rx-RVI and Anc-RVI in Theorem 11 and 12 are exactly match to the rates of Rx-VI and Anc-VI in Corollary 4 and 5, respectively.

# E   PRELIMINARIES

In this section, we define some notations and introduce elementary propositions used in proofs.

## E.1   NOTATIONS

We denote $V \leq \tilde{V}$ if $V(s) \leq \tilde{V}(s)$ for all $s \in \mathcal{S}$ and $V, \tilde{V} \in \mathbb{R}^n$.

We denote $\Pi_{i=j}^{k} A_i = A_j A_{j+1} \dots A_k$ (ascending order) and $\Pi_{i=k}^{j} A_i = A_k A_{k-1} \dots A_j$ (descending order) where $0 \leq j \leq k$ and $A_i \in \mathbb{R}^{n \times n}$ for $j \leq i \leq k$. We define $\Pi_{i=j}^{k} A_i = 1$ and $\sum_{i=j}^{k} A_i = 0$ if $0 \leq k < j$.

We denote $P^\star = \lim_{k \to \infty} \frac{1}{k} \sum_{i=0}^{k} P^i$ for Cesaro limit of stochastic matrix $P$ (Cesaro limit of stochastic matrix always exist (Puterman, 2014, Theorem A.6)).

## E.2   PROPOSITIONS

**Proposition 1.** $B_1 \leq A \leq B_2$ *implies* $\|A\|_\infty \leq \max\{\|B_1\|_\infty, \|B_2\|_\infty\}$

*Proof.* By definition of $\|\cdot\|_\infty$, we get the desired result. □

**Proposition 2.** *If $P_1, P_2$ are stochastic matrices and $0 < a, b$, there exist stochastic matrix $P$ such that $aP_1 + bP_2 = (a+b)P$.*

*Proof.* Define $P(i,j) = (a+b)^{-1}(aP_1(i,j) + bP_2(i,j))$. Then, by simple calculation, we get the desired result. □

**Proposition 3.** *(Bertsekas, 2012, Lemma 1.1.1) If $V \leq \tilde{V}$, then $T^\pi U \leq T^\pi \tilde{V}, T^\star V \leq T^\star \tilde{V}$.*

**Proposition 4.** *For any policy $\pi$, $\mathcal{P}^\pi$ is a nonexpansive linear operator such that if $V \leq \tilde{V}$, $\mathcal{P}^\pi V \leq \mathcal{P}^\pi \tilde{V}$.*

*Proof.* If $r(s, a) = 0$ for all $s \in \mathcal{S}$ and $a \in \mathcal{A}$, $T^\pi = \mathcal{P}^\pi$. Then by Proposition 3, we have the desired result.

$\square$

**Proposition 5.** *For a stochastic matrix $P$, $P^\star P = PP^\star = P^\star$.*

*Proof.* By definition of $P^\star$, we get the desired result. $\square$

**Proposition 6.** *If $\mathcal{P}^{\pi_V} g^\star = g^\star$, $\|g^\star - g^{\pi_V}\|_\infty \leq \|TV - V - g^\star\|_\infty$ where $TV = T^{\pi_V} V$.*

*Proof.* Since $\mathcal{P}^{\pi_V} g^\star = g^\star$, $(\mathcal{P}^{\pi_V})^\star g^\star = g^\star$. Then $g^{\pi_V} - g^\star = (\mathcal{P}^{\pi_V})^\star (r^{\pi_V} - g^\star) = (\mathcal{P}^{\pi_V})^\star (r^{\pi_V} + \mathcal{P}^{\pi_V} V - V - g^\star) = (\mathcal{P}^{\pi_V})^\star (TV - V - g^\star)$, where second equality is from Proposition 5. Therefore, we have $\|g^\star - g^{\pi_k}\|_\infty \leq \|TV - V - g^\star\|_\infty$. $\square$

# F   OMITTED PROOFS OF THEOREMS FOR SECTION 3 AND B

In this section, we present omitted proofs convergence theorems of Rx-VI. We prove Theorems 7, 8, and 1 in turn.

## F.1   PROOF OF THEOREM 7

First, we prove the following lemma by induction.

**Lemma 1.** *For the iterates $\{V^k\}_{k=1,2,\dots}$ of Rx-VI,*

$$V^k = \Pi^0_{i=k-1}(\lambda_{i+1}I + (1-\lambda_{i+1})\mathcal{P}^{\pi_i})V^0 + \sum_{j=0}^{k-1} \left( \Pi^{j+1}_{i=k-1}(\lambda_{i+1}I + (1 - \lambda_{i+1})\mathcal{P}^{\pi_i}) \right) (1-\lambda_{j+1})r^{\pi_j}.$$

*Proof.* If $k = 1$, $V^1 = \lambda_1 V^0 + (1 - \lambda_1)TV^0 = (\lambda_1 I + (1 - \lambda_1)\mathcal{P}^{\pi_0})V^0 + (1 - \lambda_1)r^{\pi_0}$.

By induction,

$$V^{k+1}$$

$$= \lambda_{k+1} \left( \Pi^0_{i=k-1}(\lambda_{i+1}I + (1 - \lambda_{i+1})\mathcal{P}^{\pi_i})V^0 \right.$$

$$\left. + \sum_{j=0}^{k-1} \left( \Pi^{j+1}_{i=k-1}(\lambda_{i+1}I + (1 - \lambda_{i+1})\mathcal{P}^{\pi_i}) \right) (1 - \lambda_{j+1})r^{\pi_j} \right)$$

$$+ (1 - \lambda_{k+1}) \left( \mathcal{P}^{\pi_k} \left( \Pi^0_{i=k-1}(\lambda_{i+1}I + (1 - \lambda_{i+1})\mathcal{P}^{\pi_i})V^0 \right. \right.$$

$$\left. \left. + \sum_{j=0}^{k-1} \left( \Pi^{j+1}_{i=k-1}(\lambda_{i+1}I + (1 - \lambda_{i+1})\mathcal{P}^{\pi_i}) \right) (1 - \lambda_{j+1})r^{\pi_j} \right) + r^{\pi_k} \right)$$

$$= \Pi^0_{i=k}(\lambda_{i+1}I + (1 - \lambda_{i+1})\mathcal{P}^{\pi_i})V^0 + \sum_{j=0}^{k} \left( \Pi^{j+1}_{i=k}(\lambda_{i+1}I + (1 - \lambda_{i+1})\mathcal{P}^{\pi_i}) \right) (1 - \lambda_{j+1})r^{\pi_j}.$$

$\square$

Now, we prove following key lemma.

**Lemma 2.** *For the iterates $\{V^k\}_{k=0,1,2,\ldots}$ of Rx-VI,*

$$V^k - \sum_{j=1}^{k}(1-\lambda_j)g^\star \le \Pi^0_{i=k-1}(\lambda_{i+1}I + (1-\lambda_{i+1})\mathcal{P}^{\pi_i})(V^0 - h^\star) + h^\star$$

$$h^\star + \Pi^0_{i=k-1}(\lambda_{i+1}I + (1-\lambda_{i+1})\mathcal{P}^{\pi_\star})(V_0 - h^\star) \le V^k - \sum_{j=1}^{k}(1-\lambda_j)g^\star.$$

*Proof.* For the first inequality, we have
$V^k$

$$= \Pi^0_{i=k-1}(\lambda_{i+1}I + (1-\lambda_{i+1})\mathcal{P}^{\pi_i})V^0 + \sum_{j=0}^{k-1}\left(\Pi^{j+1}_{i=k-1}(\lambda_{i+1}I + (1-\lambda_{i+1})\mathcal{P}^{\pi_i})\right)(1-\lambda_{j+1})r^{\pi_j}$$

$$\le \Pi^0_{i=k-1}(\lambda_{i+1}I + (1-\lambda_{i+1})\mathcal{P}^{\pi_i})V^0$$
$$+ \sum_{j=0}^{k-1}\left(\Pi^{j+1}_{i=k-1}(\lambda_{i+1}I + (1-\lambda_{i+1})\mathcal{P}^{\pi_i})\right)(1-\lambda_{j+1})(g^\star + (I - \mathcal{P}^{\pi_j})h^\star)$$

$$\le \Pi^0_{i=k-1}(\lambda_{i+1}I + (1-\lambda_{i+1})\mathcal{P}^{\pi_i})V^0$$
$$+ \sum_{j=0}^{k-1}\Pi^{j+1}_{i=k-1}(\lambda_{i+1}I + (1-\lambda_{i+1})\mathcal{P}^{\pi_i})(I - \lambda_{j+1}I - (1-\lambda_{j+1})\mathcal{P}^{\pi_j}) + \sum_{j=1}^{k}(1-\lambda_j)g^\star$$

$$= \Pi^0_{i=k-1}(\lambda_{i+1}I + (1-\lambda_{i+1})\mathcal{P}^{\pi_i})V^0 + (I - \Pi^0_{i=k-1}(\lambda_{i+1}I + (1-\lambda_{i+1})\mathcal{P}^{\pi_i}))h^\star$$
$$+ \sum_{j=1}^{k}(1-\lambda_i)g^\star$$

$$= \Pi^0_{i=k-1}(\lambda_{i+1}I + (1-\lambda_{i+1})\mathcal{P}^{\pi_i})(V^0 - h^\star) + h^\star + \sum_{j=1}^{k}(1-\lambda_i)g^\star,$$

where first equality comes form Lemma 1, first inequality follows from second Bellman equation, second inequality follows from first Bellman equation, and second equality is from telescoping-sum argument.

We now prove the second inequality.
$V^k$

$$= \Pi^0_{i=k-1}(\lambda_{i+1}I + (1-\lambda_{i+1})\mathcal{P}^{\pi_i})V^0 + \sum_{j=0}^{k-1}\left(\Pi^{j+1}_{i=k-1}(\lambda_{i+1}I + (1-\lambda_{i+1})\mathcal{P}^{\pi_i})\right)(1-\lambda_{j+1})r^{\pi_j}$$

$$\ge \Pi^0_{i=k-1}(\lambda_{i+1}I + (1-\lambda_{i+1})\mathcal{P}^{\pi_\star})V^0 + \sum_{j=0}^{k-1}\Pi^{j+1}_{i=k-1}(\lambda_{i+1}I + (1-\lambda_{i+1})\mathcal{P}^{\pi_\star})(1-\lambda_{j+1})r^{\pi_\star}$$

$$= \Pi^0_{i=k-1}(\lambda_{i+1}I + (1-\lambda_{i+1})\mathcal{P}^{\pi_\star})V^0$$
$$+ \sum_{j=0}^{k-1}\Pi^{j+1}_{i=k-1}(\lambda_{i+1}I + (1-\lambda_{i+1})\mathcal{P}^{\pi_\star})(1-\lambda_{j+1})(g^\star + (I - \mathcal{P}^{\pi_\star})h^\star)$$

$$= \sum_{j=1}^{k}(1-\lambda_i)g^\star + h^\star + \Pi^0_{i=k-1}(\lambda_{i+1} + (1-\lambda_{i+1})\mathcal{P}^{\pi_\star})(V_0 - h^\star)$$

where first inequality follows from Lemma 3 and the fact that $\{\pi_l\}_{l=0,1,\ldots,k}$ are greedy policies, first equality comes from second Bellman equation, and second equality is from first Bellman equation. $\square$

We are now ready to prove Theorem 7 .

*Proof of Theorem 7 .* By Lemma 2, we have

$$V^k - V^0 - \sum_{j=1}^{k}(1-\lambda_j)g^\star \leq \Pi_{i=k-1}^0(\lambda_{i+1}I + (1-\lambda_{i+1})\mathcal{P}^{\pi_i})(V^0 - h^\star) + h^\star - V^0$$

$$= \left(\Pi_{i=k-1}^0(\lambda_{i+1}I + (1-\lambda_{i+1})\mathcal{P}^{\pi_i}) - (\Pi_{i=k-1}^0\lambda_{i+1})I\right)(V^0 - h^\star)$$
$$- (1 - \Pi_{i=k-1}^0\lambda_{i+1})I(V^0 - h^\star),$$

and

$$V^k - V^0 - \sum_{j=1}^{k}(1-\lambda_j)g^\star \geq \Pi_{i=k-1}^0(\lambda_{i+1} + (1-\lambda_{i+1})\mathcal{P}^{\pi_\star})(V_0 - h^\star) + h^\star - V^0$$

$$= \left(\Pi_{i=k-1}^0(\lambda_{i+1}I + (1-\lambda_{i+1})\mathcal{P}^{\pi_\star}) - (\Pi_{i=k-1}^0\lambda_{i+1})I\right)(V^0 - h^\star)$$
$$- (1 - \Pi_{i=k-1}^0\lambda_{i+1})I(V^0 - h^\star).$$

By Proposition 1, this implies

$$\left\| V^k - V^0 - \sum_{j=1}^{k}(1-\lambda_j)g^\star \right\|_\infty \leq 2(1 - \Pi_{i=1}^k\lambda_i)\left\| V^0 - h^\star \right\|_\infty.$$

Hence, we conclude

$$\left\| \frac{V^k - V^0}{\sum_{j=1}^k(1-\lambda_j)} - g^\star \right\|_\infty \leq \frac{2(1 - \Pi_{i=1}^k\lambda_i)}{\sum_{j=1}^k(1-\lambda_j)}\left\| V^0 - h^\star \right\|_\infty.$$

$\square$

## F.2 PROOF OF THEOREM 8

First, define $a_j^k = \left(\Pi_{i=j+1}^k\lambda_i\right)(1-\lambda_j)$ for $0 \leq j \leq k$ and $a_0^0 = 1$, where $\{\lambda_k\}_{k\in\mathbf{N}} \in [0,1]$. Let $\lambda_0 = 0$ for computational conciseness. Following lemma will simplify calculation in later proof.

**Lemma 3.** *For $0 \leq k_2 < k_1$ ,*

$$\sum_{j=i}^{l} a_j^k = \Pi_{s=l+1}^k\lambda_s - \Pi_{s=i}^k\lambda_s, \qquad a_i^{k_1} - a_i^{k_2} = -\sum_{j=k_2+1}^{k_1} a_j^{k_1}a_i^{k_2},$$

$$\sum_{i=k_2+1}^{k_1}\sum_{j=0}^{k_2} a_j^{k_2}a_i^{k_1}\left(\sum_{l=j}^{i-1}(1-\lambda_l)\right) = \sum_{i=k_2+1}^{k_1}(1-\lambda_l).$$

*Proof.* First and second equality can be proved by simple calculation. With first and second equality, we prove third equality as follows.

$$\sum_{i=k_2+1}^{k_1} \sum_{j=0}^{k_2} a_j^{k_2} a_i^{k_1} \left( \sum_{l=j}^{i-1} (1 - \lambda_l) \right) = \sum_{i=k_2+1}^{k_1} \sum_{l=0}^{k_2} a_i^{k_1} (1 - \lambda_l) \left( \sum_{j=0}^{l} a_j^{k_2} \right)$$

$$+ \sum_{i=k_2+2}^{k_1} \sum_{l=k_2+1}^{i-1} a_i^{k_1} (1 - \lambda_l) \left( \sum_{j=0}^{k_2} a_j^{k_2} \right)$$

$$= \sum_{i=k_2+1}^{k_1} \sum_{l=0}^{k_2} a_i^{k_1} a_l^{k_2} + \sum_{i=k_2+2}^{k_1} \sum_{l=k_2+1}^{i-1} a_i^{k_1} (1 - \lambda_l)$$

$$= \sum_{i=k_2+1}^{k_1} a_i^{k_1} + \sum_{l=k_2+1}^{k_1-1} \sum_{i=l+1}^{k_1} (1 - \lambda_l) a_i^{k_1}$$

$$= 1 - \Pi_{l=k_2+1}^{k_1} \lambda_l + \sum_{l=k_2+1}^{k_1-1} (1 - \lambda_l)(1 - \Pi_{j=l+1}^{k_1} \lambda_j)$$

$$= \sum_{l=k_2+1}^{k_1-1} (1 - \lambda_l) + 1 - \Pi_{l=k_2+1}^{k_1} \lambda_l - \sum_{l=k_2+1}^{k_1-1} a_l^{k_1}$$

$$= \sum_{l=k_2+1}^{k_1} (1 - \lambda_l).$$

$\square$

By simple calculation, for the iterates $\{V^k\}_{k=0,1,2,\dots}$ of Rx-VI,

$$V^k = \sum_{j=0}^{k} \left( \Pi_{i=j+1}^{k} \lambda_i \right) (1 - \lambda_j) TV^{j-1},$$

where $TV^{-1} = V^0$, and we have following lemma.

**Lemma 4.** *For the iterates $\{V^k\}_{k=0,1,2,\dots}$ of Rx-VI,*

$$V^{k_1} - V^{k_2} = \sum_{j=0}^{k_2} \sum_{i=k_2+1}^{k_1} a_j^{k_2} a_i^{k_1} (TV^{i-1} - TV^{j-1})$$

*for $0 \leq k_2 \leq k_1$.*

*Proof.* By definition, we have

$$V^{k_1} - V^{k_2} = \sum_{i=0}^{k_1} a_i^{k_1} TV^{i-1} - \sum_{i=0}^{k_2} a_i^{k_2} TV^{i-1}$$

$$= \sum_{i=k_2+1}^{k_1} a_i^{k_1} TV^{i-1} + \sum_{i=0}^{k_2} (a_i^{k_1} - a_i^{k_2}) TV^{i-1}$$

$$= \sum_{j=0}^{k_2} \sum_{i=k_2+1}^{k_1} a_j^{k_2} a_i^{k_1} TV^{i-1} - \sum_{i=0}^{k_2} \sum_{j=k_2+1}^{k_1} a_j^{k_1} a_i^{k_2} TV^{i-1}$$

$$= \sum_{j=0}^{k_2} \sum_{i=k_2+1}^{k_1} a_j^{k_2} a_i^{k_1} (TV^{i-1} - TV^{j-1}).$$

where third equality is from Lemma 3.

$\square$

Following lemma will be used in proof in later proof.

**Lemma 5.** *For the iterates $\{V^k\}_{k=0,1,2,\dots}$ of Rx-VI,*

$$TV^k - V^0 \le \sum_{j=0}^{k}(1-\lambda_j)g^\star + h^\star - V^0 + \mathcal{P}^{\pi_k}\left(\Pi_{i=k-1}^0(\lambda_{i+1}I + (1-\lambda_{i+1})\mathcal{P}^{\pi_i})(V^0 - h^\star)\right),$$

$$TV^k - V^0 \ge \sum_{j=0}^{k}(1-\lambda_j)g^\star + h^\star - V^0 + \mathcal{P}^{\pi_\star}\left(\Pi_{i=k-1}^0(\lambda_{i+1}I + (1-\lambda_{i+1})\mathcal{P}^{\pi_\star})(V^0 - h^\star)\right).$$

*Proof.* We have

$$TV^k \le \mathcal{P}^{\pi_k}\left(\sum_{j=1}^{k}(1-\lambda_j)g^\star + h^\star + \Pi_{i=k-1}^0(\lambda_{i+1}I + (1-\lambda_{i+1})\mathcal{P}^{\pi_i})(V^0 - h^\star)\right) + r^{\pi_k}$$

$$\le \sum_{j=1}^{k}(1-\lambda_j)g^\star + \mathcal{P}^{\pi_k}\left(\Pi_{i=k-1}^0(\lambda_{i+1}I + (1-\lambda_{i+1})\mathcal{P}^{\pi_i})(V^0 - h^\star)\right)$$
$$+ \mathcal{P}^{\pi_k}h^\star + g^\star + (I - \mathcal{P}^{\pi_k})h^\star$$

$$= \sum_{j=0}^{k}(1-\lambda_j)g^\star + \mathcal{P}^{\pi_k}\left(\Pi_{i=k-1}^0(\lambda_{i+1}I + (1-\lambda_{i+1})\mathcal{P}^{\pi_i})(V^0 - h^\star)\right) + h^\star,$$

where first inequality is from Lemma 3 and 2, second inequality comes from second Bellman equation, and last equality follows from first Bellman equation.

Also, we have

$$TV^k \ge \mathcal{P}^{\pi_\star}\left(\sum_{j=1}^{k}(1-\lambda_j)g^\star + h^\star + \Pi_{i=k-1}^0(\lambda_{i+1}I + (1-\lambda_{i+1})\mathcal{P}^{\pi_\star})(V^0 - h^\star)\right) + r^{\pi_\star}$$

$$= \sum_{j=1}^{k}(1-\lambda_j)g^\star + \mathcal{P}^{\pi_\star}\left(\Pi_{i=k-1}^0(\lambda_{i+1}I + (1-\lambda_{i+1})\mathcal{P}^{\pi_\star})(V^0 - h^\star)\right)$$
$$+ \mathcal{P}^{\pi_\star}h^\star + g^\star + (I - \mathcal{P}^{\pi_\star})h^\star$$

$$= \sum_{j=0}^{k}(1-\lambda_j)g^\star + \mathcal{P}^{\pi_\star}\left(\Pi_{i=k-1}^0(\lambda_{i+1}I + (1-\lambda_{i+1})\mathcal{P}^{\pi_\star})(V^0 - h^\star)\right) + h^\star,$$

where first inequality is from Lemma 3 and 2 and the fact that $\{\pi_l\}_{l=0,1,\dots,k}$ are greedy policies, first equality comes from second Bellman equation, and last equality follows from first Bellman equation. $\square$

We now prove one of key lemmas for Theorem 8. For that, define

$$c_{k_1,k_2} = \sum_{j=0}^{k_2}\sum_{i=k_2+1}^{k_1} a_j^{k_2}a_i^{k_1}c_{i-1,j-1}$$

for $0 \le k_2 < k_1$ and $c_{n,-1} = 1, c_{k,k} = 0$ for all $0 \le k$. Note that $a_j^k = \left(\Pi_{i=j+1}^k\lambda_i\right)(1-\lambda_j)$ and $a_0^0 = 1$ for $0 \le j \le k$. Then, we have following lemma.

**Lemma 6.** *For the iterates $\{V^k\}_{k=0,1,2,\dots}$ of Rx-VI and $0 \le k_2 \le k_1$,*

$$V^{k_1} - V^{k_2} \le \sum_{i=k_2+1}^{k_1}(1-\lambda_i)g^\star + c_{k_1,k_2}(S_1^{k_1,k_2} - S_2^{k_1,k_2})(V^0 - h^\star),$$

*where $S_1^{k_1,k_2}, S_2^{k_1,k_2}$ are stochastic matrices.*

*Proof.* We use induction on $k_2$. Let $k_2 = 0$. Then, $c_{k,0} = \sum_{i=1}^{k} a_i^k = 1 - \Pi_{i=1}^{k} \lambda_i$ ($a_0^0 = 1$) by Lemma 3. Also, by Lemma 2, we have

$$V^k - V^0 - \sum_{j=1}^{k}(1 - \lambda_j)g^\star$$

$$\leq h^\star - V^0 + \Pi_{i=k-1}^{0}(\lambda_{i+1}I + (1 - \lambda_{i+1})\mathcal{P}^{\pi_i})(V^0 - h^\star)$$

$$= (1 - \Pi_{i=k-1}^{0}\lambda_{i+1})I(V^0 - h^\star) + (\Pi_{i=k-1}^{0}(\lambda_{i+1}I + (1 - \lambda_{i+1})\mathcal{P}^{\pi_i}) - (\Pi_{i=k-1}^{0}\lambda_{i+1})I)(V^0 - h^\star)$$

$$= c_{k,0}(S_1^{k,0} - S_2^{k,0})(V^0 - h^\star)$$

where $S_1^{k,0} = I$ and $S_2^{k,0} = (1 - \Pi_{i=k-1}^{0}\lambda_{i+1})^{-1}\left(\Pi_{i=k-1}^{0}(\lambda_{i+1}I + (1 - \lambda_{i+1})\mathcal{P}^{\pi_i}) - (\Pi_{i=k-1}^{0}\lambda_{i+1})I\right)$.

By induction,

$$V^{k_1} - V^{k_2}$$

$$= \sum_{i=k_2+1}^{k_1} \sum_{j=0}^{k_2} a_j^{k_2} a_i^{k_1}(TV^{i-1} - TV^{j-1})$$

$$= \sum_{i=k_2+1}^{k_1} \sum_{j=1}^{k_2} a_j^{k_2} a_i^{k_1}(TV^{i-1} - TV^{j-1}) + \sum_{i=k_2+1}^{k_1} a_0^{k_2} a_i^{k_1}(TV^{i-1} - V^0)$$

$$\leq \sum_{i=k_2+1}^{k_1} \sum_{j=1}^{k_2} a_j^{k_2} a_i^{k_1}(\mathcal{P}^{\pi_{i-1}}V^{i-1} + r^{\pi_{i-1}} - \mathcal{P}^{\pi_{i-1}}V^{j-1} - r^{\pi_{i-1}}) + \sum_{i=k_2+1}^{k_1} a_0^{k_2} a_i^{k_1}(TV^{i-1} - V^0)$$

$$= \sum_{i=k_2+1}^{k_1} \sum_{j=1}^{k_2} a_j^{k_2} a_i^{k_1} \mathcal{P}^{\pi_{i-1}}(V^{i-1} - V^{j-1}) + \sum_{i=k_2+1}^{k_1} a_0^{k_2} a_i^{k_1}(TV^{i-1} - V^0)$$

$$\leq \sum_{i=k_2+1}^{k_1} \sum_{j=1}^{k_2} a_j^{k_2} a_i^{k_1} \mathcal{P}^{\pi_{i-1}}\left(\sum_{l=j}^{i-1}(1 - \lambda_l)g^\star + c_{i-1,j-1}(S_1^{i-1,j-1} - S_2^{i-1,j-1})(V^0 - h^\star)\right)$$

$$+ \sum_{i=k_2+1}^{k_1} a_0^{k_2} a_i^{k_1}\left(\sum_{l=0}^{i-1}(1 - \lambda_l)g^\star + \left(\mathcal{P}^{\pi_{i-1}}\left(\Pi_{l=i-2}^{0}(\lambda_{l+1}I + (1 - \lambda_{l+1})\mathcal{P}^{\pi_l}) - I\right)(V^0 - h^\star)\right)\right)$$

$$= \sum_{i=k_2+1}^{k_1} \sum_{j=0}^{k_2} a_j^{k_2} a_i^{k_1} c_{i-1,j-1}(S_x^{i,j} - S_y^{i,j})(V^0 - h^\star) + \sum_{i=k_2+1}^{k_1} \sum_{j=0}^{k_2} a_j^{k_2} a_i^{k_1}\left(\sum_{l=j}^{i-1}(1 - \lambda_l)g^\star\right)$$

$$= c_{k_1,k_2}(S_1^{k_1,k_2} - S_2^{k_1,k_2})(V^0 - h^\star) + \sum_{i=k_2+1}^{k_1}(1 - \lambda_i)g^\star.$$

where first equality is from Lemma 4, first inequality comes from the fact that $\{\pi_l\}_{l=0,1,\ldots,k_1}$ are greedy policies, second inequality follows from induction and Lemma 5, last equality is from Lemma 3, and

$$S_x^{i,j} = \begin{cases} \mathcal{P}^{\pi_{i-1}}\Pi_{l=i-2}^{0}(\lambda_{l+1}I + (1 - \lambda_{l+1})\mathcal{P}^{\pi_l}) & j = 0, \\ \mathcal{P}^{\pi_{i-1}}S_1^{i-1,j-1} & \text{else,} \end{cases} \qquad S_y^{i,j} = \begin{cases} I & j = 0, \\ \mathcal{P}^{\pi_{i-1}}S_2^{i-1,j-1} & \text{else,} \end{cases}$$

and $S_1^{k_1,k_2} = c_{k_1,k_2}^{-1}\sum_{i=k_2+1}^{k_1}\sum_{j=0}^{k_2} a_j^{k_2} a_i^{k_1} c_{i-1,j-1} S_x^{i,j}$, and $S_2^{k_1,k_2} = c_{k_1,k_2}^{-1}\sum_{i=k_2+1}^{k_1}\sum_{j=0}^{k_2} a_j^{k_2} a_i^{k_1} c_{i-1,j-1} S_y^{i,j}$.

$\square$

To obtain lower bound of $V^{k_1} - V^{k_2}$, we need more sophisticated consideration, and following lemma is necessary for later argument.

**Lemma 7.** *Let $\{V^k\}_{k=0,1,2,\ldots}$ be the iterates of Rx-VI. Let $\limsup \lambda_k < 1$. Let $E = \{\pi : \mathcal{P}^\pi g^\star = g^\star\}$. Then there exist $K$ such that if $K \leq k$,*

$$TV^k = \max_{\pi \in E} T^\pi V^k.$$

*Proof.* Suppose $\pi$ is infinitely often repeated deterministic policy among $\{\pi_k\}_{k=0,1,2,...}$. Then there exist increasing sequence $k_n$ such that $\pi_{k_n} = \pi$ and $\lambda_{k_n}$ converge to some $\lambda < 1$. Then, since $V^{k_K+1} = \lambda_{k_n+1}V^{n_k} + (1 - \lambda_{k_n+1})TV^{k_n}$, we have

$$\frac{V^{k_n+1}}{\sum_{i=1}^{k_n+1}(1-\lambda_i)} = \lambda_{k_n+1}\frac{V^{k_n}}{\sum_{i=1}^{k_n+1}(1-\lambda_i)} + (1-\lambda_{k_n+1})\mathcal{P}^\pi \frac{V^{k_n}}{\sum_{i=1}^{k_n+1}(1-\lambda_i)}$$
$$+ (1-\lambda_{k_n+1})\frac{r^\pi}{\sum_{i=1}^{k_n+1}(1-\lambda_i)}.$$

If $k_n \to \infty$, $\limsup \lambda_k < 1$ implies $\sum_{j=1}^{\infty}(1-\lambda_j) = \infty$. Then, by Theorem 7, we have

$$g^\star = \lambda g^\star + (1-\lambda)\mathcal{P}^\pi g^\star.$$

Thus $g^\star = \mathcal{P}^\pi g^\star$ and this implies $\pi \in E$. By finiteness of action and state space, number of infinitely repeated policy $\pi$ is also finite. Therefore there exist $K$ such that $TV^k = \max_{\pi \in E} T^\pi V^k$ for $K \leq k$. $\qquad\square$

We are now ready to prove left key lemma. To obtain proper lower bound of $V^{k_1} - V^{k_2}$, roughly speaking, we need to consider $V^K$ as initial point where $N$ is iteration number in Lemma 13. For that, define $\{\lambda'_k\}_{K \leq k}$ such that $\lambda'_k = \lambda_k$ for $K+1 \leq k$ and $\lambda'_K = 0$, $b_j^k = \left(\Pi_{i=j+1}^k \lambda'_i\right)(1-\lambda'_j)$ for $K \leq j \leq k$, and $b_K^K = 1$. Also, define

$$c_{k_1,k_2}^K = \sum_{j=N}^{k_2}\sum_{i=k_2+1}^{k_1} b_j^{k_2} b_i^{k_1} c_{i-1,j-1}^K$$

for $0 \leq k_2 < k_1$ and $c_{k,K-1}^K = 1$ for all $K \leq k$. Note that if $K = 0$, $\lambda'_k = \lambda_k$, $b_j^k = a_j^k$, and $c_{k_1,k_2}^K = c_{k_1,k_2}$ for all $0 \leq k_1, k_2, k, j, k$.

**Lemma 8.** *Let* $\{V^k\}_{k=0,1,2,...}$ *be the iterates of Rx-VI. Suppose there exist $K$ such that if $K \leq k$, $TV^k = \max_{\pi \in E} T^\pi V^k$ where $E = \{\pi : \mathcal{P}^\pi g^\star = g^\star\}$. Then, for $K \leq k_2' \leq k_1'$,*

$$\sum_{i=k_2+1}^{k_1}(1-\lambda'_i)g^\star + c_{k_1,k_2}^K(S_{1'}^{k_1,k_2} - S_{2'}^{k_1,k_2})(V^0 - h^\star) \leq V^{k_1'} - V^{k_2'}$$

*where* $S_{1'}^{k_1,k_2}, S_{2'}^{k_1,k_2}$ *are stochastic matrices.*

*Proof.* For $K \leq k$, by simple calculation, we have

$$V^k = \Pi_{i=k-1}^K(\lambda'_{i+1}I + (1-\lambda'_{i+1})\mathcal{P}^{\pi_i})V^K + \sum_{j=K}^{k-1}\Pi_{i=k-1}^{j+1}(\lambda'_{i+1}I + (1-\lambda'_{i+1})\mathcal{P}^{\pi_i})(1-\lambda'_{j+1})r^{\pi_j}$$

and

$$V^k = \Pi_{i=k-1}^K(\lambda'_{i+1}I + (1-\lambda'_{i+1})\mathcal{P}^{\pi_i})V^K + \sum_{j=K}^{k-1}\Pi_{i=k-1}^{j+1}(\lambda'_{i+1}I + (1-\lambda'_{i+1})\mathcal{P}^{\pi_i})(1-\lambda'_{j+1})r^{\pi_j}$$
$$\geq \Pi_{i=k-1}^K(\lambda'_{i+1}I + (1-\lambda'_{i+1})\mathcal{P}^{\pi_\star})V^K + \sum_{j=K}^{k-1}\Pi_{i=k-1}^{j+1}(\lambda'_{i+1}I + (1-\lambda'_{i+1})\mathcal{P}^{\pi_\star})(1-\lambda'_{j+1})r^{\pi_\star}$$
$$= \Pi_{i=k-1}^K(\lambda'_{i+1}I + (1-\lambda'_{i+1})\mathcal{P}^{\pi_\star})V^K$$
$$+ \sum_{j=K}^{k-1}\Pi_{i=k-1}^{j+1}(\lambda'_{i+1}I + (1-\lambda'_{i+1})\mathcal{P}^{\pi_\star})(1-\lambda'_{j+1})(g^\star + (I - \mathcal{P}^{\pi_\star})h^\star)$$
$$= \sum_{j=K+1}^k(1-\lambda'_j)g^\star + h^\star + \Pi_{i=k-1}^K(\lambda'_{i+1}I + (1-\lambda'_{i+1})\mathcal{P}^{\pi_\star})(V^K - h^\star)$$

where first equality comes from previous equality, first inequality follows the fact that $\{\pi_l\}_{l=K,K+1,\ldots,k}$ are greedy policies, second equality comes from second Bellman equation, and last equality is from first Bellman equation.

Now, we use induction on $k_2$. If $k_2 = K$, by previous inequality,

$$V^{k_1} - V^K - \sum_{j=K+1}^{k_1} (1 - \lambda'_j)g^\star$$

$$\geq h^\star - V^K + \Pi_{i=k_1-1}^{K}(\lambda'_{i+1}I + (1 - \lambda'_{i+1})\mathcal{P}^{\pi_\star})(V^K - h^\star)$$

$$= (\Pi_{i=k_1-1}^{K}(\lambda'_{i+1}I + (1 - \lambda'_{i+1})\mathcal{P}^{\pi_\star}) - (\Pi_{i=k_1-1}^{K}\lambda'_{i+1})I)(V^K - h^\star) - (1 - \Pi_{i=k_1-1}^{K}\lambda'_{i+1})(V^K - h^\star)$$

$$\geq (\Pi_{i=k_1-1}^{K}(\lambda'_{i+1}I + (1 - \lambda'_{i+1})\mathcal{P}^{\pi_\star}) - (\Pi_{i=k_1-1}^{K}\lambda'_{i+1})I)(\Pi_{i=K-1}^{0}(\lambda_{i+1}I + (1 - \lambda_{i+1})\mathcal{P}^{\pi_\star})(V^0 - h^\star)$$

$$\quad - (1 - \Pi_{i=k_1-1}^{K}\lambda'_{i+1})(\Pi_{i=K-1}^{0}(\lambda_{i+1}I + (1 - \lambda_{i+1})\mathcal{P}^{\pi_i})(V_0 - h^\star)$$

$$= c_{k_1,K}^{K}(S_{1'}^{k_1,K} - S_{2'}^{k_1,K})(V^0 - h^\star)$$

where second inequality comes from Lemma 2 and first Bellman equation (note that $g^\star$ terms cancel out), and

$$S_{1'}^{k_1,K} = (1 - \Pi_{i=k_1-1}^{K}\lambda'_{i+1})^{-1} \left( \Pi_{i=k_1-1}^{K}(\lambda'_{i+1}I + (1 - \lambda'_{i+1})\mathcal{P}^{\pi_\star}) - (\Pi_{i=k_1-1}^{K}\lambda'_{i+1})I \right)$$

$$\quad \times (\Pi_{i=K-1}^{0}(\lambda_{i+1}I + (1 - \lambda_{i+1})\mathcal{P}^{\pi_\star}),$$

$$S_{2'}^{k_1,K} = \Pi_{i=K-1}^{0}(\lambda_{i+1}I + (1 - \lambda_{i+1})\mathcal{P}^{\pi_i}),$$

$$c_{k_1,K}^{K} = 1 - \Pi_{i=k_1-1}^{K}\lambda'_{i+1}.$$

By induction,

$$V^{k_1} - V^{k_2}$$

$$= \sum_{i=k_2+1}^{k_1} \sum_{j=K}^{k_2} b_j^{k_2} b_i^{k_1}(TV^{i-1} - TV^{j-1})$$

$$= \sum_{i=k_2+1}^{k_1} \sum_{j=K+1}^{k_2} b_j^{k_2} b_i^{k_1}(TV^{i-1} - TV^{j-1}) + \sum_{i=k_2+1}^{k_1} b_K^{k_2} b_i^{k_1}(TV^{i-1} - V^K)$$

$$\geq \sum_{i=k_2+1}^{k_1} \sum_{j=K+1}^{k_2} b_j^{k_2} b_i^{k_1}(\mathcal{P}^{\pi_{j-1}}V^{i-1} + r^{\pi_{j-1}} - \mathcal{P}^{\pi_{j-1}}V^{j-1} - r^{\pi_{j-1}}) + \sum_{i=k_2+1}^{k_1} b_K^{k_2} b_i^{k_1}(TV^{i-1} - V^K)$$

$$= \sum_{i=k_2+1}^{k_1} \sum_{j=K+1}^{k_2} b_j^{k_2} b_i^{k_1}\mathcal{P}^{\pi_{j-1}}(V^{i-1} - V^{j-1}) + \sum_{i=k_2+1}^{k_1} b_K^{k_2} b_i^{k_1}(TV^{i-1} - V^K)$$

$$\geq \sum_{i=k_2+1}^{k_1} \sum_{j=K+1}^{k_2} b_j^{k_2} b_i^{k_1}\mathcal{P}^{\pi_{j-1}} \left( \sum_{l=j}^{i-1}(1 - \lambda'_l)g^\star + c_{i-1,j-1}(S_{1'}^{i-1,j-1} - S_{2'}^{i-1,j-1})(V^0 - h^\star) \right)$$

$$\quad + \sum_{i=k_2+1}^{k_1} b_K^{k_2} b_i^{k_1} \left( \sum_{l=K}^{i-1}(1 - \lambda'_l)g^\star + \left( \mathcal{P}^{\pi_\star} \left( \Pi_{l=i-2}^{K}(\lambda'_{l+1}I + (1 - \lambda'_{l+1})\mathcal{P}^{\pi_\star}) - I \right)(V^K - h^\star) \right) \right)$$

$$
\geq \sum_{i=k_2+1}^{k_1} \sum_{j=K+1}^{k_2} b_j^{k_2} b_i^{k_1} \mathcal{P}^{\pi_{j-1}} \left( \sum_{l=j}^{i-1} (1-\lambda_l') g^\star + c_{i-1,j-1}(S_{1'}^{i-1,j-1} - S_{2'}^{i-1,j-1})(V^0 - h^\star) \right)
$$

$$
+ \sum_{i=k_2+1}^{k_1} b_K^{k_2} b_i^{k_1} \left( \sum_{l=K}^{i-1} (1-\lambda_l') g^\star + \left( \mathcal{P}^{\pi_\star} \left( \Pi_{l=i-2}^{K}(\lambda_{l+1}' I + (1-\lambda_{l+1}')\mathcal{P}^{\pi_\star}) \right. \right.\right.
$$

$$
\left.\left.\left. \times (\Pi_{i=K-1}^0 (\lambda_{i+1} I + (1-\lambda_{i+1})\mathcal{P}^{\pi_\star})) \right) - (\Pi_{i=K-1}^0 (\lambda_{i+1} I + (1-\lambda_{i+1})\mathcal{P}^{\pi_i})) \right) (V^0 - h^\star) \right)
$$

$$
= \sum_{i=k_2+1}^{k_1} \sum_{j=K}^{k_2} b_j^{k_2} b_i^{k_1} c_{i-1,j-1}^K (S_{x'}^{i,j} - S_{y'}^{i,j})(V^0 - h^\star) + \sum_{i=k_2+1}^{k_1} \sum_{j=K}^{k_2} b_j^{k_2} b_i^{k_1} \left( \sum_{l=j}^{i-1} (1-\lambda_l') g^\star \right)
$$

$$
= c_{k_1,k_2}^K (S_{1'}^{k_1,k_2} - S_{2'}^{k_1,k_2})(V^0 - h^\star) + \sum_{i=k_2+1}^{k_1} (1-\lambda_i) g^\star.
$$

where first equality is from similar argument in the proof of Lemma 4, first inequality comes from the fact that $\{\pi_l\}_{l=K,K+1,\ldots,k_1}$ are greedy policies, second inequality follows from induction and similar argument in the proof of Lemma 5, last inequality is from Lemma 2 and first Bellman equation (note that $g^\star$ terms cancel out), second from the last equality is from same argument in the proof of Lemma 3, and

$$
S_{x'}^{i,j} = \begin{cases} \mathcal{P}^{\pi_\star} \Pi_{l=i-2}^K (\lambda_{l+1}' I + (1-\lambda_{l+1}')\mathcal{P}^{\pi_\star})(\Pi_{i=K-1}^0(\lambda_{i+1} I + (1-\lambda_{i+1})\mathcal{P}^{\pi_\star})) & j = K, \\ \mathcal{P}^{\pi_{j-1}} S_{1'}^{i-1,j-1} & \text{else,} \end{cases}
$$

$$
S_{y'}^{i,j} = \begin{cases} \Pi_{i=K-1}^0(\lambda_{i+1} I + (1-\lambda_{i+1})\mathcal{P}^{\pi_i}) & j = K, \\ \mathcal{P}^{\pi_{j-1}} S_{2'}^{i-1,j-1} & \text{else,} \end{cases}
$$

$$
S_{1'}^{k_1,k_2} = (c_{k_1,k_2}^K)^{-1} \sum_{i=k_2+1}^{k_1} \sum_{j=K}^{k_2} b_j^{k_2} b_i^{k_1} c_{i-1,j-1}^K S_{x'}^{i,j},
$$

$$
S_2^{k_1,k_2} = (c_{k_1,k_2}^K)^{-1} \sum_{i=k_2+1}^{k_1} \sum_{j=0}^{k_2} b_j^{k_2} b_i^{k_1} c_{i-1,j-1}^K S_{y'}^{i,j}.
$$

$\square$

For the explicit convergence rate of Theorem 8, we will use the following Fact.

**Fact 5.** *(Cominetti et al., 2014, Section 2.3) For $0 < k$ and $K \leq k'$,*

$$
(1-\lambda_{k+1})^{-1} c_{k+1,k} \leq \frac{2}{\sqrt{\pi \sum_{i=1}^k \lambda_i (1-\lambda_i)}},
$$

$$
(1-\lambda_{k'+1}')^{-1} c_{k'+1,k'}^K \leq \frac{2}{\sqrt{\pi \sum_{i=K+1}^{k'} \lambda_i' (1-\lambda_i')}}.
$$

Now, we are ready to prove Theorem 8.

*Proof of Theorem 8.* First, by Lemma 6, we have

$$
TV^k - V^k - g^\star
$$

$$
= (1-\lambda_{k+1})^{-1}(V^{k+1} - V^k) - g^\star
$$

$$
\leq (1-\lambda_{k+1})^{-1}(c_{k+1,k}(S_1^{k+1,k} - S_2^{k+1,k})(V^0 - h^\star) + (1-\lambda_{k+1})g^\star) - g^\star
$$

$$
= (1-\lambda_{k+1})^{-1}(c_{k+1,k}(S_1^{k+1,k} - S_2^{k+1,k})(V^0 - h^\star).
$$

Similarly, by Lemma 8, we have

$$
\begin{aligned}
& TV^k - V^k - g^\star \\
&= (1 - \lambda_{k+1})^{-1}(V^{k+1} - V^k) - g^\star \\
&\geq (1 - \lambda_{k+1})^{-1} c_{k+1,k}^K (S_{1'}^{k+1,k} - S_{2'}^{k+1,k})(V^0 - h^\star).
\end{aligned}
$$

Thus, this two inequality implies that

$$
\left\| TV^k - V^k - g^\star \right\|_\infty \leq \frac{\left\| V^0 - h^\star \right\|_\infty}{\sqrt{\sum_{i=K+1}^k \lambda_i (1 - \lambda_i)}}
$$

by Fact 5 and $\lambda_k = \lambda_k'$ for $K + 1 \leq k$. Finally, by applying the Proposition 6, we conclude proof. $\qquad\square$

### F.3 PROOF OF THEOREM 1

Let $S$ be set of all deterministic policies and $\epsilon = \inf_{\pi \in S/\{\pi \,|\, \mathcal{P}^\pi g^\star = g^\star\}} \left\| \mathcal{P}^\pi g^\star - g^\star \right\|_\infty$ (note that if $S/\{\pi \,|\, \mathcal{P}^\pi g^\star = g^\star\} = \emptyset$ , $\epsilon = \infty$). By definition of Bellman optimality operator, there exist deterministic policy $\pi_k$ such that. By definition of Bellman optimality operator, there exist deterministic $\pi$ such that

$$
V^{k+1} = \frac{1}{2} V^k + \frac{1}{2} \mathcal{P}^\pi V^k + \frac{1}{2} r^\pi.
$$

for all $k$. By simple calculation, this is equivalent to

$$
-\frac{r^\pi}{\frac{k}{2}} + \frac{2V^0}{k} - \frac{2\mathcal{P}^\pi V^0}{k} = \mathcal{P}^\pi \left( \frac{V^k - V^0}{\frac{k}{2}} \right) - 2\frac{k+1}{k} \left( \frac{V^{k+1} - V^0}{\frac{k+1}{2}} \right) + \left( \frac{V^k - V^0}{\frac{k}{2}} \right)
$$

Let $\frac{V^k - V^0}{k/2} = g^\star + \epsilon_k$. By Theorem 7 with $\lambda_k = \frac{1}{2}$, we have

$$
\left\| \frac{V^k - V^0}{k/2} - g^\star \right\|_\infty \leq \frac{\left\| V^0 - h^\star \right\|_\infty}{k/4},
$$

and this implies

$$
\left\| \epsilon_k \right\|_\infty \leq \frac{\left\| V^0 - h^\star \right\|_\infty}{k/4}.
$$

Then, we have

$$
\begin{aligned}
& \mathcal{P}^\pi \left( \frac{V^k - V^0}{\frac{k}{2}} \right) - 2\frac{k+1}{k} \left( \frac{V^{k+1} - V^0}{\frac{k+1}{2}} \right) + \left( \frac{V^k - V^0}{\frac{k}{2}} \right) \\
&= \mathcal{P}^\pi (g^\star + \epsilon_k) - \frac{2(k+1)}{k}(g^\star + \epsilon_{k+1}) + (g^\star + \epsilon_k) \\
&= \mathcal{P}^\pi g^\star - g^\star + \mathcal{P}^\pi \epsilon_k - \frac{2}{k} g^\star - \frac{2(k+1)}{k} \epsilon_{k+1} + \epsilon_k.
\end{aligned}
$$

This implies

$$
\mathcal{P}^\pi g^\star - g^\star = -\frac{r^\pi}{\frac{k}{2}} + \frac{2V^0}{k} - \frac{2\mathcal{P}^\pi V^0}{k} - \mathcal{P}^\pi \epsilon_k + \frac{2}{k} g^\star - \frac{2(k+1)}{k} \epsilon_{k+1} - \epsilon_k.
$$

Then, if we take $\left\| \cdot \right\|_\infty$ in both sides of previous equality,

$$
\left\| \epsilon \right\|_\infty \leq \frac{1}{k} \left( 2\left\| r \right\|_\infty + 4\left\| V^0 \right\|_\infty + 16\left\| V^0 - h^\star \right\|_\infty + 2\left\| g^\star \right\|_\infty \right)
$$

Thus, if $k \geq \left( 2\left\| r \right\|_\infty + 4\left\| V^0 \right\|_\infty + 16\left\| V^0 - h^\star \right\|_\infty + 2\left\| g^\star \right\|_\infty \right) \epsilon^{-1}$, $\mathcal{P}^{\pi_k} g^\star = g^\star$.

Thus, if we set $K = \left( 2\left\| r \right\|_\infty + 4\left\| V^0 \right\|_\infty + 16\left\| V^0 - h^\star \right\|_\infty + 2\left\| g^\star \right\|_\infty \right) \epsilon^{-1}$, $K$ satisfied conditions of Theorem 8. Therefore, by Theorem 8 with $\lambda_i = 1/2$ for all $i$, we obtain desired rate of Bellman and policy errors.

# G   OMITTED PROOFS OF SECTION 4 AND C

In this section, we present omitted proofs convergence theorems of Anc-VI. We prove Theorem 9, 10, and 2 in turn.

## G.1   PROOF OF THEOREM 9

Define $\lambda_0 = 1$ as coefficient of Anc-VI for computational conciseness.

First, we prove the following lemma by induction.

**Lemma 9.** *For the iterates $\{V^k\}_{k=0,1,\dots}$ of Anc-VI,*

$$V^k = \sum_{i=0}^{k} (\Pi_{j=i+1}^{k}(1-\lambda_j))\lambda_i \left(\Pi_{l=k-1}^{i}\mathcal{P}^{\pi_l}\right) V^0 + \sum_{i=0}^{k-1} (\Pi_{j=i+1}^{k}(1-\lambda_j)) \left(\Pi_{l=k-1}^{i+1}\mathcal{P}^{\pi_l}\right) r^{\pi_i},$$

*Proof.* If $k = 0$, $V^0 = V^0$.

By induction,

$$
\begin{aligned}
V^{k+1} &= (1-\lambda_{k+1})TV^k + \lambda_{k+1}V^0 \\
&= (1-\lambda_{k+1})\left(\mathcal{P}^{\pi_k}\left(\sum_{i=0}^{k}(\Pi_{j=i+1}^{k}(1-\lambda_j))\lambda_i\left(\Pi_{l=k-1}^{i}\mathcal{P}^{\pi_l}\right)V^0\right.\right. \\
&\quad \left.\left. + \sum_{i=0}^{k-1}\left(\Pi_{j=i+1}^{k}(1-\lambda_j)\right)\left(\Pi_{l=k-1}^{i+1}\mathcal{P}^{\pi_l}\right)r^{\pi_i}\right) + r^{\pi_k}\right) + \lambda_{k+1}V^0 \\
&= \sum_{i=0}^{k+1}(\Pi_{j=i+1}^{k+1}(1-\lambda_j))\lambda_i\left(\Pi_{l=k}^{i}\mathcal{P}^{\pi_l}\right)V^0 + \sum_{i=0}^{k}\Pi_{j=i+1}^{k+1}(1-\lambda_j)\left(\Pi_{l=k}^{i+1}\mathcal{P}^{\pi_l}\right)r^{\pi_i}.
\end{aligned}
$$

$\square$

Now, we prove following key lemma.

**Lemma 10.** *For the iterates $\{V^k\}_{k=0,1,\dots}$ of Anc-VI,*

$$V^k - \sum_{i=0}^{k-1}\Pi_{j=i+1}^{k}(1-\lambda_j)g^\star \le h^\star + \sum_{i=0}^{k}(\Pi_{j=i+1}^{k}(1-\lambda_j))\lambda_i\left(\Pi_{l=k-1}^{i}\mathcal{P}^{\pi_l}\right)(V^0 - h^\star),$$

$$h^\star + \sum_{i=0}^{k}(\Pi_{j=i+1}^{k}(1-\lambda_j))\lambda_i\left(\Pi_{l=k-1}^{i}\mathcal{P}^{\pi_\star}\right)(V^0 - h^\star) \le V^k - \sum_{i=0}^{k-1}\Pi_{j=i+1}^{k}(1-\lambda_j)g^\star.$$

*Proof.* For the first inequality, we have

$$
\begin{aligned}
V^k &= \sum_{i=0}^{k}(\Pi_{j=i+1}^{k}(1-\lambda_j))\lambda_i\left(\Pi_{l=k-1}^{i}\mathcal{P}^{\pi_l}\right)V^0 + \sum_{i=0}^{k-1}\left(\Pi_{j=i+1}^{k}(1-\lambda_j)\right)\left(\Pi_{l=k-1}^{i+1}\mathcal{P}^{\pi_l}\right)r^{\pi_i} \\
&\le \sum_{i=0}^{k}(\Pi_{j=i+1}^{k}(1-\lambda_j))\lambda_i\left(\Pi_{l=k-1}^{i}\mathcal{P}^{\pi_l}\right)V^0 \\
&\quad + \sum_{i=0}^{k-1}\left(\Pi_{j=i+1}^{k}(1-\lambda_j)\right)\left(\Pi_{l=k-1}^{i+1}\mathcal{P}^{\pi_l}\right)(g^\star + (I - \mathcal{P}^{\pi_i})h^\star) \\
&\le \sum_{i=0}^{k-1}\Pi_{j=i+1}^{k}(1-\lambda_j)g^\star + h^\star + \sum_{i=0}^{k}(\Pi_{j=i+1}^{k}(1-\lambda_j))\lambda_i\left(\Pi_{l=k-1}^{i}\mathcal{P}^{\pi_l}\right)(V^0 - h^\star)
\end{aligned}
$$

where first equality follows from Lemma 9, first inequality comes from second Bellman equation, and second inequality is from first Bellman equation and telescoping-sum argument.

We now prove second inequality.

$$V^k \geq \sum_{i=0}^{k} (\Pi_{j=i+1}^{k}(1-\lambda_j))\lambda_i \left(\Pi_{l=k-1}^{i}\mathcal{P}^{\pi_\star}\right) V^0 + \sum_{i=0}^{k-1} \Pi_{j=i+1}^{k}(1-\lambda_j)\left(\Pi_{l=k-1}^{i+1}\mathcal{P}^{\pi_\star}\right) r^{\pi_i}$$

$$= \sum_{i=0}^{k} (\Pi_{j=i+1}^{k}(1-\lambda_j))\lambda_i \left(\Pi_{l=k-1}^{i}\mathcal{P}^{\pi_\star}\right) V^0$$

$$+ \sum_{i=0}^{k-1} \Pi_{j=i+1}^{k}(1-\lambda_j)\left(\Pi_{l=k-1}^{i+1}\mathcal{P}^{\pi_\star}\right)(g^\star + (I - \mathcal{P}^{\pi_\star})h^\star)$$

$$= \sum_{i=0}^{k-1} \Pi_{j=i+1}^{k}(1-\lambda_j)g^\star + h^\star + \sum_{i=0}^{k}(\Pi_{j=i+1}^{k}(1-\lambda_j))\lambda_i \left(\Pi_{l=k-1}^{i}\mathcal{P}^{\pi_\star}\right)(V^0 - h^\star)$$

where first inequality follows from the Lemma 3 and fact that $\{\pi_l\}_{l=0,1,\dots,k}$ are greedy policies, first equality comes from second Bellman equation, and second equality is from first Bellman equation. $\qquad\square$

We now prove Theorem 9.

*Proof of Theorem 9 .* By Lemma 10, we have

$$V^k - V^0 - \sum_{i=0}^{k-1} \Pi_{j=i+1}^{k}(1-\lambda_j)g^\star$$

$$\leq (1-\lambda_k)(h^\star - V^0) + \sum_{i=0}^{k-1}(\Pi_{j=i+1}^{k}(1-\lambda_j))\lambda_i \left(\Pi_{l=k-1}^{i}\mathcal{P}^{\pi_l}\right)(V^0 - h^\star),$$

$$V^k - V^0 - \sum_{i=0}^{k-1} \Pi_{j=i+1}^{k}(1-\lambda_j)g^\star$$

$$\geq (1-\lambda_k)(h^\star - V^0) + \sum_{i=0}^{k-1}(\Pi_{j=i+1}^{k}(1-\lambda_j))\lambda_i \left(\Pi_{l=k-1}^{i}\mathcal{P}^{\pi_\star}\right)(V^0 - h^\star).$$

If we take $\|\cdot\|_\infty$ right side of first and second inequality, we have

$$\left\|(1-\lambda_k)(h^\star - V^0) + \sum_{i=0}^{k-1}(\Pi_{j=i+1}^{k}(1-\lambda_j))\lambda_i \left(\Pi_{l=k-1}^{i}\mathcal{P}^{\pi_l}\right)(V^0 - h^\star)\right\|_\infty$$

$$\leq 2(1-\lambda_k)\left\|V^0 - h^\star\right\|_\infty,$$

$$\left\|(1-\lambda_k)(h^\star - V^0) + \sum_{i=0}^{k-1}(\Pi_{j=i+1}^{k}(1-\lambda_j))\lambda_i \left(\Pi_{l=k-1}^{i}\mathcal{P}^{\pi_\star}\right)(V^0 - h^\star)\right\|_\infty$$

$$\leq 2(1-\lambda_k)\left\|V^0 - h^\star\right\|_\infty,$$

and this implies

$$\left\|\frac{V^k - V^0}{\sum_{i=1}^{k}\Pi_{j=i}^{k}(1-\lambda_j)} - g^\star\right\|_\infty \leq \frac{2(1-\lambda_k)}{\sum_{i=1}^{k}\Pi_{j=i}^{k}(1-\lambda_j)}\left\|V^0 - h^\star\right\|_\infty.$$

$\qquad\square$

## G.2 PROOF OF THEOREM 10

Following lemma will be used in proof in later proof.

**Lemma 11.** *For the iterates $\{V^k\}_{k=0,1,...}$ of Anc-VI,*

$$TV^k - V^0 \leq \sum_{i=1}^{k+1} \Pi_{j=i}^k (1-\lambda_j) g^\star + \left( \sum_{i=0}^k (\Pi_{j=i+1}^k (1-\lambda_j)) \lambda_i \left( \Pi_{l=k}^i \mathcal{P}^{\pi_l} \right) - I \right) (V^0 - h^\star)$$

$$TV^k - V^0 \geq \sum_{i=1}^{k+1} \Pi_{j=i}^k (1-\lambda_j) g^\star + \left( \sum_{i=0}^k (\Pi_{j=i+1}^k (1-\lambda_j)) \lambda_i \left( \Pi_{l=k}^i \mathcal{P}^{\pi_\star} \right) - I \right) (V^0 - h^\star)$$

*Proof.* For the first inequality, We have

$$TV^k$$

$$\leq \mathcal{P}^{\pi_k} \left( \sum_{i=0}^{k-1} \Pi_{j=i+1}^k (1-\lambda_j) g^\star + h^\star + \sum_{i=0}^k (\Pi_{j=i+1}^k (1-\lambda_j)) \lambda_i \left( \Pi_{l=k-1}^i \mathcal{P}^{\pi_l} \right) (V^0 - h^\star) \right) + r^{\pi_k}$$

$$\leq \sum_{i=1}^k \Pi_{j=i}^k (1-\lambda_j) g^\star + \left( \sum_{i=0}^k (\Pi_{j=i+1}^k (1-\lambda_j)) \lambda_i \left( \Pi_{l=k}^i \mathcal{P}^{\pi_l} \right) (V^0 - h^\star) \right)$$
$$+ \mathcal{P}^{\pi_k} h^\star + g^\star + (I - \mathcal{P}^{\pi_k}) h^\star$$

$$= \sum_{i=1}^{k+1} \Pi_{j=i}^k (1-\lambda_j) g^\star + \sum_{i=0}^k (\Pi_{j=i+1}^k (1-\lambda_j)) \lambda_i \left( \Pi_{l=k}^i \mathcal{P}^{\pi_l} \right) (V^0 - h^\star) + h^\star,$$

where first inequality is from Lemma 10 and second inequality comes from Bellman equations.

Now, we prove the second inequality.

$$TV^k$$

$$\geq \mathcal{P}^{\pi_\star} \left( \sum_{i=0}^{k-1} \Pi_{j=i+1}^k (1-\lambda_j) g^\star + h^\star + \sum_{i=0}^k (\Pi_{j=i+1}^k (1-\lambda_j)) \lambda_i \left( \Pi_{l=k-1}^i \mathcal{P}^{\pi_\star} \right) (V^0 - h^\star) \right) + r^{\pi_\star}$$

$$= \sum_{i=1}^k \Pi_{j=i}^k (1-\lambda_j) g^\star + \sum_{i=0}^k (\Pi_{j=i+1}^k (1-\lambda_j)) \lambda_i \left( \Pi_{l=k}^i \mathcal{P}^{\pi_\star} \right) (V^0 - h^\star)$$
$$+ \mathcal{P}^{\pi_\star} h^\star + g^\star + (I - \mathcal{P}^{\pi_\star}) h^\star$$

$$= \sum_{i=1}^{k+1} \Pi_{j=i}^k (1-\lambda_j) g^\star + \sum_{i=0}^k (\Pi_{j=i+1}^k (1-\lambda_j)) \lambda_i \left( \Pi_{l=k}^i \mathcal{P}^{\pi_\star} \right) (V^0 - h^\star) + h^\star$$

where first inequality is from Lemma 10 and the fact that $\pi_k$ is greedy policy and first equality comes from Bellman equations. □

We now prove one of key lemmas.

**Lemma 12.** *For the iterates $\{V^k\}_{k=1,2,...}$ of Anc-VI,*

$$V^k - V^{k-1} \leq \left( 1 - \sum_{i=1}^k \lambda_k \Pi_{j=i}^{k-1} (1-\lambda_j) \right) g^\star + \left( 1 - \sum_{i=1}^k \lambda_i \Pi_{j=i}^{k-1} (1-\lambda_j) \right) (S_1^k - S_2^k)(V^0 - h^\star)$$

*where $S_1^k, S_2^k$ are stochastic matrices.*

*Proof.* We use induction. If $k=1$,

$$V^1 - V^0 = (1-\lambda_1) \mathcal{P}^{\pi_0} V^0 + (1-\lambda_1) r^{\pi_0} - (1-\lambda_1) V^0$$
$$\leq (1-\lambda_1) g^\star + (1-\lambda_1)(\mathcal{P}^{\pi_0} - I)(V^0 - h^\star)$$

where inequality follows from second Bellman equation.

By induction,

$$V^{k+1} - V^k$$

$$= \lambda_{k+1}V^0 + (1-\lambda_{k+1})TV^k - \lambda_k V^0 - (1-\lambda_k)TV^{k-1}$$

$$= (\lambda_k - \lambda_{k+1})(TV^k - V^0) + (1-\lambda_k)(TV^k - TV^{k-1})$$

$$\leq (\lambda_k - \lambda_{k+1})(TV^k - V^0) + (1-\lambda_k)\mathcal{P}^{\pi_k}(V^k - V^{k-1})$$

$$\leq (\lambda_k - \lambda_{k+1})\left(\sum_{i=1}^{k+1}\Pi_{j=i}^k(1-\lambda_j)g^\star + \left(\sum_{i=0}^k(\Pi_{j=i+1}^k(1-\lambda_j))\lambda_i\left(\Pi_{l=k}^i\mathcal{P}^{\pi_l}\right) - I\right)(V^0 - h^\star)\right)$$

$$+ (1-\lambda_k)\mathcal{P}^{\pi_k}\left(\left(1 - \sum_{i=1}^k\lambda_k\Pi_{j=i}^{k-1}(1-\lambda_j)\right)g^\star\right.$$

$$+ \left(1 - \sum_{i=1}^k\lambda_i\Pi_{j=i}^{k-1}(1-\lambda_j)\right)(S_1^k - S_2^k)(V^0 - h^\star)\Big)$$

$$\leq \left(\lambda_k + \sum_{i=1}^k\lambda_k\Pi_{j=i}^k(1-\lambda_j)\right)g^\star - \sum_{i=1}^{k+1}\lambda_{k+1}\Pi_{j=i}^k(1-\lambda_j)g^\star$$

$$+ \left(1 - \lambda_k - \sum_{i=1}^k\lambda_k\Pi_{j=i}^k(1-\lambda_j)\right)g^\star + \left((\lambda_k - \lambda_{k+1})\sum_{i=0}^k(\Pi_{j=i+1}^k(1-\lambda_j))\lambda_i\left(\Pi_{l=k}^i\mathcal{P}^{\pi_l}\right)\right.$$

$$+ \left(1 - \lambda_k - \sum_{i=1}^k\lambda_i\Pi_{j=i}^k(1-\lambda_j)\right)\mathcal{P}^{\pi_k}S_1^k\Big)(V^0 - h^\star)$$

$$- \left((\lambda_k - \lambda_{k+1})I + \left(1 - \lambda_k - \sum_{i=1}^k\Pi_{j=i}^k(1-\lambda_j)\lambda_i\right)\mathcal{P}^{\pi_k}S_2^k\right)(V^0 - h^\star)$$

$$= \left(1 - \sum_{i=1}^{k+1}\lambda_{k+1}\Pi_{j=i}^k(1-\lambda_j)\right)g^\star + \left(1 - \sum_{i=1}^{k+1}\lambda_i\Pi_{j=i}^k(1-\lambda_j)\right)(S_1^{k+1} - S_2^{k+1})(V^0 - h^\star)$$

where first inequality comes from the fact that $\pi_k, \pi_{k-1}$ are greedy policies, second inequality follows from induction and Lemma 11, last inequality is from the second Bellman equation, and

$$S_1^{k+1} = \left(1 - \sum_{i=1}^{k+1}\lambda_i\Pi_{j=i}^k(1-\lambda_j)\right)^{-1}\left((\lambda_k - \lambda_{k+1})\sum_{i=0}^k(\Pi_{j=i+1}^k(1-\lambda_j))\lambda_i\left(\Pi_{l=k}^i\mathcal{P}^{\pi_l}\right)\right.$$

$$+ \left(1 - \lambda_k - \sum_{i=1}^k\Pi_{j=i}^k(1-\lambda_j)\lambda_i\right)\mathcal{P}^{\pi_k}S^k\Big),$$

$$S_2^{k+1} = \left(1 - \sum_{i=1}^{k+1}\lambda_i\Pi_{j=i}^k(1-\lambda_j)\right)^{-1}\left((\lambda_k - \lambda_{k+1})I + \left(1 - \lambda_k - \sum_{i=1}^k\Pi_{j=i}^k(1-\lambda_j)\lambda_i\right)\mathcal{P}^{\pi_k}S_2^k\right).$$

Note that condition of leading coefficients positive. $\square$

To obtain lower bound of $V^k - V^{k-1}$, we need more sophisticated consideration, and following lemma is necessary for later argument.

**Lemma 13.** *Let $\{V^k\}_{k=0,1,2,\dots}$ be the iterates of Anc-VI. Let $\lambda_k \leq \lambda_{k-1}$ for $1 \leq k$ and $\lim \lambda_k = 0$. Let $E = \{\pi : \mathcal{P}^\pi g^\star = g^\star\}$. Then there exist $K$ such that if $K \leq k$,*

$$TV^K = \max_{\pi \in E} T^\pi V^K.$$

*Proof.* Suppose $\pi$ is infinitely often repeated deterministic policy among $\{\pi_k\}_{k=0,1,2,\dots}$. Then there exist increasing sequence $k_n$ such that $\pi_{k_n} = \pi$. Then, since $V^{k_n+1} = \lambda_{k_n+1}V^0 + (1-\lambda_{k_n+1})TV^{k_n}$,

we have

$$\frac{V^{k_n+1}}{\sum_{i=1}^{k_n+1}\Pi_{j=i}^{k_n}(1-\lambda_j)} = \lambda_{n_k+1}\frac{V^0}{\sum_{i=1}^{k_K+1}\Pi_{j=i}^{k_n}(1-\lambda_j)} + (1-\lambda_{n_k+1})\mathcal{P}^{\pi'}\frac{V^{k_n}}{\sum_{i=1}^{k_n+1}\Pi_{j=i}^{k_n}(1-\lambda_j)}$$
$$+ (1-\lambda_{n_k+1})\frac{r^\pi}{\sum_{i=1}^{k_n+1}\Pi_{j=i}^{k_n}(1-\lambda_j)}.$$

If $k_n \to \infty$, $\lim \lambda_k = 0$ implies $\lim_{k\to\infty}\sum_{i=0}^k \Pi_{j=i}^k(1-\lambda_j) = \infty$ by Lemma 14. Then, by Theorem 7, we have $g^\star = \mathcal{P}^\pi g^\star$, and this implies $\pi \in E$. By finiteness of action and state space, number of infinitely repeated policy $\pi$ is also finite. Therefore there exist $K$ such that $TV^k = \max_{\pi \in E} T^\pi V^k$ for $K \leq k$. $\qquad\square$

**Lemma 14.** *If* $\lim \lambda_k = 0$, *then* $\lim_{k\to\infty}\sum_{i=0}^k \Pi_{j=i}^k(1-\lambda_j)$.

*Proof.* By condition, for any $\epsilon > 0$, there exist $K_\epsilon$ such that $1 - \lambda_k > 1 - \epsilon$ if $K_\epsilon \leq k$. Hence, $\liminf_{k\to\infty}\sum_{i=0}^k \Pi_{j=i}^k(1-\lambda_j) \geq 1/\epsilon - 1$. This concludes lemma. $\qquad\square$

The following lemma will be used in the proof of key lemma.

**Lemma 15.** *Let* $\{V^k\}_{k=1,2,\dots}$ *be the iterates of Anc-VI. For* $k \leq K + 1$,

$$V^k - V^{k-1} \geq \left(1 - \sum_{i=1}^k \lambda_{k-1}\Pi_{j=i}^{k-1}(1-\lambda_j)\right)S_{3'}^k g^\star + (\lambda_{k-1} - \lambda_k)\sum_{i=1}^k \Pi_{j=i}^{k-1}(1-\lambda_j)g^\star$$
$$+ \left(1 - \sum_{i=1}^k \lambda_i\Pi_{j=i}^{k-1}(1-\lambda_j)\right)(S_{1'}^k - S_{2'}^k)(V^0 - h^\star),$$

*where* $S_{1'}^k, S_{2'}^k, S_{3'}^k$ *are stochastic matrices.*

*Proof.* We use induction. If $k = 1$, $V^1 - V^0 = (1-\lambda_1)(TV^0 - V^0) \geq (1-\lambda_1)g^\star + (1-\lambda_1)(\mathcal{P}^{\pi_\star} - I)(V^0 - h^\star)$.

By induction,

$$V^{k+1} - V^k$$
$$\geq (\lambda_k - \lambda_{k+1})(TV^k - V^0) + (1-\lambda_k)\mathcal{P}^{\pi_{k-1}}(V^k - V^{k-1})$$
$$\geq (\lambda_k - \lambda_{k+1})\left(\sum_{i=1}^{k+1}\Pi_{j=i}^k(1-\lambda_j)g^\star + \left(\sum_{i=0}^k(\Pi_{j=i+1}^k(1-\lambda_j))\lambda_i\left(\Pi_{l=k}^i \mathcal{P}^{\pi_\star}\right) - I\right)(V^0 - h^\star)\right)$$
$$+ (1-\lambda_k)\mathcal{P}^{\pi_{k-1}}\left(\left(1 - \sum_{i=1}^k \lambda_{k-1}\Pi_{j=i}^{k-1}(1-\lambda_j)\right)S_{3'}^k g^\star + (\lambda_{k-1} - \lambda_k)\sum_{i=1}^k \Pi_{j=i}^{k-1}(1-\lambda_j)g^\star\right.$$
$$+ \left.\left(1 - \sum_{i=1}^k \lambda_i\Pi_{j=i}^{k-1}(1-\lambda_j)\right)(S_{1'}^k - S_{2'}^k)(V^0 - h^\star)\right)$$

$$= (\lambda_k - \lambda_{k+1}) \left( \sum_{i=1}^{k+1} \Pi_{j=i}^k (1-\lambda_j) \right) g^\star + \left( 1 - \lambda_k - \sum_{i=1}^{k} \lambda_{k-1} \Pi_{j=i}^k (1-\lambda_j) \right) \mathcal{P}^{\pi_{k-1}} S_{3'}^k g^\star$$

$$+ (\lambda_{k-1} - \lambda_k) \sum_{i=1}^{k} \Pi_{j=i}^k (1-\lambda_j) \mathcal{P}^{\pi_{k-1}} g^\star + \left( (\lambda_k - \lambda_{k+1}) \sum_{i=0}^{k} (\Pi_{j=i+1}^k (1-\lambda_j)) \lambda_i \left( \Pi_{l=k}^i \mathcal{P}^{\pi_l} \right) \right)$$

$$+ \left( 1 - \lambda_k - \sum_{i=1}^{k} \lambda_i \Pi_{j=i}^k (1-\lambda_j) \right) \mathcal{P}^{\pi_{k-1}} S_{1'}^k \right) (V^0 - h^\star)$$

$$- \left( (\lambda_k - \lambda_{k+1}) I + \left( 1 - \lambda_k - \sum_{i=1}^{k} \Pi_{j=i}^k (1-\lambda_j) \lambda_i \right) \mathcal{P}^{\pi_{k-1}} S_{2'}^k \right) (V^0 - h^\star)$$

$$= (\lambda_k - \lambda_{k+1}) \left( \sum_{i=1}^{k+1} \Pi_{j=i}^k (1-\lambda_j) \right) g^\star + \left( 1 - \sum_{i=1}^{k+1} \lambda_k \Pi_{j=i}^k (1-\lambda_j) \right) S_{3'}^{k+1} g^\star$$

$$+ \left( 1 - \sum_{i=1}^{k+1} \lambda_i \Pi_{j=i}^k (1-\lambda_j) \right) (S_{1'}^{k+1} - S_{2'}^{k+1})(V^0 - h^\star)$$

where first inequality comes from the fact that $\pi_k, \pi_{k-1}$ are greedy policies, second inequality follows from induction and Lemma 11, and

$$S_{1'}^{k+1} = \left( 1 - \sum_{i=1}^{k+1} \lambda_i \Pi_{j=i}^k (1-\lambda_j) \right)^{-1} \left( (\lambda_k - \lambda_{k+1}) \sum_{i=0}^{k} (\Pi_{j=i+1}^k (1-\lambda_j)) \lambda_i \left( \Pi_{l=k}^i \mathcal{P}^{\pi_l} \right) \right.$$

$$+ \left( 1 - \lambda_k - \sum_{i=1}^{k} \Pi_{j=i}^k (1-\lambda_j) \lambda_i \right) \mathcal{P}^{\pi_{k-1}} S_{1'}^k \right)$$

$$S_{2'}^{k+1} = \left( 1 - \sum_{i=1}^{k+1} \lambda_i \Pi_{j=i}^k (1-\lambda_j) \right)^{-1} \left( (\lambda_k - \lambda_{k+1}) I + \left( 1 - \lambda_k - \sum_{i=1}^{k} \Pi_{j=i}^k (1-\lambda_j) \lambda_i \right) \mathcal{P}^{\pi_{k-1}} S_{2'}^k \right),$$

$$S_{3'}^{k+1} = \left( 1 - \sum_{i=1}^{k+1} \lambda_k \Pi_{j=i}^k (1-\lambda_j) \right)^{-1} \left( (\lambda_{k-1} - \lambda_k) \sum_{i=1}^{k} \Pi_{j=i}^k (1-\lambda_j) \mathcal{P}^{\pi_{k-1}} \right.$$

$$+ \left( 1 - \lambda_k - \sum_{i=1}^{k} \lambda_{k-1} \Pi_{j=i}^k (1-\lambda_j) \right) \mathcal{P}^{\pi_{k-1}} S_{3'}^k \right).$$

(Note that $\lambda_k - \lambda_{k+1} \geq 0$ implies $1 - \lambda_k - \sum_{i=1}^{k} \Pi_{j=i}^k (1-\lambda_j) \lambda_i \geq 0.$ )

$\square$

Now, we prove left key lemma.

**Lemma 16.** *Let $\{V^k\}_{k=0,1,2,\ldots}$ be the iterates of Anc-VI. Suppose there exist $K$ such that if $K \leq k$, $TV^k = \max_{\pi \in E} T^\pi V^k$ where $E = \{\pi : \mathcal{P}^\pi g^\star = g^\star\}$. Then, for $K+1 \leq k$, For the iterates $\{V^k\}_{k=K+1, K+2, \ldots}$ of Anc-VI,*

$$V^k - V^{k-1}$$

$$\geq \left( 1 - \sum_{i=1}^{k} \lambda_k \Pi_{j=i}^{k-1} (1-\lambda_j) \right) g^\star + \left( 1 - \sum_{i=1}^{k} (\Pi_{j=i}^{k-1} (1-\lambda_j)) \lambda_i \right) (S_{1'}^k - S_{2'}^k)(V^0 - h^\star)$$

$$+ \left( \Pi_{j=K+1}^{k-1} (1-\lambda_j) \right) \left( 1 - \sum_{i=1}^{K+1} \lambda_K \Pi_{j=i}^K (1-\lambda_j) \right) (S_{3'}^k - S_{4'}^k) g^\star$$

*where $S_{1'}^k, S_{2'}^k, S_{3'}^k, S_{4'}^k$ are stochastic matrices.*

*Proof.* We use induction. If $k = K + 1$, by Lemma 15, we have

$$V^{K+1} - V^K$$

$$\geq \left(1 - \sum_{i=1}^{K+1} \lambda_K \Pi_{j=i}^K (1 - \lambda_j)\right) S_{3'}^{K+1} g^\star + (\lambda_K - \lambda_{K+1}) \sum_{i=1}^{K+1} \Pi_{j=i}^K (1 - \lambda_j) g^\star$$

$$+ \left(1 - \sum_{i=1}^{K+1} \lambda_i \Pi_{j=i}^K (1 - \lambda_j)\right) (S_{1'}^{K+1} - S_{2'}^{K+1})(V^0 - h^\star)$$

$$= \left(1 - \sum_{i=1}^{K+1} \lambda_{K+1} \Pi_{j=i}^K (1 - \lambda_j)\right) g^\star + \left(1 - \sum_{i=1}^{K+1} \lambda_i \Pi_{j=i}^K (1 - \lambda_j)\right) (S_{1'}^{K+1} - S_{2'}^{K+1})(V^0 - h^\star)$$

$$\left(1 - \sum_{i=1}^{K+1} \lambda_K \Pi_{j=i}^K (1 - \lambda_j)\right) (S_{3'}^{K+1} - S_{4'}^{K+1}) g^\star.$$

where $S_{4'}^{K+1} = I$.

By induction, for $k \geq K + 2$,

$$V^{k+1} - V^k$$

$$= (\lambda_k - \lambda_{k+1})(TV^k - V^0) + (1 - \lambda_k)(TV^k - TV^{k-1})$$

$$\geq (\lambda_k - \lambda_{k+1})(TV^k - V^0) + (1 - \lambda_k)\mathcal{P}^{\pi_{k-1}}(V^k - V^{k-1})$$

$$\geq (\lambda_k - \lambda_{k+1})\left(\sum_{i=1}^{k+1} \Pi_{j=i}^k (1 - \lambda_j) g^\star + \left(\sum_{i=0}^{k} (\Pi_{j=i+1}^k (1 - \lambda_j))\lambda_i \left(\Pi_{l=k}^i \mathcal{P}^{\pi_\star}\right) - I\right)(V^0 - h^\star)\right)$$

$$+ (1 - \lambda_k)\mathcal{P}^{\pi_{k-1}}\left(\left(1 - \sum_{i=1}^{k} \lambda_k \Pi_{j=i}^{k-1} (1 - \lambda_j)\right) g^\star\right.$$

$$+ \left(1 - \sum_{i=1}^{k} \left(\Pi_{j=i}^{k-1} (1 - \lambda_j)\right)\lambda_i\right) (S_{1'}^k - S_{2'}^k)(V^0 - h^\star)$$

$$+ \Pi_{j=K+1}^{k-1}(1 - \lambda_j)\left(1 - \sum_{i=1}^{K+1} \lambda_K \Pi_{j=i}^K (1 - \lambda_j)\right) (S_{3'}^k - S_{4'}^k)g^\star\Bigg)$$

$$\geq (\lambda_k - \lambda_{k+1})\left(\sum_{i=1}^{k+1} \Pi_{j=i}^k (1 - \lambda_j)\right) g^\star + \left(1 - \lambda_k - \sum_{i=1}^{k} \lambda_k \Pi_{j=i}^k (1 - \lambda_j)\right) g^\star$$

$$+ \Pi_{j=K+1}^k (1 - \lambda_j)\left(1 - \sum_{i=1}^{K+1} \lambda_K \Pi_{j=i}^K (1 - \lambda_j)\right) (\mathcal{P}^{\pi_{k-1}} S_{3'}^k - \mathcal{P}^{\pi_{k-1}} S_{4'}^k)g^\star$$

$$+ \left((\lambda_k - \lambda_{k+1}) \sum_{i=0}^{k} (\Pi_{j=i+1}^k (1 - \lambda_j))\lambda_i \left(\Pi_{l=k}^i \mathcal{P}^{\pi_\star}\right)\right.$$

$$+ \left(1 - \lambda_k - \sum_{i=1}^{k} \Pi_{j=i}^k (1 - \lambda_j)\lambda_i\right) \mathcal{P}^{\pi_{k-1}} S_{1'}^k\right)(V^0 - h^\star)$$

$$- \left((\lambda_k - \lambda_{k+1})I + \left(1 - \lambda_k - \sum_{i=1}^{k} \Pi_{j=i}^k (1 - \lambda_j)\lambda_i\right) \mathcal{P}^{\pi_{k-1}} S_{2'}^k\right)(V^0 - h^\star)$$

$$= \left(1 - \sum_{i=1}^{k+1} \lambda_{k+1} \Pi_{j=i}^k (1 - \lambda_j)\right) g^\star + \left(1 - \sum_{i=1}^{k+1} \lambda_i \Pi_{j=i}^k (1 - \lambda_j)\right) (S_{1'}^{k+1} - S_{2'}^{k+1})(V^0 - h^\star)$$

$$+ \Pi_{j=K+1}^k (1 - \lambda_j)\left(1 - \sum_{i=1}^{K+1} \lambda_K \Pi_{j=i}^K (1 - \lambda_j)\right) (S_{3'}^k - S_{4'}^k)g^\star$$

where first inequality comes from the fact that $\pi_k, \pi_{k-1}$ are greedy policies, second inequality follows from Lemma 11 and induction, last equality is from first Bellman equation, and

$$S_{1'}^{k+1} = \left(1 - \sum_{i=1}^{k+1} \lambda_i \Pi_{j=i}^k (1 - \lambda_j)\right)^{-1} \left((\lambda_k - \lambda_{k+1}) \sum_{i=0}^k (\Pi_{j=i+1}^k (1 - \lambda_j)) \lambda_i \left(\Pi_{l=k}^i \mathcal{P}^{\pi_\star}\right)\right.$$

$$\left. + \left(1 - \lambda_k - \sum_{i=1}^k \Pi_{j=i}^k (1 - \lambda_j) \lambda_i\right) \mathcal{P}^{\pi_{k-1}} S_{1'}^k\right)$$

$$S_{2'}^{k+1} = \left(1 - \sum_{i=1}^{k+1} \lambda_i \Pi_{j=i}^k (1 - \lambda_j)\right)^{-1} \left((\lambda_k - \lambda_{k+1}) I + \left(1 - \lambda_k - \sum_{i=1}^k \Pi_{j=i}^k (1 - \lambda_j) \lambda_i\right) \mathcal{P}^{\pi_{k-1}} S_{2'}^k\right)$$

$$S_{3'}^{k+1} = \mathcal{P}^{\pi_{k-1}} S_{3'}^k$$

$$S_{4'}^{k+1} = \mathcal{P}^{\pi_{k-1}} S_{4'}^k.$$

$\square$

Now, we prove Theorem 10.

*Proof of Theorem 10.* First, we have

$$TV^k - V^k - g^\star$$

$$\leq \lambda_k (TV^k - V^0) + (1 - \lambda_k) \mathcal{P}^{\pi_k} (V^k - V^{k-1}) - g^\star$$

$$\leq \lambda_k \sum_{i=1}^{k+1} \Pi_{j=i}^k (1 - \lambda_j) g^\star + \lambda_k \left(\sum_{i=0}^k (\Pi_{j=i+1}^k (1 - \lambda_j)) \lambda_i \left(\Pi_{l=k}^i \mathcal{P}^{\pi_l}\right) - I\right) (V^0 - h^\star)$$

$$+ (1 - \lambda_k) \mathcal{P}^{\pi_k} \left(\left(1 - \sum_{i=1}^k \lambda_k \Pi_{j=i}^{k-1} (1 - \lambda_j)\right) g^\star\right.$$

$$\left. + \left(1 - \sum_{i=1}^k \lambda_i \Pi_{j=i}^{k-1} (1 - \lambda_j)\right) (S_1^k - S_2^k)(V^0 - h^\star)\right) - g^\star$$

$$= \left(\lambda_k \left(\sum_{i=0}^k (\Pi_{j=i+1}^k (1 - \lambda_j)) \lambda_i\right) \Pi_{l=k}^i \mathcal{P}^{\pi_l} + \left(1 - \lambda_k - \sum_{i=1}^k \lambda_i \Pi_{j=i}^k (1 - \lambda_j)\right) \mathcal{P}^{\pi_k} S_1^k\right) (V^0 - h^\star)$$

$$- \left(\lambda_k I + \left(1 - \lambda_k - \sum_{i=1}^k \lambda_i \Pi_{j=i}^k (1 - \lambda_j)\right) \mathcal{P}^{\pi_k} S_2^k\right) (V^0 - h^\star)$$

where first inequality comes from the fact that $\pi_k, \pi_{k-1}$ are greedy policies, second inequality follows from induction and Lemma 11 and 12, and last equality is from the second Bellman equation.

Similarly,

$$TV^k - V^k - g^\star$$

$$\geq \lambda_k (TV^k - V^0) + (1 - \lambda_k) \mathcal{P}^{\pi_{k-1}} (V^k - V^{k-1}) - g^\star$$

$$\geq \lambda_k \sum_{i=1}^{k+1} \Pi_{j=i}^k (1 - \lambda_j) g^\star + \lambda_k \left(\sum_{i=0}^k (\Pi_{j=i+1}^k (1 - \lambda_j)) \lambda_i \left(\Pi_{l=k}^i \mathcal{P}^{\pi_\star}\right) - I\right) (V^0 - h^\star)$$

$$+ (1 - \lambda_k) \mathcal{P}^{\pi_{k-1}} \left(\left(1 - \sum_{i=1}^k \lambda_k \Pi_{j=i}^{k-1} (1 - \lambda_j)\right) g^\star\right.$$

$$+ \left(1 - \sum_{i=1}^k \lambda_i \Pi_{j=i}^{k-1} (1 - \lambda_j)\right) (S_{1'}^k - S_{2'}^k)(V^0 - h^\star)$$

$$\left. + \Pi_{j=K+1}^{k-1} (1 - \lambda_j) \left(1 - \sum_{i=1}^{K+1} \lambda_K \Pi_{j=i}^K (1 - \lambda_j)\right) (S_{3'}^k - S_{4'}^k) g^\star\right) - g^\star\right) - g^\star$$

$$= \left( \lambda_k \left( \sum_{i=0}^{k} (\Pi_{j=i+1}^{k}(1-\lambda_j))\lambda_i \right) \Pi_{l=k}^{i} \mathcal{P}^{\pi_\star} + \left( 1 - \lambda_k - \sum_{i=1}^{k} \lambda_i \Pi_{j=i}^{k}(1-\lambda_j) \right) \mathcal{P}^{\pi_{k-1}} S_{1'}^{k} \right) (V^0 - h^\star)$$

$$- \left( \lambda_k I + \left( 1 - \lambda_k - \sum_{i=1}^{k} \lambda_i \Pi_{j=i}^{k}(1-\lambda_j) \right) \mathcal{P}^{\pi_{k-1}} S_{2'}^{k} \right) (V^0 - h^\star)$$

$$+ \Pi_{j=K+1}^{k}(1-\lambda_j) \left( 1 - \sum_{i=1}^{K+1} \lambda_K \Pi_{j=i}^{K}(1-\lambda_j) \right) (\mathcal{P}^{\pi_{k-1}} S_{3'}^{k} - \mathcal{P}^{\pi_{k-1}} S_{4'}^{k})g^\star \right)$$

where first inequality comes from the fact that $\pi_k, \pi_{k-1}$ are greedy policies, second inequality follows from Lemma 11 and 16, and last equality is from the second Bellman equation.

If we take $\|\cdot\|_\infty$ right side of first and second inequality, we have

$$2 \left( 1 - \sum_{i=1}^{k} \lambda_i \Pi_{j=i}^{k}(1-\lambda_j) \right) \left\| V^0 - h^\star \right\|_\infty,$$

$$2 \left( 1 - \sum_{i=1}^{k} \lambda_i \Pi_{j=i}^{k}(1-\lambda_j) \right) \left\| V^0 - h^\star \right\|_\infty + 2\Pi_{j=K+1}^{k}(1-\lambda_j) \left( 1 - \sum_{i=1}^{K+1} \lambda_K \Pi_{j=i}^{K}(1-\lambda_j) \right) \|g^\star\|_\infty,$$

respectively. Therefore, we get

$$\left\| TV^k - V^k - g^\star \right\|_\infty \le 2 \left( 1 - \sum_{i=1}^{k} \lambda_i \Pi_{j=i}^{k}(1-\lambda_j) \right) \left\| V^0 - h^\star \right\|_\infty + 2\Pi_{j=K}^{k}(1-\lambda_j) \|g^\star\|_\infty.$$

since $\Pi_{j=K+1}^{k}(1-\lambda_j) \left( 1 - \sum_{i=1}^{K+1} \lambda_K \Pi_{j=i}^{K}(1-\lambda_j) \right) \le \Pi_{j=K}^{k}(1-\lambda_j)$. Finally, by applying the Proposition 6, we conclude proof. $\qquad\square$

### G.3 Proof of Theorem 2

Let $S$ be set of all deterministic policies and $\epsilon = \inf_{\pi \in S/\{\pi \mid \mathcal{P}^\pi g^\star = g^\star\}} \|\mathcal{P}^\pi g^\star - g^\star\|_\infty$ (note that if $S/\{\pi \mid \mathcal{P}^\pi g^\star = g^\star\} = \emptyset$, $\epsilon = \infty$). By definition of Bellman optimality operator, there exist deterministic policy $\pi_k$ such that

$$V^{k+1} = \frac{2}{k+3}V^0 + \frac{k+1}{k+3}\mathcal{P}^{\pi_k}V^k + \frac{k+1}{k+3}r^{\pi_k}.$$

for all $k$. By simple calculation, this is equivalent to

$$-\frac{k+1}{k+3}\frac{r^{\pi_k}}{k/3} = \frac{k+1}{k+3}\mathcal{P}^{\pi_k}\left(\frac{V^k - V^0}{\frac{k}{3}}\right) - \frac{k+1}{k}\left(\frac{V^{k+1} - V^0}{\frac{k+1}{3}}\right)$$

Let $\frac{V^k - V^0}{k/3} = g^\star + \epsilon_k$. By Theorem 9 with $\lambda_k = \frac{2}{k+2}$, we have

$$\left\| \frac{V^k - V^0}{k/3} - g^\star \right\|_\infty \le \frac{\|V^0 - h^\star\|_\infty}{k/6},$$

and this implies

$$\|\epsilon_k\|_\infty \le \frac{\|V^0 - h^\star\|_\infty}{k/6}.$$

Then, we have

$$\frac{k+1}{k+3}\mathcal{P}^{\pi_k}\left(\frac{V^k - V^0}{\frac{k}{3}}\right) - \frac{k+1}{k}\left(\frac{V^{k+1} - V^0}{\frac{k+1}{3}}\right)$$

$$= \frac{k+1}{k+3}\mathcal{P}^{\pi_k}(g^\star + \epsilon_k) - \frac{k+1}{k}(g^\star + \epsilon_{k+1})$$

$$= \mathcal{P}^{\pi_k}g^\star - g^\star - \frac{2}{k+3}\mathcal{P}^{\pi_k}g^\star - \frac{1}{k}g^\star + \frac{k+1}{k+3}\epsilon_k - \frac{k+1}{k}\epsilon_{k+1}.$$

This implies

$$\mathcal{P}^{\pi_k} g^\star - g^\star = -\frac{k+1}{k+3} \frac{r^{\pi_k}}{k/3} + \frac{2}{k+3} \mathcal{P}^{\pi_k} g^\star + \frac{1}{k} g^\star - \frac{k+1}{k+3} \epsilon_k + \frac{k+1}{k} \epsilon_{k+1}.$$

Then, if we take $\|\cdot\|_\infty$ in both sides of previous equality,

$$0 < \epsilon \le \frac{1}{k} \left( 3 \|r\|_\infty + 12 \|V^0 - h^\star\|_\infty + 3 \|g^\star\|_\infty \right)$$

Thus, if $k \ge \left( 3 \|r\|_\infty + 12 \|V^0 - h^\star\|_\infty + 3 \|g^\star\|_\infty \right) \epsilon^{-1}$, $\mathcal{P}^{\pi_k} g^\star = g^\star$.

Thus, if we set $K = \left( 3 \|r\|_\infty + 12 \|V^0 - h^\star\|_\infty + 3 \|g^\star\|_\infty \right) \epsilon^{-1}$, $K$ satisfied conditions of Theorem 8. Therefore, by Theorem 8 with $\lambda_i = \frac{2}{i+2}$ for all $i$, we have desired rate of Bellman and policy errors.

## H  OMITTED PROOFS IN SECTION 5

### H.1  PROOF OF THEOREM 3

First, we prove the case $V^0 = 0$ for $n \ge k + 2$. Consider the MDP $(\mathcal{S}, \mathcal{A}, P, r)$ such that

$$\mathcal{S} = \{s_1, \ldots, s_n\}, \ \mathcal{A} = \{a_1\}, \ P(s_i \mid s_j, a_1) = \mathbb{1}_{\{(i,j)=(n-1,1), \, j=i+1\}}, \ r(s_i, a_1) = \mathbb{1}_{\{i=1\}},$$

where $\{s_1, \ldots, s_{n-1}\}$ is closed irreducible set and $\{s_n\}$ is transient set. Thus, given MDP is unichain. Moreover, $T = \mathcal{P}^\pi U + [1, 0, \ldots, 0]^\intercal$, and since $(\mathcal{P}^\pi)^m = (\mathcal{P}^\pi)^{m+n}$ for $1 \le m \le n-1$, $g^\star = \lim_{k \to \infty} \frac{\sum_{i=0}^k (\mathcal{P}^\pi)^i}{k} r^\pi = [1/(n-1), \ldots, 1/(n-1)]^\intercal$ and $h^\star = [(n-1)/(2n-2), (n-3)/(2n-2), \ldots, -(n-3)/(2n-2), -(n-1)/(2n-2)]^\intercal$ satisfy modified Bellman equation. Therefore, $\|V^0 - h^\star\|_\infty = 1/2$, and under the span condition, we can show following lemma.

**Lemma 17.** *Let $T \colon \mathbb{R}^n \to \mathbb{R}^n$ be defined as before. Then, under span condition, $(V^i)_j = 0$ for $0 \le i \le k$, $i+1 \le j \le n$.*

*Proof.* We use induction. Case $i = 0$ is obvious. By induction, $(V^l)_j = 0$ for $0 \le l \le i-1$, $l+1 \le j \le n$. Then $(TV^l)_j = 0$ for $0 \le l \le i-1$, $l+2 \le j \le n$ and this implies that $(TV^l - V^l)_j = 0$ for $0 \le l \le i-1$, $l+2 \le j \le n$. Therefore, $(V^i)_j = 0$ for $i+1 \le j \le n$. $\square$

Thus, under the span condition, for $i \le k$, we get

$$TV^i - V^i = (1 - (V^i)_1, (V^i)_1 - (V^i)_2, \ldots, (V^i)_{i-1} - (V^i)_i, (V^i)_i, \underbrace{0, \ldots, 0}_{n-i-1})$$

and this implies that

$$(TV^i - V^i)_1 + \cdots + (TV^i - V^i)_n = 1.$$

Then

$$a_i \sum_{l=1}^n (TV^i - V^i)_l = a_i$$

for $0 \le i \le k$. If $\sum_{i=0}^k a_i = 1$, we have

$$\sum_{i=0}^k a_i \sum_{l=1}^n (TV^i - V^i)_l = 1$$

and taking the absolute value on both sides,

$$\sum_{l=1}^n \left| \sum_{i=0}^k a_i (TV^i - V^i)_l \right| \ge 1.$$

Since $(TV^i - V^i)_l = 0$ for $k + 2 \leq l$, we have

$$(k+1) \max_{1 \leq l \leq n} \left| \sum_{i=0}^{k} a_i (TV^i - V^i)_l \right| \geq 1.$$

Therefore, this implies

$$\left\| \sum_{i=0}^{k} a_i (TV^i - V^i) \right\|_\infty \geq \frac{1}{k+1}.$$

Since $g^\star = [1/(n-1), 1/(n-1), \dots, 1/(n-1)]$. we conclude

$$\left\| \sum_{i=0}^{k} a_i (TV^i - V^i) - g^\star \right\|_\infty \geq \max \left\{ \frac{1}{k+1} - \frac{1}{n-1}, \frac{1}{n-1} \right\}$$

$$\geq \frac{1}{k+1} \left\| V^0 - h^\star \right\|_\infty.$$

Now, we show that for any initial point $V^0 \in \mathbb{R}^n$, there exists an MDP which exhibits same lower bound with the case $V^0 = 0$. Denote by MDP(0) and $T_0$ the worst-case MDP and Bellman optimality operator constructed for $V^0 = 0$. Define an MDP($V^0$) $(\mathcal{S}, \mathcal{A}, P, r)$ for $V^0 \neq 0$ as

$$\mathcal{S} = \{s_1, \dots, s_n\}, \quad \mathcal{A} = \{a_1\}, \quad P(s_i \mid s_j, a_1) = \mathbb{1}_{\{(i,j)=(n-1,1),\, j=i+1\}}$$

$$r(s_i, a_1) = \left( V^0 - \mathcal{P}^\pi V^0 \right)_i + \mathbb{1}_{\{i=1\}}.$$

Then, Bellman optimality operator $T$ satisfies

$$TV = T_0(V - V^0) + V^0.$$

Let $\tilde{g}^\star$ be average reward of $T_0$ and $\tilde{h}^\star$ solution of optimlaity equation. Then, since $\lim_{k \to \infty} \frac{\sum_{i=0}^{k} (\mathcal{P}^\pi)^i}{k} (I - \mathcal{P}^\pi) = 0$, $g^\star = \tilde{g}^\star$ is average reward of $T$ and $h^\star = V^0 + \tilde{h}^\star$ is also solution of Bellman equation. Furthermore, if $\{V^i\}_{i=0}^{k}$ satisfies span condition

$$V^i \in V^0 + span\{TV^0 - V^0, TV^1 - V^1, \dots, TV^{i-1} - V^{i-1}\}, \qquad i = 1, \dots, k,$$

$\tilde{V}^i = V^i - V^0$ is a sequence satisfying

$$\tilde{V}^i \in \underbrace{\tilde{V}^0}_{=0} + span\{T_0\tilde{V}^0 - \tilde{V}^0, T_0\tilde{V}^1 - \tilde{V}^1, \dots, T_0\tilde{V}^{i-1} - \tilde{V}^{i-1}\}, \qquad i = 1, \dots, k,$$

which is the same span condition in Theorem 4 with respect to $T_0$. This is because

$$TV^i - V^i = T_0(V^i - V^0) - (V^i - V^0) = T\tilde{V}^i - \tilde{V}^i$$

for $i = 0, \dots, k$. Thus, $\{\tilde{U}^i\}_{i=0}^{k}$ is a sequence starting from 0 and satisfy the span condition for $T_0$. This implies that

$$\left\| \sum_{i=0}^{k} a_i (TV^i - V^i) - h^\star \right\|_\infty = \left\| \sum_{i=0}^{k} a_i (T\tilde{V}^i - \tilde{V}^i) - h^\star \right\|_\infty$$

$$\geq \frac{1}{k+1} \left\| \tilde{V}^0 - \tilde{h}^\star \right\|_\infty$$

$$= \frac{1}{k+1} \left\| V^0 - h^\star \right\|_\infty.$$

Hence, MDP($V^0$) is indeed our desired worst-case instance.

## H.2 Proof of Theorem 4

We now present the proof of Theorem 4.

*Proof of Theorem 4.* First, we prove the case $V^0 = 0$ for $n \geq k + 3$. Consider the MDP $(\mathcal{S}, \mathcal{A}, P, r)$ such that

$$\mathcal{S} = \{s_1, \ldots, s_n\}, \ \mathcal{A} = \{a_1\}, \ P(s_i \mid s_j, a_1) = \mathbb{1}_{\{i=j=1, \, j=i+1\leq n-1, \, i=j=n\}}, \ r(s_i, a_1) = \mathbb{1}_{\{i=2, i=n\}}.$$

where $\{s_1\}, \{s_n\}$ are closed irreducible sets and $\{s_2, \ldots, s_{n-1}\}$ is transient set. Thus, given MDP is multichain. Morevoer, $T = \mathcal{P}^\pi U + [0, 1, 0, \ldots, 0, 1]^\intercal$, and since $(\mathcal{P}^\pi)^m = (\mathcal{P}^\pi)^{k+1}$ for $m \geq k + 1$, $g^\star = \lim_{k\to\infty} \frac{\sum_{i=0}^{k}(\mathcal{P}^\pi)^i}{k} r^\pi = [0, \ldots, 0, 1]^\intercal$ and $h^\star = [-1/2, 1/2, 1/2, \ldots, 1/2, 0]^\intercal$ which satisfy Bellman equation. Thus, $\left\| V^0 - h^\star \right\|_\infty = 1/2$. Under the span condition, we can show following lemma.

**Lemma 18.** *Let $T \colon \mathbb{R}^n \to \mathbb{R}^n$ be defined as before. Then, under span condition, $\left(V^i\right)_1 = 0$ for $0 \leq i \leq k$, and $\left(V^i\right)_j = 0$ for $0 \leq i \leq k$ and $i + 2 \leq j \leq n - 1$.*

*Proof.* We use induction. Case $i = 0$ is obvious. By induction, $\left(V^l\right)_1 = 0$ for $0 \leq l \leq i - 1$. Then $\left(TV^l\right)_1 = 0$ for $0 \leq l \leq i - 1$. This implies that $\left(TV^l - V^l\right)_1 = 0$ for $0 \leq l \leq i - 1$. Hence $\left(V^i\right)_1 = 0$. Again, by induction, $\left(V^l\right)_j = 0$ for $0 \leq l \leq i-1, l+2 \leq j \leq n-1$. Then $\left(TV^l\right)_j = 0$ for $0 \leq l \leq i - 1, \, l + 3 \leq j \leq n - 1$ and this implies that $\left(TV^l - V^l\right)_j = 0$ for $0 \leq l \leq i - 1$, $l + 3 \leq j \leq n - 1$. Therefore, $\left(V^i\right)_j = 0$ for $i + 2 \leq j \leq n - 1$. $\qquad\square$

Thus, under the span condition, for $0 \leq i \leq k$, we get

$$TV^i - V^i = \left(0, 1 - \left(V^i\right)_2, \left(V^i\right)_2 - \left(V^i\right)_3, \ldots, \left(V^i\right)_i - \left(V^i\right)_{i+1}, \left(V^i\right)_{i+1}, \underbrace{0, \ldots, 0}_{n-i-3}, 1\right),$$

and this implies that

$$(TV^i - V^i - g^\star)_1 + \cdots + (TV^i - V^i - g^\star)_n = 1,$$

where $g^\star = e_n$. Then,

$$a_i \sum_{l=1}^{n} (TV^i - V^i - g^\star)_l = a_i$$

for $0 \leq i \leq k$. If $\sum_{i=0}^{k} a_i = 1$, we have

$$\sum_{i=0}^{k} a_i \sum_{l=1}^{n} (TV^i - V^i - g^\star)_l = 1$$

and taking the absolute value on both sides,

$$\sum_{l=1}^{n} \left| \sum_{i=0}^{k} a_i (TV^i - V^i - g^\star)_l \right| \geq 1.$$

Since $(TV^i - V^i - g^\star)_l = 0$ for $l = 1$ and $k + 3 \leq l$, we have

$$(k+1) \max_{1 \leq l \leq n} \left| \left( \sum_{i=0}^{k} a_i (TV^i - V^i) - g^\star \right)_l \right| \geq 1.$$

Therefore, we conclude

$$\left\| \sum_{i=0}^{k} a_i (TV^i - V^i) - g^\star \right\|_\infty \geq \frac{2}{k+1} \left\| V^0 - h^\star \right\|_\infty.$$

With the same argument in proof of Theorem 3, we can extend this result to arbitrary $V^0$. $\qquad\square$

# I OMITTED PROOFS IN SECTION 6 AND D

## I.1 PROOF OF THEOREM 11

Consider Rx-VI and $V^0 = h^0$. Let $\{V^k\}_{k=0,1,2,\ldots}$ be the iterates of Rx-VI. Then, since $T(v + c\mathbf{1}) = c\mathbf{1} + T(v)$ for arbitrary $v \in \mathbb{R}^n$ and $c \in \mathbb{R}$, $V^k = h^k + c_k\mathbf{1}$ for some $c_k \in \mathbb{R}$. This implies $TV_k - V_k = Th_k - h_k$ and by Corollary 4, we have

$$\|g^\star - g^{\pi_k}\|_\infty \le \|Th^k - h^k - g^\star\|_\infty \le \frac{2\|h^0 - h^\star\|_\infty}{\sqrt{\pi \sum_{j=1}^k \lambda_i(1-\lambda_i)}}.$$

Now, consider following iteration

$$V_g^k = \lambda_k V_g^{k-1} + (1-\lambda_k)(TV_g^{k-1} - g^\star)$$

for $1 \le k$ where $g^\star$ is average reward of $T$ and $V_g^0 = h^0$. Then, $\|V_g^k - h^\star\|_\infty \le \|V_g^{k-1} - h^\star\|_\infty$ for $1 \le k$ where $h^\star$ is solution of Bellman equation. This is because

$$\begin{aligned}
\|V_g^k - h^\star\|_\infty &= \|\lambda_k(V_g^{k-1} - h^\star) + (1-\lambda_k)(TV_g^{k-1} - g^\star - h^\star)\|_\infty \\
&\le \lambda_k\|V_g^{k-1} - h^\star\|_\infty + (1-\lambda_k)\|TV_g^{k-1} - g^\star - h^\star\|_\infty \\
&\le \lambda_k\|V_g^{k-1} - h^\star\|_\infty + (1-\lambda_k)\|V_g^{k-1} - h^\star\|_\infty \\
&\le \|V_g^{k-1} - h^\star\|_\infty
\end{aligned}$$

where second inequality is from the fact that $h^\star$ is fixed point of nonexpansive operator $T(\cdot) + g^\star$. This implies that $V_g^k$ is bounded. Then, there exist convergent subsequence $V_g^{k_n}$ which converges to some $V_g \in \mathbb{R}^d$. Since $g$ is uniform constant vector, using previous argument and Theorem 8, we have $\|TV_g^k - V_g^k - g^\star\|_\infty \le \frac{2}{\sqrt{\pi \sum_{j=1}^k \lambda_i(1-\lambda_i)}}\|V_g^0 - h^\star\|_\infty$ by condition of $\lambda_k$. This implies that $V_g$ is solution of Bellman equation, and since $\|V_g^k - V_g\|_\infty \le \|V_g^{k-1} - V_g\|_\infty$, we have $V_g^k \to V_g$.

By previous argument, there exist $c_k \in \mathbb{R}$ such that $V_g^k = h^k + c_k\mathbf{1}$ for $0 \le k$ where $c_0 = 0$. Also, we have

$$\begin{aligned}
V_g^k - h^k &= \lambda_k(V_g^{k-1} - h^{k-1}) + (1-\lambda_k)(TV_g^{k-1} - g - Th^{k-1} + f(h^{k-1})\mathbf{1}) \\
&= \lambda_k c_{k-1}\mathbf{1} + (1-\lambda_k)(c_{k-1}\mathbf{1} - g + f(h^{k-1})\mathbf{1}) \\
&= \lambda_k c_{k-1}\mathbf{1} + (1-\lambda_k)(f(V_g^{k-1})\mathbf{1} - g).
\end{aligned}$$

where last equality comes from the property of $f$. This implies $c_k = \lambda_k c_{k-1} + (1-\lambda_k)(f(V_g^{k-1}) - \hat{g}) = \sum_{j=1}^k (\Pi_{i=j+1}^k \lambda_i)(1-\lambda_j)(f(V_g^{j-1}) - \hat{g})$ where $g^\star = \hat{g}\mathbf{1}$. We now prove $c_k \to f(V_g) - \hat{g}$. Since $f$ is a continuous function and $V_g^k \to V_g$, $|f(V_g^k) - \hat{g}| \le M$ for some $0 < M$, and there exist $K$ such that $|f(V_g) - g - (f(V_g^k) - \hat{g})| < \epsilon$ for any $0 < \epsilon$ and all $K \le k$. For $K + 1 \le k$, we have

$$\begin{aligned}
|f(V_g) - \hat{g} - c_k| &= \left(\Pi_{i=1}^k \lambda_i\right)|f(V_g) - \hat{g}| + \left|\sum_{j=1}^k (\Pi_{i=j+1}^k \lambda_i)(1-\lambda_j)\left(f(V_g) - \hat{g} - (f(V_g^{j-1}) - \hat{g})\right)\right| \\
&\le \left(\Pi_{i=1}^k \lambda_i\right)|f(V_g) - \hat{g}| + \sum_{j=K+1}^k (\Pi_{i=j+1}^k \lambda_i)(1-\lambda_j)\left|f(V_g) - \hat{g} - (f(V_g^{j-1}) - \hat{g})\right| \\
&\quad + \sum_{j=1}^K (\Pi_{i=j+1}^k \lambda_i)(1-\lambda_j)\left|f(V_g) - \hat{g} - (f(V_g^{j-1}) - \hat{g})\right| \\
&\le \left(\Pi_{i=1}^k \lambda_i\right)M + \epsilon + \left(\sum_{j=1}^K (\Pi_{i=j+1}^k \lambda_i)(1-\lambda_j)\right)2M.
\end{aligned}$$

Thus, as $k \to \infty$, $c_k \to f(V_g) - \hat{g}$ if $\limsup \lambda_j < 1$. This implies $h^k$ converges to $h^\star$ solution of Bellman equations, and we have $h^\star = Th^\star - f(h)\mathbf{1}$. By uniqueness of $g^\star$, $f(h^\star)\mathbf{1} = g^\star$.

## I.2 PROOF OF THEOREM 5

If $\lambda_i = 1/2$ for all $i$, it satisfies condition of Theorem 11. Then, by Theorem 11 with $\lambda_i = 1/2$, we get the desired result.

## I.3 PROOF OF THEOREM 12

Consider Anc-VI and $V^0 = h^0$. Let $\{V^k\}_{k=0,1,2,...}$ be the iterates of Anc-VI. Then, since $T(v + c\mathbf{1}) = c\mathbf{1} + T(v)$ for arbitrary $v \in \mathbb{R}^n$ and $c \in \mathbb{R}$, $V^k = h^k + c_k\mathbf{1}$ for some $c_k \in \mathbb{R}$. This implies $TV_k - V_k = Th_k - h_k$ and by Corollary 5, we have

$$\left\| g^\star - g^{\pi_k} \right\|_\infty \le \left\| Th^k - h^k - g^\star \right\|_\infty \le 2 \left( \sum_{i=0}^k \Pi_{j=i+1}^k (1 - \lambda_j) \lambda_i^2 \right) \left\| h^0 - h^\star \right\|_\infty.$$

Consider following iteration

$$V_g^k = \lambda_k V_g^0 + (1 - \lambda_k)(TV_g^{k-1} - g)$$

for $1 \le k$ where $g^\star$ is average reward of $T$ and $V_g^0 = h^0$. Then, $\left\| V_g^k - h^\star \right\|_\infty \le \left\| V_g^0 - h^\star \right\|_\infty$. By induction, we have

$$\begin{aligned}
\left\| V_g^k - h^\star \right\|_\infty &= \left\| \lambda_k(V_g^0 - h^\star) + (1 - \lambda_k)(TV_g^{k-1} - g^\star - h^\star) \right\|_\infty \\
&\le \lambda_k \left\| V_g^0 - h^\star \right\|_\infty + (1 - \lambda_k) \left\| TV_g^{k-1} - g^\star - h^\star \right\|_\infty \\
&\le \lambda_k \left\| V_g^0 - h^\star \right\|_\infty + (1 - \lambda_k) \left\| V_g^{k-1} - h^\star \right\|_\infty \\
&\le \left\| V_g^0 - h^\star \right\|_\infty,
\end{aligned}$$

where second inequality is from the fact that $h^\star$ is fixed point of nonexpansive operator of $T(\cdot) - g^\star$ and last inequality comes from induction. This implies that $V_g^k$ is bounded.

By previous argument, there exist $c_k \in \mathbb{R}$ such that $V_g^k = h^k + c_k\mathbf{1}$ for $0 \le k$ since $g^\star$ is uniform constant vector. Also, we have

$$\begin{aligned}
V_g^k - h^k &= (1 - \lambda_k)(TV_g^{k-1} - g^\star - Th^{k-1} + f(h^{k-1})\mathbf{1}) \\
&= (1 - \lambda_k)(c_{k-1} - g^\star + f(h^{k-1})\mathbf{1}) \\
&= (1 - \lambda_k)(f(V_g^{k-1}) - g^\star).
\end{aligned}$$

where last equality comes from the property of $f$. This implies $c_k = (1 - \lambda_k)(f(V_g^{k-1}) - g^\star)$. Since $f$ is a continuous function, the boundedness of $V_g^k$ implies the boundedness of $c_k$ and this also implies boundedness of $h^k$.

Now for convergence of $h^k$, we use folloiwng fact.

**Fact 6** (Classical result, (Schweitzer & Federgruen, 1978, Remark 3))**.** *If MDP is unichain and $h$ is solution of modified Bellman equations, $h = h^\star + c\mathbf{1}$ for arbitrary $c \in \mathbb{R}$ and some fixed solution of modified Bellman equations $h^\star$.*

Suppose there exist convergent subsequence $h^{k_n}$ of $h^k$. Then, there also exist subsequence $h^{k'_n - 1}$ of $h^{k_n - 1}$ which converges to some $h$. By the previous convergence result, $h$ must be solution of modified Bellman equation, and by Fact 6, $h = h^\star + c\mathbf{1}$ for some constant $c \in \mathbb{R}$ and $h^\star$ a fixed solution of modified Bellman equation. This implies $\lambda_{k'_n - 1}(h^{k'_n - 1}) + (1 - \lambda_{k'_n - 1})(Th^{k'_n - 1} - f(h^{k'_n - 1})\mathbf{1}) \to Th^\star + f(h^\star)\mathbf{1}$. Since $Th^\star + f(h^\star)\mathbf{1}$ is fixed, $h^k$ is converge to $h$ where $h = Th - f(h)\mathbf{1}$. By uniqueness of $g^\star$, $f(h^\star)\mathbf{1} = g^\star$.

## I.4 PROOF OF THEOREM 6

If $\lambda_i = 2/(i + 2)$ for all $i$, it satisfies condition of Theorem 12. Then, by Theorem 12 with $\lambda_i = 2/(i + 2)$, we get the desired result.

