# OpenReview forum: "Optimal Non-Asymptotic Rates of Value Iteration for Average-Reward Markov Decision Processes"
_ICLR.cc/2025/Conference — ICLR 2025 Poster_

### Official Review · Reviewer_uKP2 · 2024-10-31

**Soundness:** 2
**Presentation:** 3
**Contribution:** 2
**Rating:** 6
**Confidence:** 3

**Summary:**

This paper consider the problem of learning average reward MDPs given the underlying dynamics. The major contribution is to  propose a simple but efficient iteration method (Anchored Value Iteration)  to learn a near-optimal policy. In comparison, the naive Bellman iteration method could only approximate the optimal value function, instead of a near optimal policy.

**Strengths:**

Overall I appreciate the technical effort to improve the order of $k$ by designing a proper learning rate sequence $\\{\lambda_k\\}_{k\geq 1}$.

**Weaknesses:**

The major concern is about the significance of the result. The condition number $\epsilon$ might be a problem in the worst case. In particular, I do not understand how  a $\delta-$sub-optimal state-action pair ($\delta\to 0$) forces a larger iteration number. Is there any lower bounds related to the condition number $\epsilon$?  I also wonder whether  the same problem exists in the case of discounted MDP.

Another concern is about the motivation. This work assumes access to the Bellman operator oracle, which is relatively strong in related fields.

Minor point: Line 1556 and possible other places: the index in $\Pi$ should be $\Pi_{\ell=i}^{k-1}$, not $\Pi_{\ell=k-1}^{i}$.

**Questions:**

Please see the comments above.

---

> ### Author Response · Authors · 2024-11-22
> **Official Comment by Authors**
>
> We thank the reviewer for the comments. We address the comments in the following.
>
> **Weakness**
>
> (i) $\delta$-suboptimal solution is directly induced by our non-asymptotic rates in Theorem 1 and 2. For Anc-VI, by Theorem $1$, we have
> $$\|\|g^{\star}-g^{\pi_k}\|\| \le \|\|TV^k-V^k-g^{\star}\|\| \le \frac{8}{k+1}\|\|V^0-h^{\star}\|\|+ \frac{K}{k+1} \|\|g^\star\|\|$$
> where $K=(3\|\|r\|\|+12\|\|V^0-h^{\star}\|\| +3\|\|g^{\star}\|\|)  /\epsilon$ and $k$ is number of iteration.
> Then, to obtain $\delta$-suboptimal solution, we consider the following inequality
> $$\frac{8}{k+1}\|\|V^0-h^{\star}\|\|+ \frac{K}{k+1} \|\|g^\star\|\| \le \delta.$$
> By simple calculation, we get
> $$\frac{8\|\|V^0-h^{\star}\|\|}{\delta}+\frac{ (3\|\|r\|\|+12\|\|V^0-h^{\star}\|\|+3\|\|g^{\star}\|\|) \|\|g^{\star}\|\|}{\epsilon\delta}  \le  k.$$
> Since $\delta$ is in the denominator of the first and second terms, we can easily check that $k \rightarrow \infty$ as $\delta \rightarrow 0$ (with the same argument, we can obtain  $\delta$-suboptimal solution of Rx-VI).
>
> In this work, we didn't provide lower bound of $\epsilon$, but in Corollary $2$, we show that if general (multichain) MDP satisfying $\mathcal{P}^{\pi}g^{\star}=g^{\star}$ for any policy $\pi$, then $\epsilon = \infty$ (Note that the weakly communicating MDPs satisfies this assumption as we clarified in the main text). In this case, convergence rate is independent of $\epsilon$, and morevoer, if MDP is weakly communicating MDP, this rate is optimal as it matches the lower bound established in Theorem 3.
>
>
> In the discounted case, to the best of our knowledge, it was well known that VI exhibits $O(\gamma^k)$-rate irrelevant to the structure of MDP. This rate guarantees the convergence whenever $\gamma<1$ even if $\epsilon$ is relatively small. But analyzing the convergence rate of VI by taking into account both the MDP structure and the discount factor would be an interesting future work.
>
>
>
>
>
>
>
>
> (ii) We acknowledge that this paper primarily focuses on the tabular setup, which assumes access to a Bellman operator oracle -- a relatively strong assumption compared to practical setups like generative models or episodic sampling. However, as shown in Table 1, even within this tabular setup, a non-asymptotic analysis of VI had not been successfully conducted for multichain MDPs, and the corresponding lower bound was unknown. To the best of our knowledge, our work establishes the first non-asymptotic rate of VI in multichain MDPs setup, and additionally, first demonstrates the optimality of Anc-VI by providing the lower bound and matching it to the upper bound. We believe our result contributes to the RL theory literature since non-asymptotic convergence analysis and establishing optimality through matching lower bounds are the main topics in this area [1-6].
>
>  As reviewer is concerned, we agree that considering practical algorithms in real-world scenarios is undeniably important. In this regard, we hope the reviewer sees our work as foundational groundwork for developing such practical algorithms. As VI serves as the foundational basis of practical RL algorithms such as fitted value iteration and Q-learning, we expect that Anc-VI and Rx-VI will give insight into designing new practical algorithms or improving existing ones by incorporating the anchoring and averaging mechanism. This will certainly be an interesting direction that we plan to pursue in our future work, and to clarify the motivation behind our work in this context, we have added a discussion on the significance of VI in the RL literature in prior works section.
>
> (iii) In Section E.1 of the Appendix, we denote $\Pi^k_{i=j}A_i =A_jA_{j+1}\dots A_k$ (ascending order) and $\Pi^j_{i=k}A_i =A_kA_{k-1}\dots A_j$ (descending order) where $ 0 \le j \le k $  and $A_i \in \mathbb{R}^{n\times n}$ for $j \le i \le k$, to classify multiplication order of matrix.
>
> [1] A. Sidford, M. Wang, X. Wu, L. F. Yang, Y. Ye,  Near-optimal time and sample complexities for solving discounted Markov decision process with a generative model, Neural Information Processing Systems, 2018.
>
> [2] L. Xiao, On the convergence rates of policy gradient methods, Journal of Machine Learning Research, 23(282), 1-36, 2022.
>
> [3] Y. Wu, W. Zhang, P. Xu, Q. Gu, A finite-time analysis of two time-scale actor-critic methods, Neural Information Processing Systems, 2020.
>
> [4] J. Bhandari, R. Daniel, R. Singal. A finite time analysis of temporal difference learning with linear function approximation, Conference on learning theory, 2018.
>
> [5] G. Azar, M. Munos, H. J. Kappen, Minimax PAC bounds on the sample complexity of reinforcement learning with a generative model. Machine learning (91): 325-349,  2013.
>
> [6] J. Yujia, and A. Sidford, Towards tight bounds on the sample complexity of average-reward MDPs. International Conference on Machine Learning, 2021.

---

> > ### Comment · Reviewer_uKP2 · 2024-11-26
> > **Discussion**
> >
> > Thanks for the detailed reply. My major concern is still about the role of the parameter $\epsilon$ in the convergence analysis. The computation in your reply does not address my concern. In particular, my intuition is that, adding a  sub-optimal state-action pair to the MDP **should not** worsen the convergence rate very much. I wonder if there is a concrete proof against this intuition, which might make the $1/\epsilon$ dependence necessary. At the moment, I prefer to keep the score.

---

> ### Author Response · Authors · 2024-11-29
> **Official Comment by Authors**
>
> We thank the reviewer for the response and for elaborating on the concern. To address the reviewer's point, we provide a toy example demonstrating that how adding a sub-optimal state-action pair to the MDP can significantly increase the number of iterations of VI to converge an optimal policy.
>
> **Original MDP**
>
> Consider the original MDP defined as follows:
>
> States: $\mathcal{S}=\\{s_1,s_2\\}$, Action: $\mathcal{A}=\\{a_1\\}$, Transition probability: $P(s_1  \, | \, s_1, a_1)=1, P(s_2  \, | \, s_2, a_1)=1$, Reward: $r(s_1,a_1)=\frac{1}{2}, r(s_2,a_1)=0$.
>
> This average-reward MDP has optimal policy $\pi^{\star} (s_1)=a_1, \pi^{\star} (s_2)=a_1$, optimal average reward $g^{\star} = (\frac{1}{2},0)$, and condition number $\epsilon = \infty$ . Since this MDP has only one action, VI converges to the optimal policy at once.
>
> **Modified MDP**
>
> Now, consider adding a sub-optimal action $a_2$ to this MDP and this modified MDP defined as follows:
>
> States: $\mathcal{S}=\\{s_1,s_2\\}$, Actions: $\mathcal{A}=\\{a_1, a_2\\}$, Transition probability $P(s_1  \, | \, s_1, a_1)=1, P(s_1  \, | \, s_1, a_2)=\frac{1}{T}, P(s_2  \, | \, s_1, a_2)=1-\frac{1}{T}, P(s_2  \, | \, s_2, a_1)=1$ for $T>1$, Reward: $r(s_1,a_1)=\frac{1}{2}, r(s_1,a_2)=1, r(s_2,a_1)=0$.
>
> This modified MDP has same optimal policy $\pi^{\star} (s_1)=a_1, \pi^{\star} (s_2)=a_1$ and optimal average reward $g^{\star} = (\frac{1}{2},0)$, but different condition number $\epsilon = \frac{1}{2}(1-\frac{1}{T})$ due to addition of the suboptimal action. Also, if we run VI with initial point $(0,0)$ on the modified MDP, it requires $\frac{\log 2}{\log T}+1$ iterations to obtain optimal policy. Specifically, for $k <  \frac{\log 2}{\log T}+1$, the near-optimal policy $\pi_k(s_1)=a_2$ whereas optimal policy $\pi^{\star}(s_1)=a_1$. Furthermore, as $T \rightarrow 1$,   $\frac{\log 2}{\log T}+1 \rightarrow \infty$, indicating that the number of iterations required by VI can become arbitrarily large. Additionally, the condition number $\epsilon = \frac{1}{2}(1-\frac{1}{T})$ converges to $0$ as $T\rightarrow 1$. Therefore, this toy example shows that adding a suboptimal action to MDP can force an arbitrarily large number of iterations for VI to obtain optimal policy with arbitrarily small condition number $\epsilon$. This result is summarized in the table and a similar example can be found in Figure 1 of [1, Section 4].
>
>
> We hope this example addresses the reviewer's concern by illustrating that adding a sub-optimal state-action pair to the MDP **could** worsen the convergence in the worst-case analysis in the VI-type setups. We acknowledge that the reviewer's intuition may hold in more complex MDPs, where characterizing the condition number $\epsilon$ could be an interesting direction for future work. Lastly, we note that in our convergence analysis presented in Sections F.3 and G.3 of the appendix, the condition number $\epsilon$ seems to be an essential factor in arguing for the convergence of VI.
>
> [1] M. Zurek, Y. Chen, Span-Based Optimal Sample Complexity for Weakly Communicating and General Average Reward MDPs, arXiv preprint:2403.11477, 2024.
>
> &nbsp;
>
> **Table: Comparison of Original and Modified MDPs**
> | **MDP**         | **Number of Iterations of VI**        | **Condition number $\epsilon$**                      |
> |------------------|---------------------------------------------------|-----------------------------------------|
> | Original MDP     | 1                              | $\infty  $                              |
> | Modified MDP     |  $\frac{\log 2}{\log T} + 1$       |  $\frac{1}{2} \left( 1 - \frac{1}{T} \right)$ |

---

> > ### Comment · Reviewer_uKP2 · 2024-11-29
> > **Discussion**
> >
> > Thanks for the explanation. I have raised my score to 6. Please summary the discussion in the revision if space allowed.

---

### Official Review · Reviewer_Wd6w · 2024-11-03

**Soundness:** 3
**Presentation:** 2
**Contribution:** 2
**Rating:** 6
**Confidence:** 3

**Summary:**

The paper investigates non-asymptotic convergence rates for value iteration algorithms in average-reward Markov decision processes (MDPs). The analysis convergence rates of O(1/k) for Anchored Value Iteration and O(1/sqrt(k)) for Relaxed Value Iteration under specific conditions, leading to a better understanding of these algorithms in multichain settings. Additionally, they establish matching lower bounds that indicate the optimality of their results for weakly communicating and unichain MDPs. While the theoretical contributions are notable, the paper could benefit from clearer explanations and empirical validation to support the theoretical findings.

**Strengths:**

1. The paper provides a detailed analysis of non-asymptotic convergence rates for value iteration in average-reward MDPs, addressing a gap in existing literature.

2. It establishes both upper and lower bounds for convergence rates, contributing to a clearer understanding of the performance of value iteration algorithms.

3. The theoretical results are presented with comprehensive proofs, demonstrating rigorous mathematical foundations.

4. The work includes novel findings on the complexity lower bounds for average-reward MDPs, which may inform future research directions in this area.

**Weaknesses:**

1. The paper lacks empirical validation of its theoretical results, making it difficult to assess the practical applicability of the proposed methods in real-world scenarios. Additionally, there is a limited discussion on how these theoretical findings might translate into practical implementations or inform algorithm design.

2. The presentation is highly technical and may pose challenges for ML community researchers who are not deeply familiar with the underlying mathematical concepts, potentially limiting the accessibility of the research and further algorithm design. The authors should consider enhancing the understanding by providing more intuitive explanations, intermediate remarks and some visual/diagrammatic aids to complement their analysis.

3. There is no discussion of the computational complexity or scalability, nor any exploration of sensitivity analyses for key parameters. This omission raises questions about how well the characterized bounds hold for larger or more complex MDP settings.

**Questions:**

In addition to the mentioned weaknesses, I request authors to provide further clarifications on the following points :

1. How sensitive are the convergence rates of Relaxed Value Iteration (Rx-VI) and Anchored Value Iteration (Anc-VI) to the choice of the relaxation parameter ? Are there optimal choices that might specifically apply for different MDP classes?

2. The presented results show improved convergence rates for Anc-VI compared to Rx-VI. Are there scenarios where Rx-VI might be preferable, or is Anc-VI generally superior?

3. The paper establishes optimal complexity for standard VI and Anc-VI in certain MDP classes. Are there other value iteration variants or MDP classes not covered in this work? If so, it might be worth having a discussion on whether similar optimality is anticipated for those settings.

4. Are results presented applicable only for stationary MDPs ? It will be worth investigating how resilient these bounds are when the MDP is non-stationary i.e., unknown drifts/distribution shifts occur across time.

---

> ### Author Response · Authors · 2024-11-22
> **Official Comment by Authors**
>
> We thank the reviewer for the constructive feedback.
>
> **Weakness**
>
> (i) We performed experiments for Anc-VI and Rx-VI in toy examples such as Chainwalk and Cliffwalk, and found that Anc-VI and Rx-VI provided a practical acceleration and converges in periodic MDP while standard VI exhibits slower convergence and failed to converge. However, we were not yet able to find a practical environment that tells us when we can expect the anchoring and averaging to provide a practical acceleration, so we chose not to present them and left this issue to future work. As VI serves as the foundational basis of practical RL algorithms such as fitted value iteration and temporal difference learning, we expect that Anc-VI and Rx-VI will give insight for designing new practical algorithms or improving existing ones by incorporating the anchoring or averaging mechanism in real-world scenarios. This is certainly an interesting direction that we plan to pursue in our future work.
>
>
> (ii) We agree that our paper is highly technical and including an intuitive explanation of analysis in the main text could enhance readability. However, frankly speaking, we had a hard time finding such an intuitive explanation of the analysis without oversimplification. Our full analysis presented in the appendix involves several layers of reasoning considering the modified Bellman equation and anchoring or averaging mechanism.  For us, condensing these into a simple idea without oversimplifying the key logic is quite challenging and we worry that an incomplete summary would actually confuse the reader.
>
>
> Thus, we instead clarify how our analysis differs from other prior works with detailed historical context. Unlike prior analyses, our technique specifically utilizes the structure of Bellman operators and modified Bellman equation. By combining the previous point and insight of anchoring and averaging mechanism, our technique first achieves a non-asymptotic convergence rate while none of the prior analyses are applicable to multichain MDP setup. We elaborated on this point in the last paragraph of Section 4.
>
>
> (iii) The computational complexity of Anc-VI, Rx-VI, and VI per iteration are basically the same; the operation of adding an anchor term or averaging term is a vector-vector operator and is negligible compared to the evaluation of the Bellman operator. More precisely, rough calculation tells that number of arithmetic operations per iteration of Anc-VI, Rx-VI, and VI are $|\mathcal{S}|^2|\mathcal{A}|+|\mathcal{S}||\mathcal{A}|+2|\mathcal{S}|$, $|\mathcal{S}|^2|\mathcal{A}|+|\mathcal{S}||\mathcal{A}|+2|\mathcal{S}|$, $|\mathcal{S}|^2|\mathcal{A}|+|\mathcal{S}||\mathcal{A}|$ where $|\mathcal{S}|$ and $|\mathcal{A}|$ are cardinality of state and action space, and those are basically $O(|\mathcal{S}|^2|\mathcal{A}|)$ complexity.
>
> One common technique handling scalability issue is parallel computing, and it can be applied to the evaluation of the Bellman operator $TV$ which is basically a matrix-vector operation. As we explained in previous paragraph, the dominant computational complexity of Rx-VI and Anc-VI comes from the evaluation of Bellman operators. So, like standard-VI, Anc-VI and Rx-VI could share the benefit of parallel computing to address scalablity issue.

---

> ### Author Response · Authors · 2024-11-22
> **Official Comment by Authors**
>
> **Questions**
>
> *(i) (+Weakness (iii))*
>
> In Sections B and C of the Appendix, we obtained non-asymptotic convergence rates of Rx-VI and Anc-VI with arbitrary relaxation parameter $\lambda _k$. As Theorem 8 and 10 stated, under the specific assumption, Rx-VI and Anc-VI exhibit the following convergence rates
>
> $\|\|g^{\star} - g^{\pi_k}\|\|\leq \|\|TV^k - V^k - g^{\star}\|\|\le \frac{2\|\|V^0-h^{\star}\|\|}{\sqrt{\pi\sum^{k}_{i=K+1}\lambda_i(1-\lambda_i)}},$
>
> $$|\|g^{\star} - g^{\pi_k}\|\|\leq \|\|TV^k - V^k - g^{\star}\|\|\le2\left( 1 - \sum_{i=1}^{k} \lambda_i \prod_{j=i}^{k} (1 - \lambda_j) \right) \|\|V^0 - h^{\star}\|\|+ 2 \prod_{j=K}^{k} (1 - \lambda_j) \|\|g^\star\|\|,
> $$
>
>  respectively, where $K$ is constant depending on $\lambda_k$ and class of MDP and $k$ is number of iteration. Based on this result, we can observe how a change of $\lambda_k$ influences the convergence rate, and further, figure out the optimal choice of $\lambda_k$. For Rx-VI, by arithmetic-geometric mean inequality, $\lambda_k (1-\lambda_k) \le 1/4$ and this implies that $\lambda_k  = \frac{1}{2}$ in Theorem 1 is optimal choice independent of the class of MDP. In contrast, the optimal parameter choice of Anc-VI depends on the class of MDP because $\Pi^{k}_{j=K} (1-\lambda_j)$ term has no good upper bound like Rx-VI. But, if MDP is weakly communicating chain, then $K=0$, and due to parameter analysis in [1, Section 4], we know that $2/(n+2)$ choice gives optimal convergence rate up to constant factor.
>
>
> (ii) The reviewer raised a good point. As the reviewer pointed out, Rx-VI exhibits a worse non-asymptotic convergence rate than Anc-VI in a tabular setup. However, we expect that in the sampling setup, Rx-VI will converge better than Anc-VI due to its averaging mechanism. As we mentioned in Section 3, the averaging mechanism has an effect to stabilize randomness and ensure convergence, and further, [2, Remark 2.3] showed that it has a variance reduction effect. Since TD-learning and Q-learning also employ averaging mechanisms, analyzing the convergence of Rx-VI and Anc-VI in a sampling setup would be an interesting direction for future work.
>
>
>
> (iii) First, we clarify that our non-asymptotic results in Theorem 1 and 2 hold for any MDPs (since every MDP is multichain MDP as we defined). But, our lower bound results in Theorem 3 and 4 hold for variants of VI satisfying span condition, and standard VI, Rx-VI, and Anc-VI certainly satisfy it. As we noted in Section 5, the span condition is commonly used in the construction of complexity lower bounds for first-order optimization methods and also has been used in the lower bound construction for variants of VI [3-4].
> However, designing an algorithm that breaks the lower bound of our Theorem by violating the span condition might be possible.
>
>
> (iv) In our paper, we focus on stationary MDP, but it would be an interesting future direction to further extend our analysis to non-stationary MDP.
>
> [1] J. P. Contreras, R. Cominetti, Optimal error bounds for non-expansive fixed-point iterations in normed spaces, Mathematical Programming, 199(1), 343-374, 2023.
>
> [2] M. Bravo, R. Cominetti,  Stochastic fixed-point iterations for nonexpansive maps: Convergence and error bounds, SIAM Journal on Control and Optimization, 62(1), 191-219, 2024.
>
>  [3] V. Goyal and J, Grand-Clément, A first-order approach to accelerated value iteration. Operations Research, 71(2):517–535, 2022.
>
>  [4] J. Lee and E. K. Ryu, Accelerating Value Iteration with anchoring, Neural Information Processing Systems, 2023.

---

> > ### Comment · Reviewer_Wd6w · 2024-11-26
> >
> > I have read the detailed responses that authors have provided for my comments and other review comments. To this end, I have raised my review score and would like to thank authors for carefully responding to the highlighted concerns.

---

### Official Review · Reviewer_M4JL · 2024-11-03

**Soundness:** 3
**Presentation:** 2
**Contribution:** 2
**Rating:** 6
**Confidence:** 4

**Summary:**

This paper analyzes value iteration methods for average-reward MDPs, focusing on the case of general/multichain MDPs. The algorithms studied are based off of Halpern and Kransnosel’skii-Mann iterations, as well as relative versions (which prevent divergence of iterates). New nonasymptotic upper bounds are established for the multichain setting, and lower bounds are also provided.

**Strengths:**

Since multichain MDPs are the most general setting and yet they have received relatively little attention compared to weakly-communicating settings, I think the nonasymptotic results on multichain MDPs are of good significance.

It is nice to have lower bounds, and I think it is interesting that the standard VI is optimal in terms of the normalized iterates.

The paper is thorough in its presentation of related work, which I think will benefit the community.

**Weaknesses:**

It doesn't seem like there is a large amount of technical novelty/insight used to establish the upper bounds for multichain MDPs. I think the paper would be stronger if the authors could discuss any interesting technical novelties. It would also be nice if the main body of the paper could include some proof ideas.

Related to the above point, the proofs are not very easy to follow and could benefit from some discussion about the steps beyond just the statements of the lemmas. In particular, it would be nice for there to be discussion particularly about how the analysis differs from standard analyses for KM/Halpern iterations.

The upper bounds for general MDPs do not match the lower bounds. This is another area which would benefit from some more technical discussion in the main body of the paper.

It seems unclear what the normalized iterate performance measure is actually useful for/why should we care about it (contrasting the Bellman error, which at least under some conditions is related to actual policy performance)

**Questions:**

Are there any interesting technical novelties within the analysis of algorithms for multichain MDPs? Because value iteration is such a widely used algorithmic template in RL, I think it is easier to overlook the seemingly lower level of technical novelty of the paper if there are any simple but novel insights which might be applicable more broadly.

(Related) How does the analysis differ from standard analyses for KM/Halpern iterations?

Do you think it is possible to improve the Bellman error rate for general MDPs to match the lower bound? What are some challenges in doing so?

What is the normalized iterate performance measure useful for?

Is there a reason why it seems to be easier to get good performance for the normalized iterate performance measure rather than the Bellman error? (In the sense that standard VI works well.)

Typos:
The tables in Appendix A say "muAlti MDP"

---

> ### Author Response · Authors · 2024-11-22
> **Official Comment by Authors**
>
> We thank the reviewer for the thoughtful comments.
>
> **Weakness and Questions**
>
> *(i, ii) (+Question (i, ii))*
>
>  Thank you for the suggestion. We agree that our paper is highly technical and the proof is not easy to follow. To give a high-level idea, as we addressed in the general response, we bring in the anchoring mechanisms from fixed point iteration and minimax literature. However, none of the prior analyses are applicable to VI in the multichain setup. So, to resolve this issue, we specifically utilize the structure of Bellman operators and modified Bellman equation, and combine it with anchor and averaging mechanism to deduce non-asymptotic rates. We believe that our technique is novel and general so it should be extendable to stochastic setups such as generative models and synchronous update setups.
>
>  We agree reviewer's point that including simple proof ideas in the main text would effectively reveal the novelty of our work. However, frankly speaking, we had a hard time finding such an intuitive explanation of the analysis without oversimplification. Our full analysis presented in the appendix involves several layers of reasoning considering the modified Bellman equation and anchoring or averaging mechanism.  For us, condensing these into a simple idea without oversimplifying the key logic is quite challenging and we worry that an incomplete summary would actually confuse the reader.
>
>  Thus, we instead clarify how our analysis differs from other prior works with detailed historical context. Unlike prior analyses, our technique specifically utilizes the structure of Bellman operators and modified Bellman equation. By combining the previous point and insight of anchoring and averaging mechanism, our technique first achieves a non-asymptotic convergence rate while none of the prior analyses are applicable to multichain MDP setup. We elaborated on this point in the last paragraph of Section 4.
>
>
>
>
>
>
>
>
> *(iii) (+Question (iii))*
>
> As the reviewer pointed out, our current upper bound in Theorem 2 does not match the lower bound of Theorem 4, and we hypothesize that it is the lower bound that is loose. In our view, the $\epsilon$ defined in Theorem 3 and 4, which depend on the multichain structure of MDP, seems to be an essential factor in arguing for convergence of VI.  For this reason, we expect that finding a more complex worst-case multichain MDP than the one in Theorem 4 would lead to an improved lower bound that matches the current upper bound. This would be an interesting future direction and following the reviewer's suggestion, we added this discussion to the conclusion of the updated version.
>
>
>
> *(iv) (+Question (iv))*
>
> The normalized iterates can be useful in policy evaluation problems. When you have specific policy $\pi$ and want to obtain average reward with respect to that policy, running VI with Bellman consistency operator $T^{\pi}$ is one way and it guarantees $O(1/k)$ rate independent of the structure of MDP. Also, VI with Bellman optimality operator $T^{\star}$ gives an optimal average reward with $O(1/k)$ rate. We also point out that the normalized iterates are conventionally considered in the classical literature [1, Section 9.4].
>
> Roughly speaking, standard VI asymptotically behaves as $V^k\sim k g^\star$ where $V^k$ is $k$-th iterate. This is because in the analysis of convergence of normalized iterate, for every iteration, optimal average reward term $g^{\star}$ is generated by modified Bellman equation, and it leads to the convergence $V^k/k \rightarrow g^\star$ as $k \rightarrow \infty$. In this argument, near-optimal policy doesn't affect convergence of normalized iterate and this is why can get optimal average reward even if policy error does not converge. In contrast, in the analysis of convergence of Bellman error, near-optimal policy is a crucial factor in convergence and this leads to the fact that convergence of Bellman error implies convergence of policy error. We believe this difference comes from forms of performance measure ($V^k/k $ and $TV^k-V^k-g^{\star}$) and structure of modified Bellman equation. This is an interesting point and we clarified these facts in Section 2. For more detailed facts and proof, please refer to [1].
>
>
> **Typo**
>
> Thank you for pointing out the typo. We revised the paper accordingly.
>
>
> **Conclusion**
>
> We believe our response addresses the reviewer's main concern regarding the technical novelties. If so, we kindly ask the reviewer to adjust the score accordingly.
>
>
> [1]  M. L. Puterman, Markov Decision Processes: Discrete Stochastic Dynamic Programming, John Wiley and Sons, 2014

---

> > ### Comment · Reviewer_M4JL · 2024-11-29
> >
> > Thank you for your responses.
> >
> > Regarding the second paragraph in the response (iv), what do the authors mean by "near-optimal policy"? (In the sentences "near-optimal policy doesn't affect convergence of normalized iterate and this is why can get optimal average reward even if policy error does not converge. In contrast, in the analysis of convergence of Bellman error, near-optimal policy is a crucial factor in convergence and this leads to the fact that convergence of Bellman error implies convergence of policy error.")
> >
> > Overall, I think the response slightly addresses my concerns regarding technical novelties, but I still think that if the authors consider the proof techniques to be a main contribution then they should be presented in some capacity in the main body of the paper (with sketches, or discussion of key steps). I will increase my score to a 6 since I think that this problem is very important and this paper will likely inspire more work, despite the lack of presentation of key technical insights.

---

> ### Author Response · Authors · 2024-11-30
> **Official Comment by Authors**
>
> We thank reviewer for the response and constructive feedback.
>
> (i) Sorry for confusion. In second paragraph of the previous response (iv), the term ``near-optimal policy'' refer to the policy $\pi_k$ derived from VI at the $k$-th iterate.
>
> (ii) We acknowledge the reviewer's point. We will consider including a proof sketch in the main text, provided that space limitations allow.

---

### Official Review · Reviewer_DZ2n · 2024-11-06

**Soundness:** 4
**Presentation:** 3
**Contribution:** 3
**Rating:** 8
**Confidence:** 4

**Summary:**

This paper investigates the convergence guarantees of the value iteration algorithm (VI) and some of its variants for average-reward MDPs. The following variants are considered: standard value iteration (Standard VI), Relaxed VI (Rx-VI), Anchored VI (Anc-VI). Further, the corresponding variants of the Relative VI algorithm are considered. The paper presents several results for these methods as well related lower bounds, all of which hold non-asymptotically. The lower bounds include a broad class of VI-type algorithms that satisfy a standard span-condition. Further, one lower bound is derived for the general class of multi-chain MDPs. In terms of upper bounds, it is established that a convergence rate of $O(1/k)$ is optimal as it matches the rate asserted by the lower bounds. They also apply to multi-chain MDPs.

**Strengths:**

The paper focuses on value iteration (and some of its variants) in the average-reward setting and presents several new results on their iteration complexities, substantially improving state-of-the-art, to my best knowledge. The presented upper bounds exhibit, in my view, some key strengths: they hold non-asymptotically; they cover multi-chain MDPs, which are quite general; and some of them are provably optimal. Further, the presented lower bounds that hold for a broad class of MDPs (multi-chain ones) are quite interesting.

VI plays a key role in model-based reinforcement learning algorithms, and often appears there as a routine that must be run several times at each policy update. Hence, understanding iteration complexity of VI-style algorithms is important for the RL community.

The paper does a great job in discussing and presenting VI (and the variants) in a systematic and precise way as fixed-point iterations. Although such connections are already known, most literature fail to provide precise pointers to relevant results on convergence of fixed-point iterations from communities beyond RL.

The paper is well-written and well-organized, and the results and algorithmic ideas are presented very clearly. (I report below some typos and other relevant minor comments below.)

**Weaknesses:**

Comments about Literature
-
1- It was a good idea to provide pointer to works on average-reward RL. However, at least two key papers [1,2] appears missing. Further, (Zanette & Brunskill, 2019) must not have been cited as it deals with RL in episodic MDPs, and more importantly, it does not use any type of value iteration.

2- VI plays a key role in many algorithms designed for average-reward RL, beyond the tabular setting. For instance, it is used as a routine in algorithms developed for a wide range of settings including factored MDPs (e.g., [3]), robust MDPs [4], MDPs with reward machines [5], MDPs with options [6], etc. To further highlight the importance of VI, it might be necessary to enrich discussion in this part. I suggest the authors briefly expand the related work to address this. This makes the paper a better fit to the audience of ICLR.



Some Minor Comments
-
1. In Fact 2, you write $\pi_V$ to seemingly denote the greedy policy w.r.t. a value function $V$. However, it is not formally defined.
2. In several places (e.g., Theorem 1), you use ‘/’ to denote set exclusion. Shouldn’t it be ‘\’ or ‘\setminus’ command in LaTeX?
3. In line 317: In the second half of this rather long sentence (i.e., starting from “and in tabular setup”), you cite 4 papers, covering both discounted and undiscounted. But it is not fully clear that ‘respectively’ in the end maps which paper to which category.
4. That you wrote 3.14 in lieu of the number $\pi$ to avoid overloading notation sounds rather strange. A fix could be to use text mode to write the number $\pi$.
5. In line 456, did you mean Theorem 4 in the sentence “… improves the lower bound by constant factor of 2.”?
6. Line 394: Definition of “span” is missing.


Some Typos
-
- l. 188: There exist line ==> … exists a line
- l. 211: an near-optimal ==> a near-optimal
- l. 218: … of the Appendix ==> of Appendix OR of the appendix  ---- In some places (e.g., l. 280) you correctly used the latter form)
- l. 221: Table A.5 and A.6 ==> Tables A.5 and A.6  ---- It appeared elsewhere, e.g., l. 218, 370,…
- Corollary 2: a a general ==> remove extra ‘a’
- l. 306: … MDPs satisfies ==> … satisfy
- Theorem 4: such that such that ==> such that


References
-
[1] Jaksch, Thomas, Ronald Ortner, and Peter Auer. "Near-optimal Regret Bounds for Reinforcement Learning." Journal of Machine Learning Research 11 (2010): 1563-1600.

[2] Burnetas, Apostolos N., and Michael N. Katehakis. "Optimal adaptive policies for Markov decision processes." Mathematics of Operations Research 22.1 (1997): 222-255.

[3] Rosenberg, Aviv, and Yishay Mansour. "Oracle-efficient regret minimization in factored mdps with unknown structure." Advances in Neural Information Processing Systems 34 (2021): 11148-11159.

[4] Kumar, Navdeep, et al. "Efficient Value Iteration for s-rectangular Robust Markov Decision Processes." Forty-first International Conference on Machine Learning.

[5] Bourel, Hippolyte, et al. "Exploration in reward machines with low regret." International Conference on Artificial Intelligence and Statistics, 2023.

[6] Fruit, Ronan, et al. "Regret minimization in mdps with options without prior knowledge." Advances in Neural Information Processing Systems 30 (2017).

**Questions:**

See above.

---

> ### Author Response · Authors · 2024-11-22
> **Official Comment by Authors**
>
> We appreciate for detailed and thoughtful feedback.
>
>
> **Comments about Literature**
>
> First of all, thank you for recognizing our effort to contextualize our result in the literature of fixed point iteration and RL theory. Following the reviewer's feedback, we removed [1] and added [2-6] to our discussions of prior works. In the revised paper, we also expand the discussion on the importance of VI in RL literature to better align with the interests of the ICLR audience.
>
> **Some Minor Comments**
>
> (i) Sorry for the confusion.
>  We explicitly denoted $\pi_V$ as greedy policy in Fact 2.
>
> (ii) The reviewer raised a valid point. Yes, it should be $\setminus$. We updated this in the revised version. Thank you for pointing out our mistake.
>
> (iii) Sorry for the confusion. Those cited 4 papers only cover discounted MDP setup. We clarify this in the revised version.
>
> (iv) Thanks for the nice suggestion. Following the reviewer's suggestion, we used the text version of $\pi$, π in Theorem instead of 3.14.
>
> (v) Sorry for the confusion. Yes, it should be Theorem 4. We revised this in the updated version.
>
>
> (vi) Thank you for pointing out this.
>  We defined the span right after we introduce it.
>
> **Some Typos**
>
> Thank you for pointing out typos. We updated all typos and reflected in the revised version.

---

> ### Comment · Reviewer_DZ2n · 2024-11-29
> **Further minor comments**
>
> I have read the author response to my comments as well as the other reviews. I would like to thank the authors for their responses. They resolved the issues I raised, but there are two minor comments that must be addressed:
> - Attributing "finite horizon" to [2] is incorrect (cf. line 148). Indeed, both [1] and [2] deal with the same problem of regret minimization in average-reward MDPs: [2] derives a problem-dependent and optimal, but asymptotic, regret for unichain MDPs, whereas [1] provides finite-time regret bounds, albeit sub-optimal, for communicating MDPs.
> - [4] deals with robust MDPs, whereas [5] considers RL with reward machines. In the revised version, you swapped these pointers.
>
> Overall, I believe the paper offers novel contributions that hold for the most general class of MDPs one may encounter in RL. I therefore would like to keep my score, thus recommending acceptance.

---

> ### Author Response · Authors · 2024-11-29
> **Official Comment by Authors**
>
> Once again, we sincerely appreciate the reviewer's highly detailed feedback throughout the review process. We will correct our mistakes in the references in the revised version.

---

### Author Response · Authors · 2024-11-22
**Common Response**

First of all, we thank the reviewers for their constructive and positive feedback. We are excited to see that the reviewers generally agree that our results are valuable. However, there were some questions regarding the novelty of our proof technique.

Our results are indeed obtained by adapting the anchoring mechanism and its proof technique from fixed point iteration and minimax optimization literature to the present RL theory setup. The prior results and proof techniques were not immediately applicable to value iteration (VI) in the multichain setup, and it was crucial to utilize the structure of the Bellman operator for our analysis.

Although we did not make this point in our paper, we view our new proof technique to be a major contribution of this work. The arguments are sufficiently general so that they should be extendable to stochastic setups with generative models, and we believe that such follow-up work will lead to insights into designing more effective practical RL algorithms.

In the end, however, the novelty of a proof technique and its extendibility are subjective and speculative things to judge. Perhaps, the objective way to evaluate our contribution is to consider the strength and value of the results presented in our current paper. We believe our results, the first non-asymptotic rate on multichain MDPs and optimality with respect to the new lower bounds, are novel and valuable results, and we ask the reviewers to consider this to be an indication that our proof technique is novel and valuable.

---

### Meta-Review · Area_Chair_aWbs · 2024-12-21

**Metareview:**

This submission studies convergence guarantees of value iteration algorithms for average-reward MDPs.

This paper gives non-asymptotic rates for value iteration algorithms for average-reward MDPs as well as related lower bounds. The authors show that a convergence rate of $O(1/k)$ is optimal as it matches the rate asserted by the lower bounds.

 These theoretical contributions could be of interest to the RL theory community. The reviewers also voted unanimously for acceptance.

**Additional Comments On Reviewer Discussion:**

The reviewers raised concerns regarding missing necessary pointers to relevant results, clarity of the proof, as well as significance of the new results . However, the authors provided detailed responses which successfully addressed those concerns and resulted in improved scores.

---

### Decision · Program_Chairs · 2025-01-22

Accept (Poster)